



Atmospheric
Chemistry
and Physics

# Intercomparison in spatial distributions and temporal trends derived from multi-source satellite aerosol products

**Jing Wei[1], Yiran Peng[1], Rashed Mahmood[2], Lin Sun[3], and Jianping Guo[4]**

[1]Ministry of Education Key Laboratory for Earth System Modeling, Department of Earth System Science,
Tsinghua University, Beijing, China
[2]Department of Atmospheric Science, School of Environmental Studies, China University of Geosciences,
Wuhan, Hubei, China
[3]College of Geomatics, Shandong University of Science and Technology, Qingdao Shandong, China
[4]State Key Laboratory of Severe Weather, Chinese Academy of Meteorological Sciences, Beijing, China

**Correspondence:** Yiran Peng (pyiran@mail.tsinghua.edu.cn)

**Abstract.** Satellite-derived aerosol products provide long-term and large-scale observations for analysing aerosol distributions and variations, climate-scale aerosol simulations, and aerosol–climate interactions. Therefore, a better understanding of the consistencies and differences among multiple aerosol products is important. The objective of this study is to compare 11 global monthly aerosol optical depth (AOD) products, which are the European Space Agency Climate Change Initiative (ESA-CCI) Advanced Along-Track Scanning Radiometer (AATSR), Advanced Very High Resolution Radiometer (AVHRR), Multi-angle Imaging SpectroRadiometer (MISR), Moderate Resolution Imaging Spectroradiometer (MODIS), Sea-viewing Wide Field-of-view Sensor (SeaWiFS), Visible Infrared Imaging Radiometer (VIIRS), and POLarization and Directionality of the Earth's Reflectance (POLDER) products. AErosol RObotic NEtwork (AERONET) Version 3 Level 2.0 monthly measurements at 308 sites around the world are selected for comparison. Our results illustrate that the spatial distributions and temporal variations of most aerosol products are highly consistent globally but exhibit certain differences on regional and site scales. In general, the AATSR Dual View (ADV) and SeaWiFS products show the lowest spatial coverage with numerous missing values, while the MODIS products can cover most areas (average of 87 %) of the world. The best performance is observed in September–October–November (SON) and the worst is in June–July–August (JJA). All the products perform unsatisfactorily over northern Africa and Middle East, southern and eastern Asia, and their coastal areas due to the influence from surface brightness and human activities. In general, the MODIS products show the best agreement with the AERONET-based AOD values on different spatial scales among all the products. Furthermore, all aerosol products can capture the correct aerosol trends at most cases, especially in areas where aerosols change significantly. The MODIS products perform best in capturing the global temporal variations in aerosols. These results provide a reference for users to select appropriate aerosol products for their particular studies.

## 1 Introduction

Atmospheric aerosols originating from both natural and anthropogenic sources have noticeable effects on the ecological environment, climate change, urban air quality, and human health; these issues also attract increasing attention from national governments and scientists (Cao et al., 2012; Guo et al., 2016, 2017; Li et al., 2011, 2017; Pöschl, 2005). On the one hand, the increase in anthropogenic aerosols over the past century has significantly affected the radiation budget balance by scattering or absorbing solar radiation and by changing cloud microphysical properties (Ramanathan et al., 2001; Rosenfeld et al., 2008). On the other hand, fine-particulate matter greatly endangers human health by causing various respiratory and cardiovascular diseases (Brauer et al., 2012; Bartell et al., 2013; Crouse et al., 2012). How-

ever, due to the complex sources, compositions, and short lifetimes of atmospheric aerosol particles, large uncertainties exist in the estimation of aerosol–climate forcing and health effects. To better understand the spatial and temporal variability of aerosol distributions from regional to global scales, long-term data records with reasonable accuracy are needed as benchmarks to evaluate aerosol effects based on climate model simulations.

Since the 20th century, several aerosol ground-based observation networks, such as the worldwide AErosol RObotic NEtwork (AERONET), Interagency Monitoring of Protected Visual Environments (IMPROVE), European Monitoring and Evaluation Programme (EMEP), and Chinese Sun Hazemeter Network (CSHNET), have been established. The monitoring stations are sparsely distributed, and the observation periods at different sites vary across a large range due to instrumental or weather conditions. Therefore, ground-based observational data are limited to representing aerosol characteristics in long-term and large-scale studies. For the last few decades, satellite instruments have been launched with increasing capability for remote sensing of aerosol measurements, which have provided long-term data records with wide spatial coverage. Meanwhile, an abundance of mature aerosol retrieval algorithms has been developed according to the characteristics of different satellite sensors and atmospheric radiative transfer models, and these algorithms have been successfully applied to generate global-coverage aerosol products for over 10 years. These satellite instruments include the Advanced Very High Resolution Radiometer (AVHRR), Total Ozone Mapping Spectrometer (TOMS), Advanced Along-Track Scanning Radiometer (AATSR), Multi-angle Imaging SpectroRadiometer (MISR), Moderate Resolution Imaging Spectroradiometer (MODIS), Sea-viewing Wide Field-of-view Sensor (SeaWiFS), Visible Infrared Imaging Radiometer (VIIRS), Polarization and Directionality of Earth's Reflectance (POLDER), and Cloud-Aerosol Lidar with Orthogonal Polarization (CALIPSO).

Based on these long-term space-borne aerosol products, numerous researchers have begun to explore the spatial and temporal variations in aerosols on regional and global scales as well as the potential climate effects of aerosols. For example, Guo et al. (2011) analysed the temporal and spatial distributions and trends in aerosol optical depth (AOD) over eight typical regions in China by combining TOMS (1980–2001) and Terra MODIS (2000–2008, Collection 5.1, C5.1) aerosol products. Hsu et al. (2012) explored the global and regional AOD trends over land and the oceans from 1997 to 2010 based on the SeaWiFS monthly aerosol products. Nabat et al. (2013) used different satellite-derived monthly AOD products (e.g. MODIS, MISR, and SeaWiFS) and model datasets to create a 4-D climatology of the monthly tropospheric AOD distribution and analyse the variations from 1979 to 2009 over Europe, the Mediterranean Sea, and northern Africa. Zhao et al. (2013) analysed the AVHRR AOD datasets over the global oceans and explored the effects of

subpixel cloud contamination on aerosol retrievals from 1981 to 2009. Floutsi et al. (2016) examined the spatio-temporal variations in the AOD, fine-particle fraction and Ångström exponent over the Mediterranean basin from 2002 to 2014 with the Aqua MODIS C6 aerosol products. Klingmüller et al. (2016) studied the aerosol trends over the Middle East and explored the effects of rainfall, soil moisture, and surface winds on aerosols with Terra MODIS C6 aerosol products from 2000 to 2015. Mehta et al. (2016) presented the spatio-temporal AOD variations and their spatial correlations globally and over six subregions using the Terra MODIS (C5.1) and MISR monthly products from 2001 to 2014. Sayer et al. (2018a) extracted and compared the AOD distributions and variations using multi-satellite monthly aerosol products (e.g. VIIRS, Aqua MODIS, and MISR) over the main oceans (e.g. Tropical Pacific and North and South Atlantic oceans). Sogacheva et al. (2018) discussed the spatial and seasonal variations in aerosols over China based on 2 decades of multi-satellite observations using AATSR (1995–2012) and Terra MODIS (2000–2017, C6.1) aerosol products.

In most of the above studies, satellite-derived aerosol products are arbitrarily selected for research applications by simply following the usage in previous studies or are based on data availability. However, noticeable inconsistencies exist among the aerosol datasets generated from different satellite sensors and aerosol retrieval algorithms. Few studies have focused on exploring the similarities and differences among aerosol datasets (Holzer-Popp et al., 2013; Nabat et al., 2013; De Leeuw et al., 2015; Sayer et al., 2018a). The selection of an accurate and appropriate aerosol product that represents the long-term aerosol variations and trends for their respective studies is of great importance for users, especially interdisciplinary scholars. Otherwise, problematic aerosol characteristics will inevitably lead to questionable conclusions.

The objective of this study is to comprehensively investigate the consistencies and differences in aerosol characteristics among multiple global aerosol products from satellites. For this purpose, a total of 11 of the most up-to-date global aerosol products are selected in this paper, including the European Space Agency's Climate Change Initiative (ESA-CCI) products: AATSR Dual View (AATSR-ADV), AATSR Swansea University (AATSR-SU), AATSR-Oxford-RAL Retrieval of Aerosol and Cloud (AATSR-ORAC) and AATSR-ENSEMBLE (AATSR-EN), which cover the period from 2002 to 2012, AVHRR (2006–2011), MISR (2000–2017), Terra MODIS (2000–2017), Aqua MODIS (2002–2017), POLDER (2005–2013), Sea-WiFS (1997–2010), and VIIRS (2012–2017) products. It should be noted that, while these data up to 2017 are used in the current study, many of them (i.e. MISR, MODIS, and VIIRS) are ongoing as the instruments are still returning data. The newest AERONET Version 3 monthly AOD measurements at 308 globally distributed sites over land and the oceans are collected for comparison.

This paper is organized as follows: descriptions of the 10 CE1 satellite global aerosol products and AERONET data sources are provided in Sect. 2. In Sect. 3, the matching methods for the comparisons, the calculation approaches for the aerosol distributions and trends, and quantitative evaluation metrics are presented. The statistical evaluation results for the monthly AOD retrieval are presented in Sect. 4. In Sect. 5, the regional and global AOD distributions are analysed and comparisons of the aerosol trends are provided in Sect. 6. A summary and conclusions are presented in the final section.

## 2 Data description

### 2.1 Satellite-derived aerosol products

#### 2.1.1 ESA-CCI aerosol products

Four typical ESA-CCI global-coverage aerosol products are selected, including the AATSR-ADV, AATSR-SU, AATSR-ORAC, and AATSR-EN. The AATSR-ADV product is generated using the dual-view (ADV, Veefkind et al., 1998a) algorithm over land and the single-view (ASV, Veefkind and de Leeuw, 1998b) algorithm over the ocean. The ADV algorithm uses the dual-view feature and K-ratio approach to eliminate the contribution from the surface to the apparent reflectance. However, this approximation is not reliable over bright surfaces or in the presence of coarse-mode aerosols. The ASV algorithm assumes the water is a dark surface at the near-infrared channel, and an ocean reflectance model is applied to correct for the effects of chlorophyll and white-caps (Kolmonen et al., 2013). The SU algorithm employs a parameterized model of the surface angular anisotropy and estimates the surface spectral reflectance using the dual-view feature over land. Over the ocean, the SU algorithm estimates the water-leaving radiance from the ocean at the red and infrared channels at both nadir and along-track view angles with a simple model (North et al., 1999; North, 2002; Bevan et al., 2012). The ORAC algorithm is an optimal estimation retrieval scheme for multispectral images (Thomas, 2009; Sayer et al., 2010), which uses a forward model to fit all the shortwave forward and nadir radiances through the DIScrete Ordinate Radiative Transfer (DISORT) model. Meanwhile, the retrieved errors for aerosol parameters are estimated by propagating the measurement and forward model uncertainties into the state space. The AATSR-EN product is integrated based on different ESA-AATSR aerosol products using likelihood estimate approaches (Holzer-Popp et al., 2013). In this study, the latest versions of the above four ESA-CCI products (Table 1) are collected.

#### 2.1.2 MISR aerosol product

The MISR aerosol product provides aerosol distributions over both land and oceans. Over land, MISR is initially based on the dense dark vegetation (DDV) algorithm (King et al.,

**Table 1.** Summary of satellite-derived and ground-observed monthly aerosol products used in this study.

| Product | Version | Spatial resolution | Temporal availability | Scientific data set | Literature |
|---|---|---|---|---|---|
| AATSR-ADV | V2.31 | 1° × 1° | May 2002–April 2012 | AOD550_mean | Veefkind et al. (1998), Veefkind and de Leeuw (1998) |
| AATSR-SU | V4.3 | 1° × 1° | May 2002–April 2012 | AOD550_mean | North et al. (1999), North (2002), Bevan et al. (2012) |
| AATSR-ORAC | V4.01 | 1° × 1° | July 2002–April 2012 | AOD550_mean | Thomas et al. (2009), Sayer et al. (2010) |
| AATSR-EN | V2.6 | 1° × 1° | July 2002–April 2012 | AOD550 | Holzer-Popp et al. (2013) |
| MISR | V23 | 0.5° × 0.5° | March 2000–December 2017 | Optical depth average (550 nm) | Witek et al. (2018) |
| MOD08 | C6.1 | 1° × 1° | March 2000–December 2017 | AOD_550_Dark_Target_Deep_Blue_Combined_Mean | Hsu et al. (2019), Sayer et al. (2014), Wei et al. (2019a, b, c) |
| MYD08 | C6.1 | 1° × 1° | July 2002–December 2017 | AOD_550_Dark_Target_Deep_Blue_Combined_Mean | Hsu et al. (2019), Sayer et al. (2014), Wei et al. (2019a, b, c) |
| SeaWiFS | V4 | 0.5° × 0.5° | September 1997–December 2010 | aerosol_optical_thickness_550_land | Hsu et al. (2019) |
| AVHRR | V1 | 1° × 1° | January 2006–December 2011 | aerosol_optical_thickness_550_land_ocean_mean | Hsu et al. (2017) |
| VIIRS | V1 | 1° × 1° | March 2012–December 2017 | Aerosol_Optical_Thickness_550_Land_Ocean_Mean | Hsu et al. (2019) |
| POLDER | V1.1 | 1° × 1° | March 2005–October 2013 | AOD565 | Dubovik et al. (2011, 2014) |
| AERONET | V3 | site | January 2003–December 2010 | AOD | Giles et al. (2019) |

1992) and uses spatial contrasts to explore an empirical orthogonal function of the angular variations in apparent reflectance. Then, the MISR product is used to estimate the scene path radiance and determine the best-fitting aerosol models. Additionally, the spectral and angular shapes of the reflectance function are assumed to be constant. The algorithm is continuously revised and developed to generate the AOD product with high spatial resolution (4.4 km) based on the primary underlying physical assumptions. Over the ocean, water bodies are essentially assumed to be black at the visible and near-infrared wavelengths, and with an additional assumption of an ocean aerosol model, the aerosol retrieval is realized using the radiative transfer theory. MISR multi-angle radiances are used to improve the definition of aerosol models for aerosol retrieval. Recently, a new method was introduced to improve dark-water aerosol retrievals by considering the entire range of cost functions associated with each aerosol mixture, and a new aerosol retrieval confidence index was established to screen high-AOD retrieval blunders caused by cloud contamination or other factors (Witek et al., 2018). In this study, the latest MISR Version 23 monthly aerosol product was selected (Table 1).

### 2.1.3 MODIS aerosol products

The MODIS aerosol products are generated from three well-known algorithms, including the dark-target (DT) algorithms over both the oceans and dark land and the deep-blue (DB) algorithm over bright and dark land. Over the oceans, the DT algorithm considers the water to be a dark surface from visible to longer wavelengths and neglects the water surface reflectance. Over land, the DT algorithm assumes that the surface reflectances in the visible channels exhibit stable statistical empirical relationships with the 2.1 μm apparent reflectance over the dark-target surfaces (Kaufman et al., 1997; Levy et al., 2007). The aerosol retrieval can be realized based on the atmospheric radiative transfer model using the look-up table (LUT) approach. In contrast, the DB algorithm is designed to overcome the flaw in the DT algorithms and realizes aerosol retrieval over bright surfaces, where the surface reflectance in the visible channels is estimated based on the pre-calculated surface reflectance database using atmospherically corrected data from the long time series of measurements. Both algorithms have been continuously improved with refinements and improvements made to the above aerosol retrieval algorithms, and the second-generation operational DT (Levy et al., 2013) and the enhanced DB algorithms (Hsu et al., 2019) were used to generate the latest Collection C6.1 (C6.1) aerosol products (Sayer et al., 2019; Wei et al., 2019a, b). The C6.1 DT land algorithm has an update to reduce biases in urban areas by using a different surface reflectance model (Gupta et al., 2016). The C6.1 DB land algorithm has some updates in surface reflectance estimation using three different approaches depending on land cover type and performing aerosol re-

trievals based on pre-calculated LUTs for a range of solar and satellite-viewing geometry, aerosol and surface conditions (Hsu et al., 2019). To increase the data coverage, a new combined DT and DB (DTB) dataset was recently generated according to the independently derived MODIS monthly normalized difference vegetation index (NDVI) products that leverage the strengths of the DT and DB algorithms (Sayer et al., 2014). In this study, the newly released Terra (MOD08) and Aqua (MYD08) C6.1 DTB monthly aerosol products (Sayer et al., 2019; Wei et al., 2019c) are selected (Table 1).

### 2.1.4 SeaWiFS, AVHRR, and VIIRS aerosol products

The SeaWiFS, AVHRR, and VIIRS aerosol products over land are generated from the same DB algorithm as MODIS but with some extensions and refinements (Hsu et al., 2017, 2019). Over the ocean, these products are based on the Satellite Ocean Aerosol Retrieval (SOAR) algorithm (Sayer et al., 2012, 2017, 2018b) and include three phases: the selection of suitable pixels to exclude the sun glint, clouds, or suspect of excessively turbid water; pixel-level retrieval; and a post-processing stage (data downscaling and quality assurance). In the SOAR algorithm, the aerosol retrieval simultaneously retrieved the AOD at 550 nm, fine-mode fraction (FMF) and the best fit aerosol optical model based on the linear interpolation of pre-calculated LUTs through the Vector LInearized Discrete Ordinate Radiative Transfer (VLIDORT) model. In this study, the newly released SeaWiFS Version 4, AVHRR Version 1, and VIIRS Version 1 monthly aerosol products are selected (Table 1).

### 2.1.5 POLDER aerosol product

The POLDER/PARASOL aerosol product is generated using the Generalized Retrieval of Aerosol and Surface Properties (GRASP) algorithm over land and ocean (Dubovik et al., 2011, 2014). The GRASP algorithm is based on the AERONET inversion algorithm and was developed for enhanced characterization of aerosol properties from spectral, multi-angular polarimetric remote-sensing observations. POLDER is of great interest as it builds on the design of the forthcoming multi-viewing, multi-channel, multi-polarization (3 MI) instrument (Marbach et al., 2015). POLDER has provided a variety of aerosol characteristics, including spectral AOD, single-scattering albedo (SSA), and Ångström exponent (AE); however, the data are only available at latitudes equatorward of 60°. It should be noted that POLDER AOD is defined at 565 nm. The effect of this restriction on the global analysis is expected to be small because high latitudes are frequently unavailable due to clouds, snow, polar night, and continental land masses (Sayer et al., 2018a). In this study, the latest POLDER Version 1.1 monthly aerosol products are selected (Table 1).

## 2.2 AERONET ground measurements

AERONET is a widely used ground-based observation network with long-term data records at numerous monitoring sites around the world. The AOD observations are available over a wide spectral range from visible to near-infrared channels (0.34–1.02 μm), and they are measured with a high temporal resolution of 15 min and a low bias of 0.01–0.02. The data quality has been divided into three levels (L): L1.0 (unscreened), L1.5 (cloud screened), and L2.0 (cloud screened and quality assured) (Holben et al., 1998; Smirnov et al., 2000, 2009). Meanwhile, the instantaneous AOD observations are further processed and released at daily and monthly levels. In the current study, the newly released AERONET Version 3 L2.0 monthly AOD observations (Giles et al., 2019) are collected and compared with the multi-source satellite-derived monthly aerosol products over land and ocean. The globe is divided into 10 custom regions of land, four coastal areas, and four open-ocean areas, as illustrated in Fig. 1. Table 1 summarizes all the data sources used in this study.

## 3 Methodology

### 3.1 Spatial comparison

For multi-satellite aerosol products, the monthly retrievals at 550 nm are collected from the listed scientific dataset (SDS, Table 1) and used for the current analysis in this study. Due to different spatial resolutions, all datasets are uniformly integrated into $1° \times 1°$ grid cells using the bidirectional linear interpolation method. For comparison, monthly retrievals for diverse aerosol products are defined by the pixel centred on the AERONET site, and the corresponding monthly AERONET AOD is regarded as the true value. Notably, the AERONET sites do not provide the AOD observations at 550 nm; thus, the AOD values at 550 nm are interpolated using the Ångström exponent ($\alpha$) algorithm from 440–675 nm using the AERONET AOD measured at those wavelengths (Eq. 1). Moreover, the spatial coverage for satellite-derived aerosol products is calculated through the area-weighting approach where each grid cell is weighted by cosine of central latitude. The annual mean AOD value is averaged from at least eight available monthly values over 1 year.

$$AOD_{550} = AOD_{\lambda}(550/\lambda)^{-\alpha} \tag{1}$$

### 3.2 Temporal trend

The satellite-derived and AERONET-measured monthly mean AOD values are selected for temporal variation and trend analysis; however, to remove the noticeable influence of the annual cycle, the data are first deseasonalized by calculating the time series of the AOD anomalies. An anomaly is defined as the difference between the monthly mean AOD in 1 year and the monthly AOD average over all years. Then, the ordinary least squares fitting method (Lai and Wei, 1978; Zdaniuk, 2014) is selected to minimize the sum of residual squares of all observed values and obtain the coefficient of the linear regression slope that represents the temporal trend (AOD yr$^{-1}$, Eq. 2).

$$Y_t = aX_t + b + N_t, t = 1, \ldots, T, \tag{2}$$

where $Y_t$ is the AOD time series anomaly, a is the trend (AOD yr$^{-1}$), b is the offset term, and $X_t$ is the annual time series ($X_t = t/12$, where t is the individual months in the time series). The term $N_t$ represents the residuals in the time series. However, large-scale systems and seasonal patterns can persist for weeks to months and affect the temporal aerosol trend, and the 1-month lag autocorrelation in the time series is considered in the AOD trend analyses. The uncertainty ($\sigma$, represents 1 standard deviation) in the estimated trend is approximated by the following approach (Weatherhead et al., 1998):

$$\sigma \approx \frac{\sigma_N}{N^{3/2}} \sqrt{\frac{R'}{1 - R'}}, \tag{3}$$

where $\sigma_N$ is the standard deviation of the residuals $N_t$ on the fit and $R'$ is the autocorrelation coefficient. The mathematical value and uncertainty range of the AOD trend are represented by a $\pm\sigma$. The statistical significance of the trend is assessed using the two-side test approach, where p values less than 0.05 or 0.1 represent trends that are significant at the 95 % or 90 % confidence levels, respectively. TS1 The p value represents the probability of obtaining results at least as extreme as those found, under the null hypothesis of there being no relation between AOD and time.

Moreover, the false-discovery rate (FDR) is also considered to decrease the fraction of false positives for multiple-hypothesis testing (Wilks, 2006). The discovery refers to the rejection of a hypothesis, and a false discovery is an incorrect rejection of a hypothesis, and the FDR is the likelihood that such a rejection occurs. The well-known Benjamini–Hochberg procedure is selected to calculate the FDR in this paper (Benjamini and Hochberg, 1995). This procedure begins by ordering the m hypothesis by ascending p values, where $P_i$ is the p value at the ith position with the associated hypothesis $H_i$. Let k be the largest i for which

$$P_i \leq \frac{i}{m}\alpha. \tag{4}$$

Reject hypotheses $i = 1, 2, 3 \ldots k$. In this study, the FDR is controlled for all tests at the expected level ($\alpha = 0.05$), where no more than 5 % of the significant results are in fact false positives.

### 3.3 Statistical metrics

To quantitatively evaluate the quality and uncertainty of the retrievals, four main metrics are calculated between the

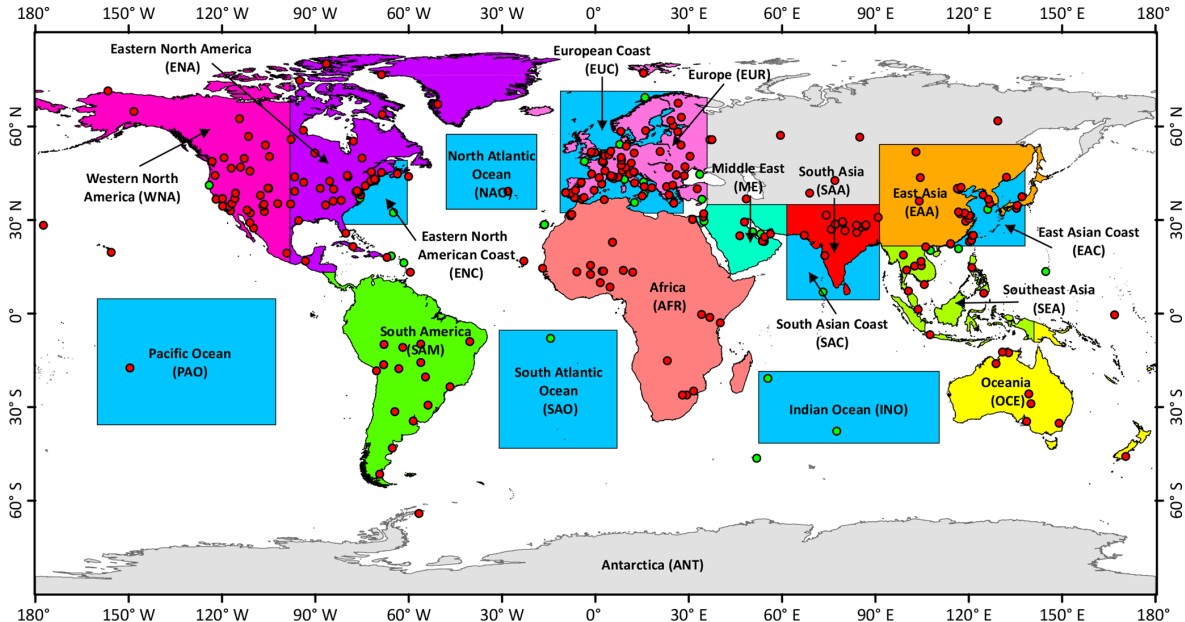

**Figure 1.** Locations of the AERONET sites and geographical bounds of the custom regions used in this study, where red and green dots represent land and ocean sites, respectively.

satellite-derived AOD ($AOD_S$) and AERONET-based AOD ($AOD_A$). The Pearson product-moment correlation coefficient ($R$) is selected to measure the linear correlation between the above two variables. The mean absolute error (MAE, Eq. 5) represents the overall estimation accuracy. The root mean square error (RMSE, Eq. 6) and relative mean bias (RMB, Eq. 7) represent the overall estimation uncertainty, where RMB > 1.0 or RMB < 1.0 indicate the over- or underestimation uncertainty. Although several satellite products have provided an expected level of uncertainty on AOD, this refers to level 2 products and is not applicable to the level 3 products in studies like this one. Level 3 uncertainty estimates have not yet been developed for these AOD products. Moreover, to quantify the performance of each satellite aerosol product in capturing aerosol trends, an additional correct-trend percentage (CTP) is defined as the percentage of sites where the satellite-derived and AERONET-based trends are consistent within each uncertainty (1 standard deviation) or not, and they are compared by combining trend uncertainties in quadrature by assuming that the uncertainty estimates from different data sets are independent.

$$MAE = \frac{1}{n}\sum_{i=1}^{n}|AOD_S - AOD_A| \tag{5}$$

$$RMSE = \sqrt{\frac{1}{n}\sum_{i=1}^{n}(AOD_S - AOD_A)^2} \tag{6}$$

$$RMB = \frac{1}{n}\sum_{i=1}^{n}|AOD_S/AOD_A| \tag{7}$$

## 4 Performance of monthly aerosol products

### 4.1 Global-scale comparison

Figure 2 compares the monthly $AOD_S$ values derived from 10 satellite aerosol products and $AOD_A$ values at a total of 268 available AERONET sites for the common period 2006–2010 throughout the world (VIIRS data are not discussed in Sect. 4 because they start in 2012). Table S1 in the Supplement also summarizes the comparison of $AOD_S$ and $AOD_A$ values from the 10 products over land and ocean for the common period 2006–2010. Due to the differences in aerosol retrieval algorithms and satellite observation conditions, the spatial coverage is not uniform among these products, which results in noticeable differences in the number of data collections (sample size, $N$). The four ESA-CCI monthly aerosol products show similar overall performance with comparable evaluation metrics. The AOD retrievals ($N = 7938$–$9467$) agree well with $AOD_A$ ($R = 0.7$–$0.8$), with MAE values ranging from 0.07 to 0.09 and RMSE values ranging from 0.13 to 0.15. Among them, the AATSR-SU (AATSR-ADV) product shows the best (worst) performance with the smallest (largest) differences on the global scale. These results are consistent with those reported by a previous study (de Leeuw et al., 2015). The AVHRR $AOD_S$ values ($N = 8382$) are well correlated with the AERONET $AOD_A$ values with MAE and RMSE of 0.077 and 0.145, respectively. The Terra MISR product provides a sample size of 8418, which is smaller than the Terra MODIS sample size ($N = 9196$) and is possibly due to the narrower swath width. MISR $AOD_S$ values are highly correlated with the ground-

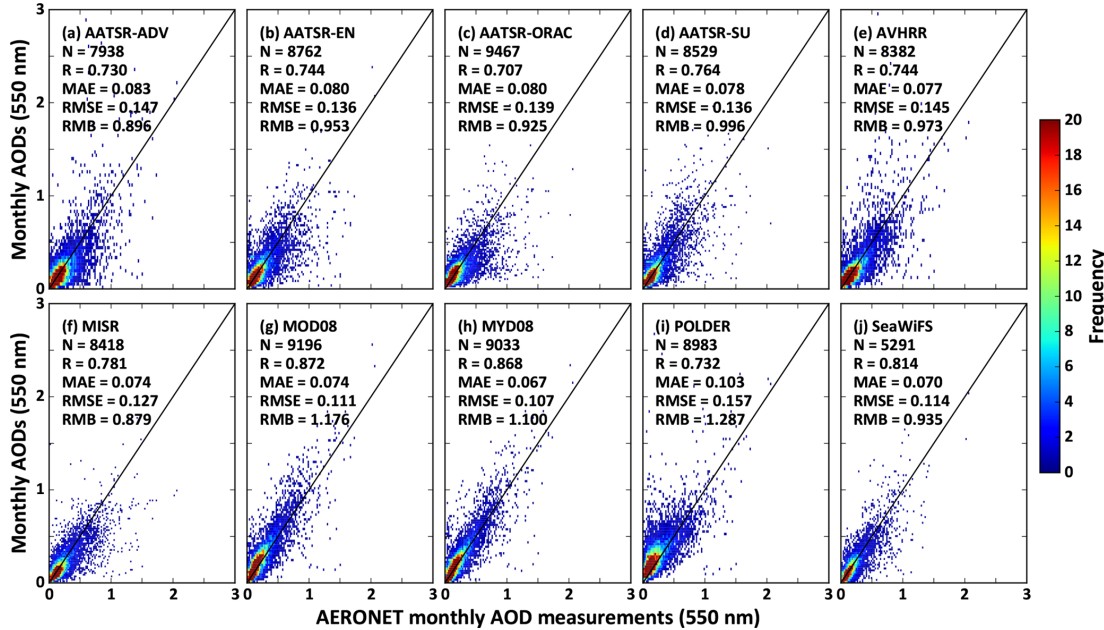

**Figure 2.** Density scatter plots of the monthly averages of satellite-derived $AOD_S$ vs. AERONET $AOD_A$ throughout the world.

measured $AOD_A$ values ($R = 0.781$), with a MAE of 0.074 and RMSE of 0.127. The Terra MODIS product is generally better than the MISR product with a high correlation and low RMSE. Due to the afternoon imaging time, the Aqua MODIS product provides approximately 2 % fewer data collections than Terra MODIS, but it exhibits superior performance in terms of most of the evaluation metrics (i.e. $R = 0.868$, MAE $= 0.067$, and RMSE $= 0.107$) among all 10 products. In contrast, the POLDER product exhibits an inferior performance, with the largest MAE and RMSE errors among all the products, significantly overestimating the monthly aerosol loads (RMB $= 1.287$). This result could be partially attributed to the relatively low accuracy of cloud detection results in the current POLDER product, and an upcoming version of the POLDER product with an advanced algorithm will improve the AOD retrievals. The SeaWiFS product has the smallest sample size, which provides 33 %–44 % fewer data collections than other products but exhibits overall good performance. The reason is partly the temporary failures during the studied time period that cause missing monthly data. In general, both MODIS and POLDER products overestimate the monthly average aerosol loads and other products underestimate them, especially the MISR and AATSR-ADV products.

## 4.2 Continent-scale comparison

Aerosol characteristics over land are more diverse than those over the ocean due to complex surface structures, varying aerosol compositions, and influences of natural and human factors. Therefore, this section focuses on the comparison between monthly $AOD_S$ and $AOD_A$ on the continental scale over land. For this purpose, 10 main customized conti-

nents (Fig. 1) are considered, including eastern North America (ENA), western North America (WNA), South America (SAM), Europe (EUR), Africa (AFR), the Middle East (ME), southern Asia (SAA), eastern Asia (EAA), southeastern Asia (SEA), and Oceania (OCE). Figure 3 shows the continent-scale performance for 10 $AOD_S$ products for the common period 2006–2010 over land, and the statistical results are given in Table S2.

The results show some common features of the 10 $AOD_S$ products. In general, a large number of data samples are collected over Europe and North America due to intensive ground-based observation sites. In contrast, the sample sizes are small over the Middle East, eastern Asia, southeastern Asia, and Oceania due to the sparse observation sites and algorithm limitations over the high-brightness underlying surfaces. Most aerosol products exhibit good performances with low MAE and RMSE values less than 0.06 and 0.08 over Europe, North America, and Oceania. The main reason for this result is that the relatively high vegetation coverage and dark underlying surface allow for more accurate aerosol retrievals by different aerosol algorithms (Wei et al., 2018a, b). However, poor performances with large MAE and RMSE values occur over southern Asia, eastern Asia, Africa, and the Middle East. This result is mainly due to the complex and bright underlying surfaces (e.g. desert, bare land, and urban areas), as well as intense human activities, which increase the difficulty of aerosol estimation (Wei and Sun, 2017; Wei et al., 2017, 2018a, b, 2019d). Overall, most aerosol products overestimate the monthly AOD over North America and Oceania, while general underestimations occur over South America, Africa, and eastern Asia.

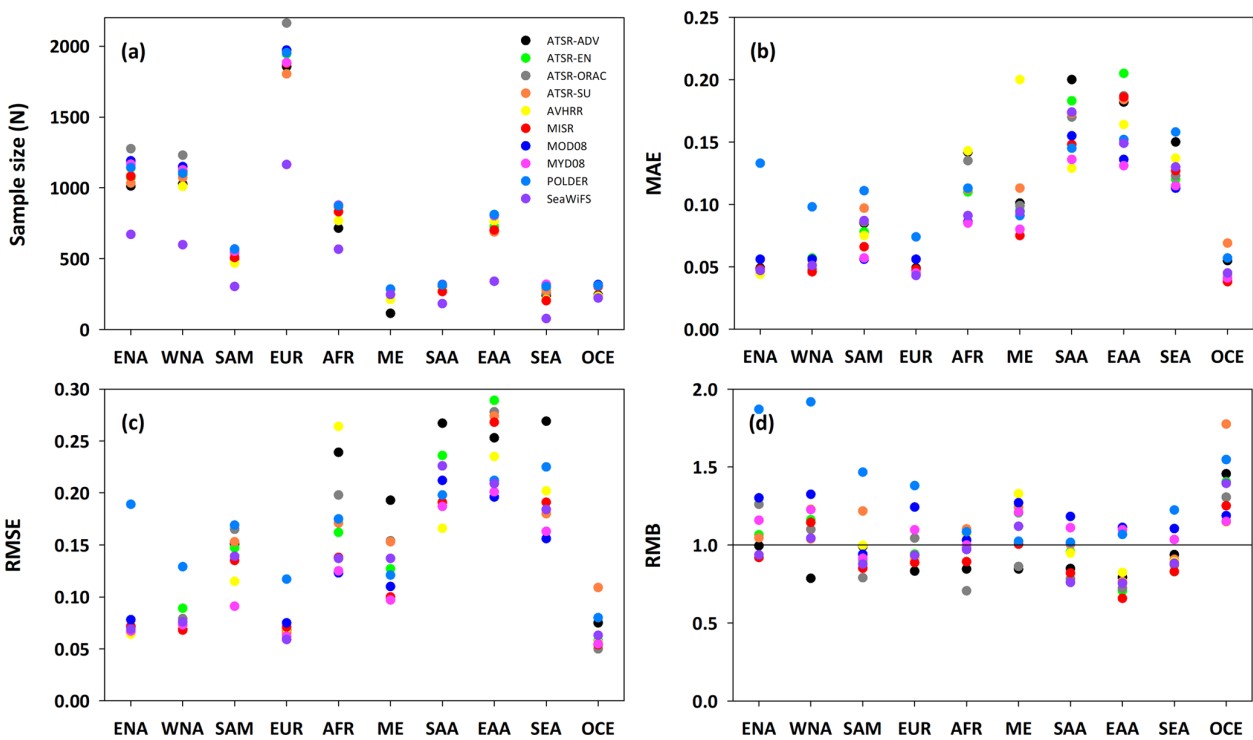

**Figure 3.** Continent-scale performance for satellite-derived monthly $AOD_S$ against AERONET monthly $AOD_A$ measurements from 2006 to 2010 in terms of **(a)** sample size ($N$), **(b)** MAE, **(c)** RMSE, and **(d)** RMB.

The performance of each $AOD_S$ product is also distinct in each specific region. In general, the AATSR-ORAC, POLDER, and MODIS products provide a larger number of data samples than the other products. In particular, the AATSR-ADV product provides fewer data samples over the Middle East than over the other regions because the ADV algorithm cannot be applied in bright desert areas. In terms of the retrieved AODs, all the products perform almost equally with similar evaluation metrics (e.g. MAE, RMSE) over North America, Europe, and Oceania, except for the POLDER product. In the other regions, large differences are found among the 10 $AOD_S$ products. In general, the MODIS and MISR products exhibit better performances (with low MAE and RMSE values) than the other products over South America, Africa, the Middle East, eastern Asia, and southeastern Asia. The POLDER and MODIS products overestimate the monthly aerosol loads over most continents, especially America and Europe. In contrast, the AATSR-ORAC, AATSR-ADV, and MISR products usually underestimate the monthly aerosol loads except for a few specific regions (i.e. western North America and Oceania).

### 4.3 Site-scale comparison

The global- and continent-scale comparisons show the overall performance of ten satellite aerosol products. However, the selected AERONET sites are unevenly distributed around the world, with most sites concentrated in densely populated land regions. Therefore, the site-scale comparison at a total of 308 available sites is performed in this section. For this purpose, four main evaluation metrics are calculated, including the sample size ($N$), MAE, RMSE, and RMB. For statistical significance, only those sites with at least half a year of observations (six matchups) are used for analysis. Figures 4–6 shows the site-scale performance map for $AOD_S$ against $AOD_A$, and Table 2 summarizes the percentages of the sites within a certain range of evaluation metrics for all $AOD_S$ products in the common period 2006–2010.

Figure 4 illustrates the number of data collections for the different $AOD_S$ products at each site over both land and ocean, where the black dots represent an insufficient number of matchups. Most products can provide enough data samples at more than 95 % of the sites around the world, especially the AATSR and MODIS products. However, the SeaWiFS product has approximately 21 % of the sites with no or few matchup samples, which are mainly distributed over North America, Europe, Asia, and southeastern Asia. The AATSR-ADV product has approximately 8 % of the sites lacking matched samples, which are spread over northern Africa, southern Europe, the Middle East, and central Asia. The main reason for this result is that the ADV algorithm cannot be adequately applied over bright surfaces. Moreover, the sites with no matched data samples from the POLDER product are concentrated in high-latitude areas because the

**Table 2.** Percentage of sites within certain ranges of evaluation metrics for different satellite-derived monthly $AOD_S$ products from 2006 to 2010.

| Products | $N$ | MAE | | RMSE | | RMB | | |
|---|---|---|---|---|---|---|---|---|
| | > 6 | < 0.08 | > 0.12 | < 0.08 | > 0.12 | < 0.8 | [0.9, 1.1] | > 1.2 |
| AATSR-ADV | 92 | 59 | 19 | 47 | 28 | 35 | 22 | 16 |
| AATSR-EN | 96 | 66 | 16 | 55 | 25 | 26 | 28 | 23 |
| AATSR-ORAC | 99 | 69 | 20 | 56 | 28 | 26 | 24 | 30 |
| AATSR-SU | 95 | 63 | 19 | 56 | 28 | 18 | 32 | 17 |
| AVHRR | 96 | 67 | 17 | 57 | 25 | 20 | 29 | 18 |
| MISR | 95 | 69 | 15 | 50 | 25 | 25 | 30 | 23 |
| MOD08 | 99 | 67 | 12 | 52 | 23 | 9 | 14 | 54 |
| MYD08 | 97 | 71 | 12 | 60 | 21 | 12 | 24 | 34 |
| POLDER | 93 | 35 | 31 | 21 | 47 | 2 | 17 | 61 |
| SeaWiFS | 79 | 56 | 14 | 46 | 21 | 12 | 27 | 24 |

POLDER algorithm is designed for aerosol retrieval between 60° latitude CE2.

Figures 5 and 6 plot the MAE and RMSE errors between $AOD_S$ and $AOD_A$ at each site across the world. The MAE and RMSE maps have very similar spatial patterns for each aerosol product. Good performances are exhibited at most North American and European sites with low MAE and RMSE values less than 0.04 and 0.06. The sites with poor performances are mainly aggregated in northern Africa, eastern Asia, and southern Asia, where the MAE and RMSE values are generally greater than 0.16 and 0.20. This result indicates that the overall performance of the aerosol products on the site scale is spatially heterogeneous and highly dependent on the type of underlying surfaces and the impact of human activities. Among the 10 aerosol products, the Aqua MODIS product shows the best performance, having a large percentage of sites (71 % and 60 %) with MAE and RMSE values less than 0.08 throughout the world. By contrast, the POLDER product performs the worst, having more than 31 % and 47 % of the sites with MAE and RMSE values greater than 0.12.

Figure 7 shows the spatial distribution of the site-scale $AOD_S$ bias. For the 10 products, only 14 %–32 % of the sites show good estimations, with RMB values ranging from 0.9 to 1.1. The POLDER and MOD08 products overestimate at most sites, especially in North America and Europe, and more than 54 % and 61 % of the sites show significant overestimations (RMB > 1.2) according to the statistics in Table 2. The other products mostly underestimate at sites over Europe, Africa, the Middle East, and Asia and overestimate at sites over South America and Australia.

## 5 AOD spatial coverage and distribution

### 5.1 Global and regional distribution

In this section, we compare the AOD distribution among the 11 aerosol products (VIIRS data are included). Figure 8 illustrates the global spatial coverage and mean value of all $AOD_S$ products for their respective available periods from 1997 to 2017. There are several missing monthly data records for the AATSR-ADV, AATSR-ORAC, AVHRR, and SeaWiFS products, which are given in Table S3.

All the aerosol products present a similar and obvious annual cycle, with high spatial coverage in August and September and low coverage in December and January (Fig. 8a). In general, the MODIS and VIIRS products provide the largest spatial coverage, covering more than 86 % of the area of the world. In contrast, the AATSR-ADV and SeaWiFS products have the lowest spatial coverage, with global averages of 68 % and 69 %. The AATSR-EN, AATSR-SU, AVHRR, and POLDER products have similar spatial coverages, with an average of 72 %–76 %. The spatial coverage decreased significantly as the SeaWiFS and POLDER satellite services approached their end stages. Figure 8b shows similar annual variations among the 11 $AOD_S$ products, with the peak from July to September and the trough from November to January. The POLDER product exhibits the highest AOD values among all products, while the SeaWiFS and MISR products show the lowest values. The other products have relatively similar $AOD_S$ values, ranging from 0.13 to 0.18. Finally, we found that the VIIRS product is almost identical to the Aqua MODIS $AOD_S$, as shown in Fig. 8, due to the similar satellite parameters and algorithms. Considering the relatively short data records of VIIRS, we will not include these data in the subsequent comparison and analysis.

Considering the remarkable seasonal variations, we plot the seasonal spatial distributions of the 10 aerosol products for their common period 2006–2010 in Fig. 9. Meanwhile, we also reproduce the satellite-derived global $AOD_S$

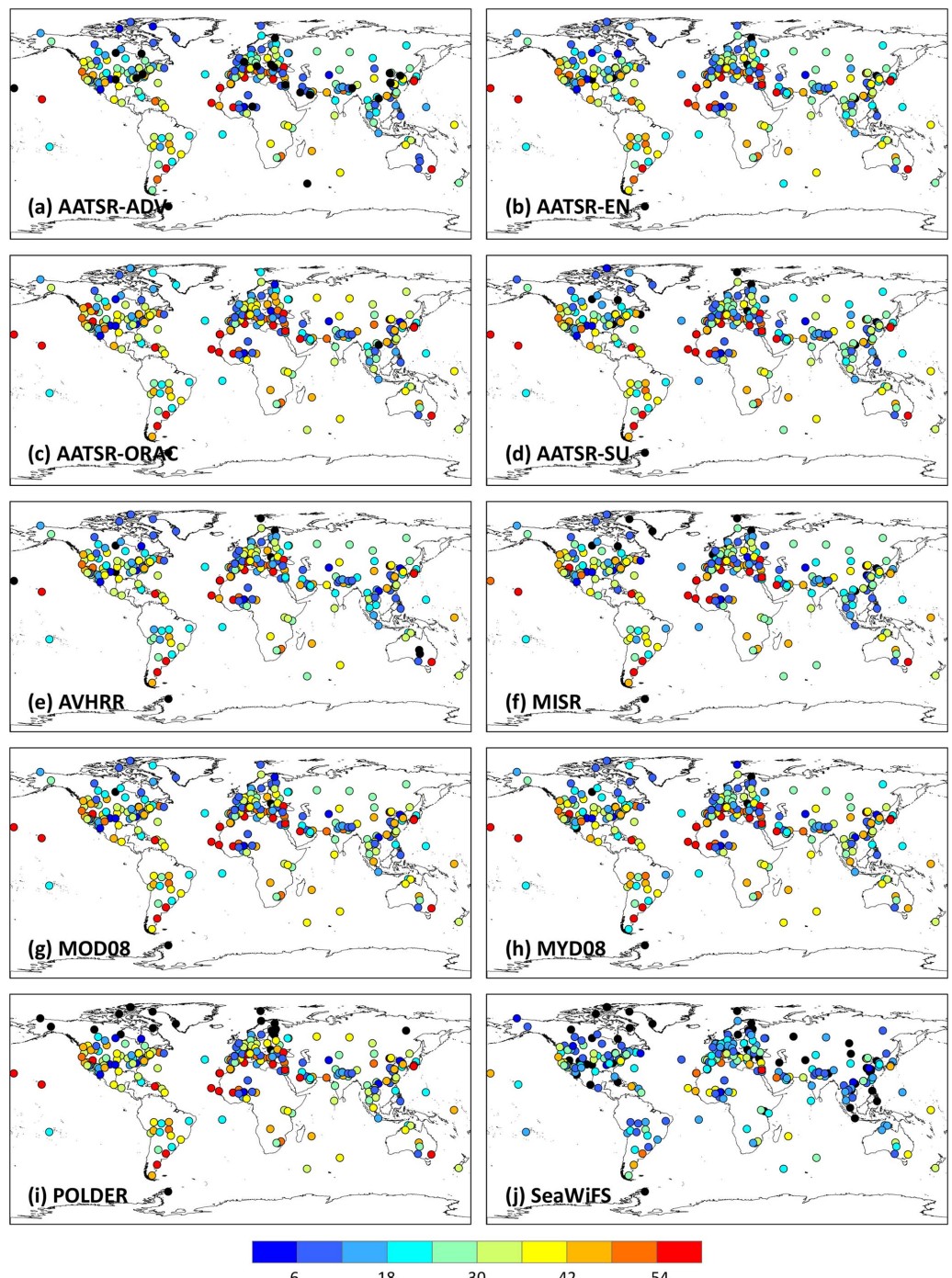

**Figure 4.** Site-scale performance map for satellite-derived monthly AOD$_S$ against AERONET monthly AOD$_A$ measurements from 2006 to 2010 in terms of sample size ($N$), where black dots represent the sites with zero matchup samples.

maps considering the common points in all datasets separately over land and ocean (Figs. S1–S2). Table 3 summarizes the average spatial coverage and AOD$_S$ values in December–January–February (DJF), March–April–May (MAM), June–July–August (JJA), and September–October–November (SON) for each product. In DJF, the space cover-

age is the lowest, with an average cover rate less than 90 % for most aerosol products, especially for AATSR-ADV product ($\sim 73$ %). The missing data are mainly for the Northern Hemisphere in winter and in high-latitude areas with bright surfaces covered by snow and ice, where most of the retrieval algorithms cannot be implemented. By contrast, the spatial

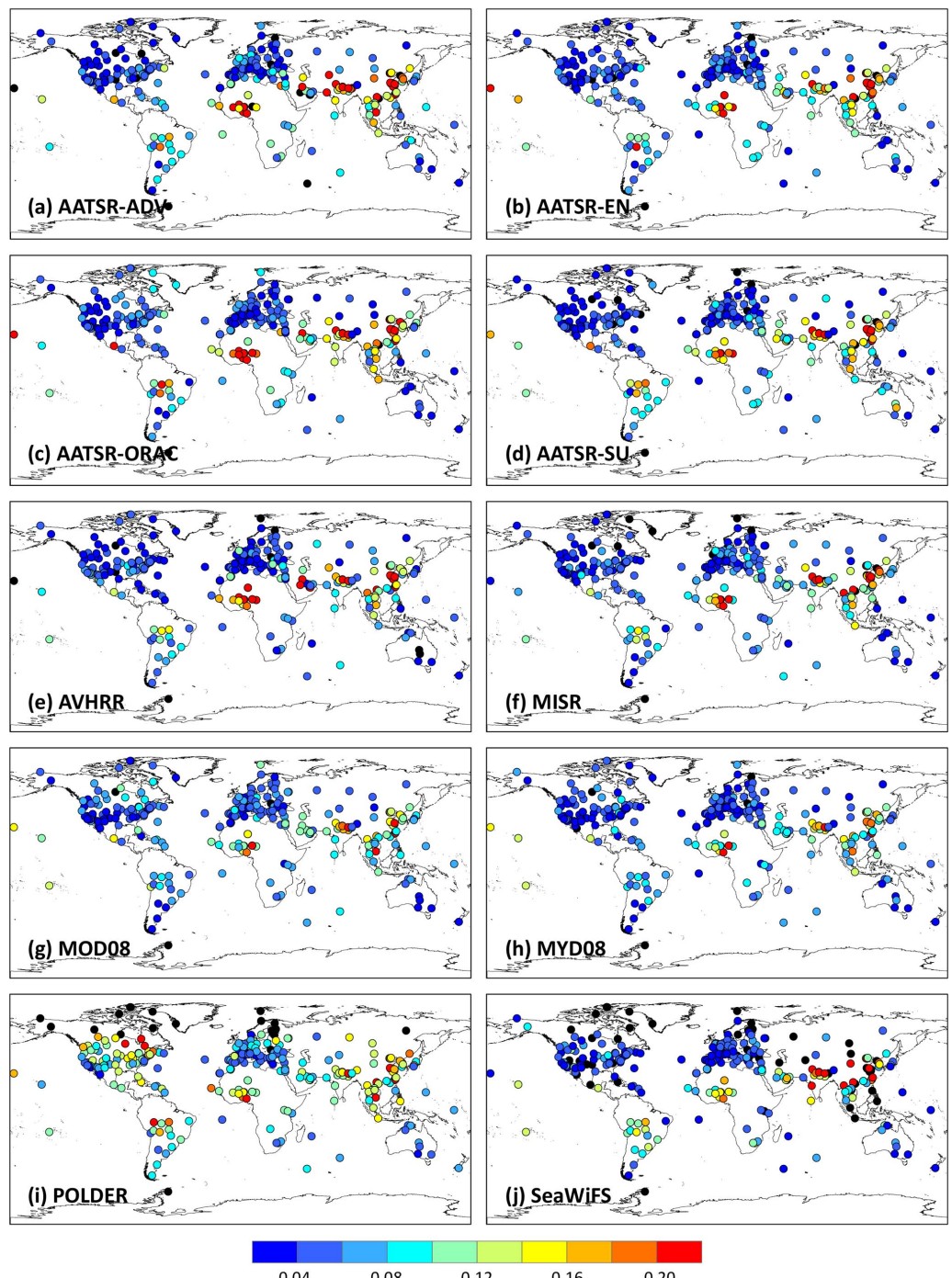

**Figure 5.** Same as Fig. 4 but for MAE.

coverage is increased in the other seasons and the highest values are always observed in SON for most aerosol products. Among these 10 products, two MODIS aerosol products can provide almost the largest spatial coverage with average cover rates of 88 %, 94 %, 93 %, and 95 % for DJF, MAM, JJA, and SON. By contrast, AATSR-ADV and POLDER products are generally narrower than other products in spatial coverage for each season.

For the spatial distribution of $AOD_S$, noticeable spatial heterogeneity occurs over land with low values in North America, Europe, and Australia and high values in northern Africa, the Middle East, southern Asia, and eastern Asia. Deserts, dry areas, and their downwind regions have $AOD_S$

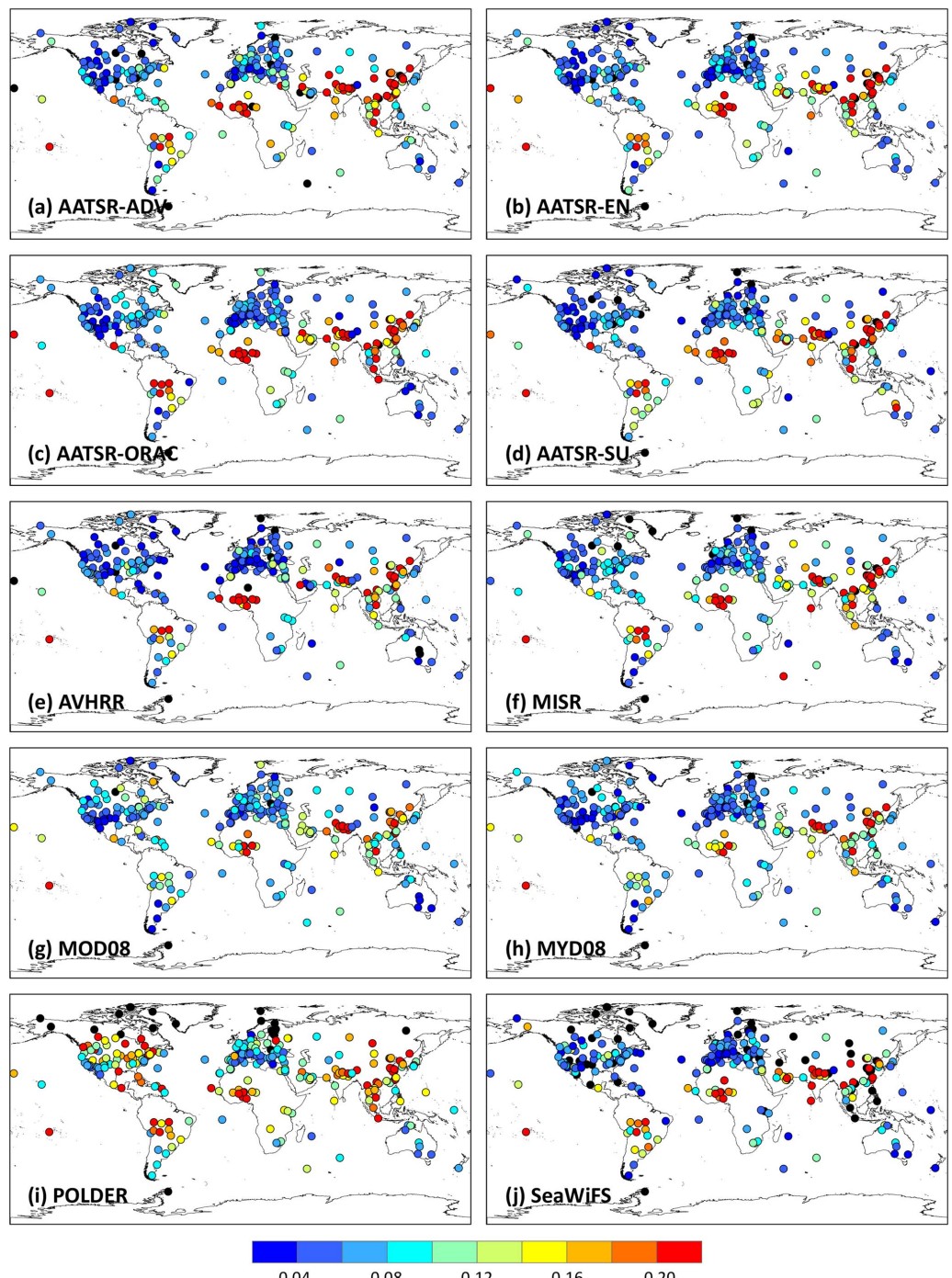

**Figure 6.** Same as Fig. 4 but for RMSE.

peaks in spring (eastern Asia) or summer (northern Africa and the Middle East) in accordance with the prevailing time of dust. Anthropogenic polluted regions exhibit peaks in high-emission seasons, such as dry seasons in the savanna and Amazon due to biomass burning, summer in eastern Asia due to the formation of large numbers of fine particles and water uptake by hygroscopic particles. There is also strong

diversity in the seasonal or annual mean $AOD_S$ over northern Africa and eastern Asia among most datasets (Fig. S1 in the Supplement). This diversity is mainly due to the different aerosol algorithms applied over bright surfaces (i.e. desert and urban areas). Both high surface reflectance and complex underlying surfaces increase the difficulty of aerosol retrieval (Wei and Sun, 2017; Wei et al., 2017, 2018a, b, 2019d). For

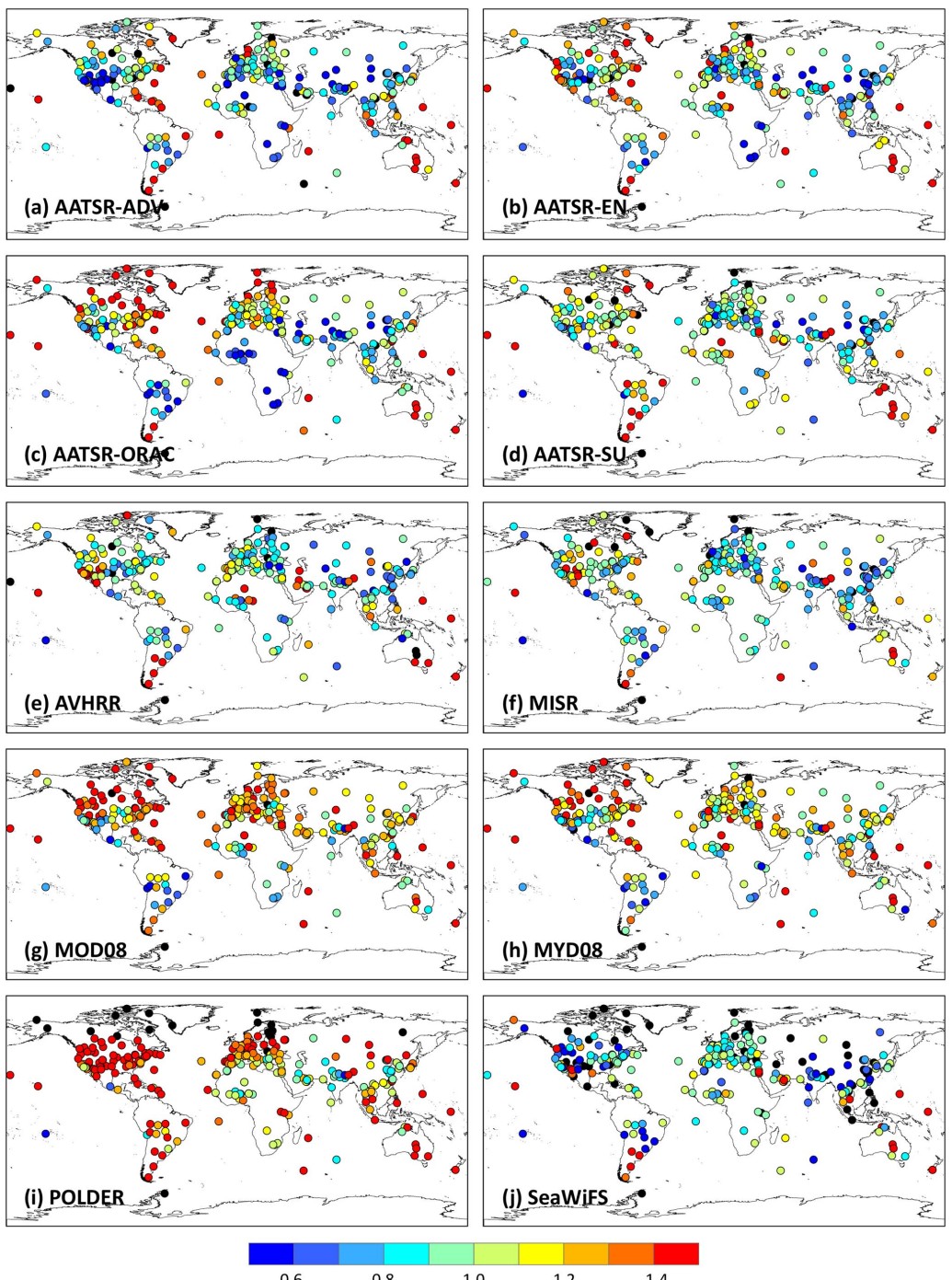

**Figure 7.** Same as Fig. 4 but for RMB.

the spatial distributions over the ocean, the seasonal and annual mean $AOD_S$ values are generally lower than 0.1 in most areas, especially open seas (Fig. S2). In coastal areas near central and northern Africa, southern Middle East, southern India, and eastern China, the $AOD_S$ values are strongly influenced by the source regions. The seasonal mean $AOD_S$ values are generally greater than 0.4, and the seasonal variation in $AOD_S$ in the downstream plume areas is consistent with that in the upstream land area.

Figures 10 and 11 plot the seasonal spatial coverage and mean $AOD_S$ values over 10 land and 8 oceanic customized regions (see Fig. 1) for each product during the common period 2006–2010. The results illustrate that the SeaWiFS and AVHRR products have much lower spatial coverage than the

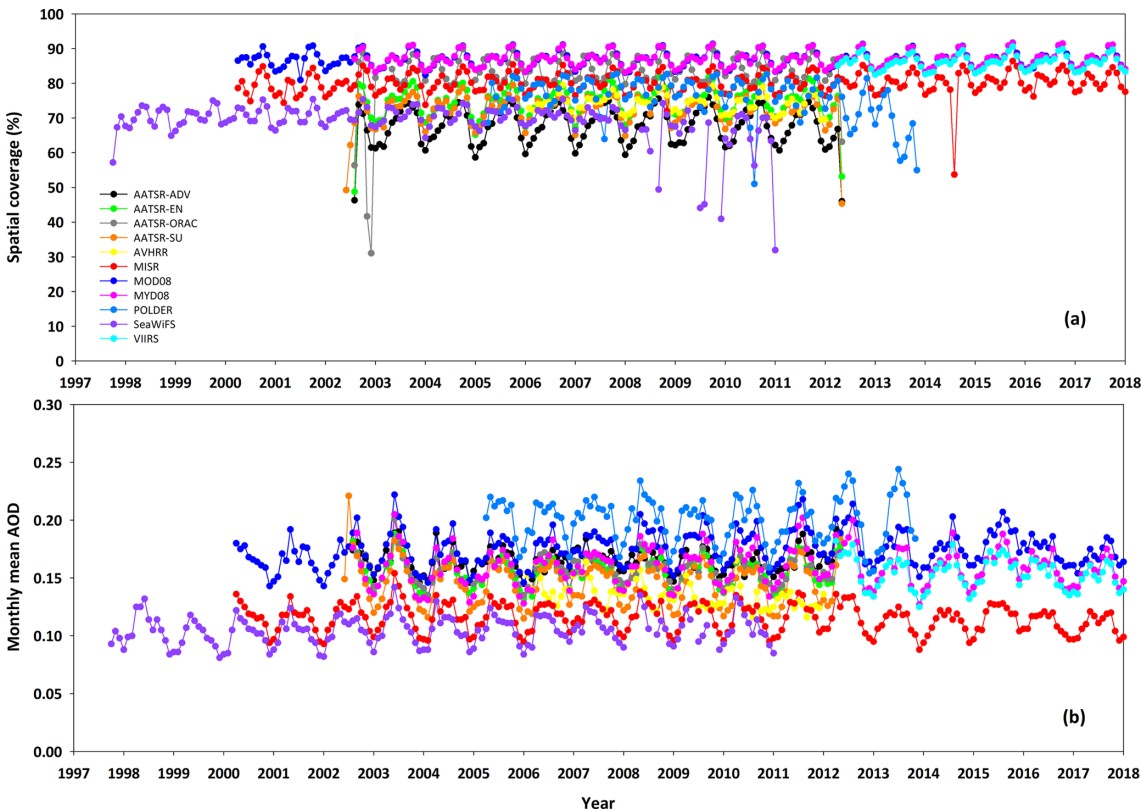

**Figure 8.** Time series of global spatial coverage and mean value of satellite-derived monthly aerosol products for their respective available periods from 1997 to 2017.

**Table 3.** Seasonal statistics of spatial coverage and global means of satellite-derived AOD$_S$ from 2006 to 2010.

| Products | Spatial coverage (%) | | | | Mean AOD | | | |
|---|---|---|---|---|---|---|---|---|
| | DJF | MAM | JJA | SON | DJF | MAM | JJA | SON |
| AATSR-ADV | 73 | 81 | 82 | 83 | $0.16 \pm 0.10$ | $0.17 \pm 0.13$ | $0.17 \pm 0.12$ | $0.16 \pm 0.10$ |
| AATSR-EN | 86 | 91 | 89 | 92 | $0.13 \pm 0.08$ | $0.16 \pm 0.13$ | $0.15 \pm 0.11$ | $0.14 \pm 0.09$ |
| AATSR-ORAC | 92 | 93 | 90 | 94 | $0.15 \pm 0.09$ | $0.16 \pm 0.10$ | $0.16 \pm 0.10$ | $0.16 \pm 0.08$ |
| AATSR-SU | 87 | 93 | 91 | 93 | $0.12 \pm 0.09$ | $0.15 \pm 0.14$ | $0.15 \pm 0.13$ | $0.13 \pm 0.09$ |
| AVHRR | 85 | 89 | 88 | 91 | $0.13 \pm 0.09$ | $0.14 \pm 0.14$ | $0.14 \pm 0.13$ | $0.13 \pm 0.09$ |
| MISR | 89 | 93 | 91 | 93 | $0.12 \pm 0.08$ | $0.13 \pm 0.12$ | $0.14 \pm 0.11$ | $0.12 \pm 0.08$ |
| MOD08 | 88 | 94 | 93 | 95 | $0.16 \pm 0.09$ | $0.19 \pm 0.14$ | $0.19 \pm 0.13$ | $0.17 \pm 0.10$ |
| MYD08 | 88 | 94 | 93 | 95 | $0.15 \pm 0.09$ | $0.17 \pm 0.14$ | $0.17 \pm 0.12$ | $0.15 \pm 0.09$ |
| POLDER | 84 | 86 | 83 | 86 | $0.19 \pm 0.13$ | $0.20 \pm 0.15$ | $0.21 \pm 0.15$ | $0.19 \pm 0.12$ |
| SeaWiFS | 82 | 88 | 85 | 88 | $0.10 \pm 0.08$ | $0.12 \pm 0.11$ | $0.13 \pm 0.12$ | $0.11 \pm 0.08$ |

other products over most land regions, especially for South America, southern Asia, and southeastern Asia. The AATSR-ADV product has the lowest spatial coverage in Africa and the Middle East due to the limitations of aerosol retrieval algorithms. Meanwhile, the POLDER product yields the minimum spatial coverage in the high latitudes due to the lack of retrievals above 60° (i.e. eastern and western North America, and Europe). In general, the range in spatial coverage of all AOD$_S$ products is greater in all seasons (especially in

DJF) over North America, Europe, and the European coast. By contrast, most aerosol products are more consistent and have higher spatial coverage in the remaining areas, especially for open seas, where the average spatial coverage can even reach up to 100 %.

For the seasonal mean AOD$_S$, the POLDER product has the highest values, and the SeaWiFS product has the lowest values over most customized regions. The AATSR-ADV product exhibits the lowest seasonal AOD values in the Mid-

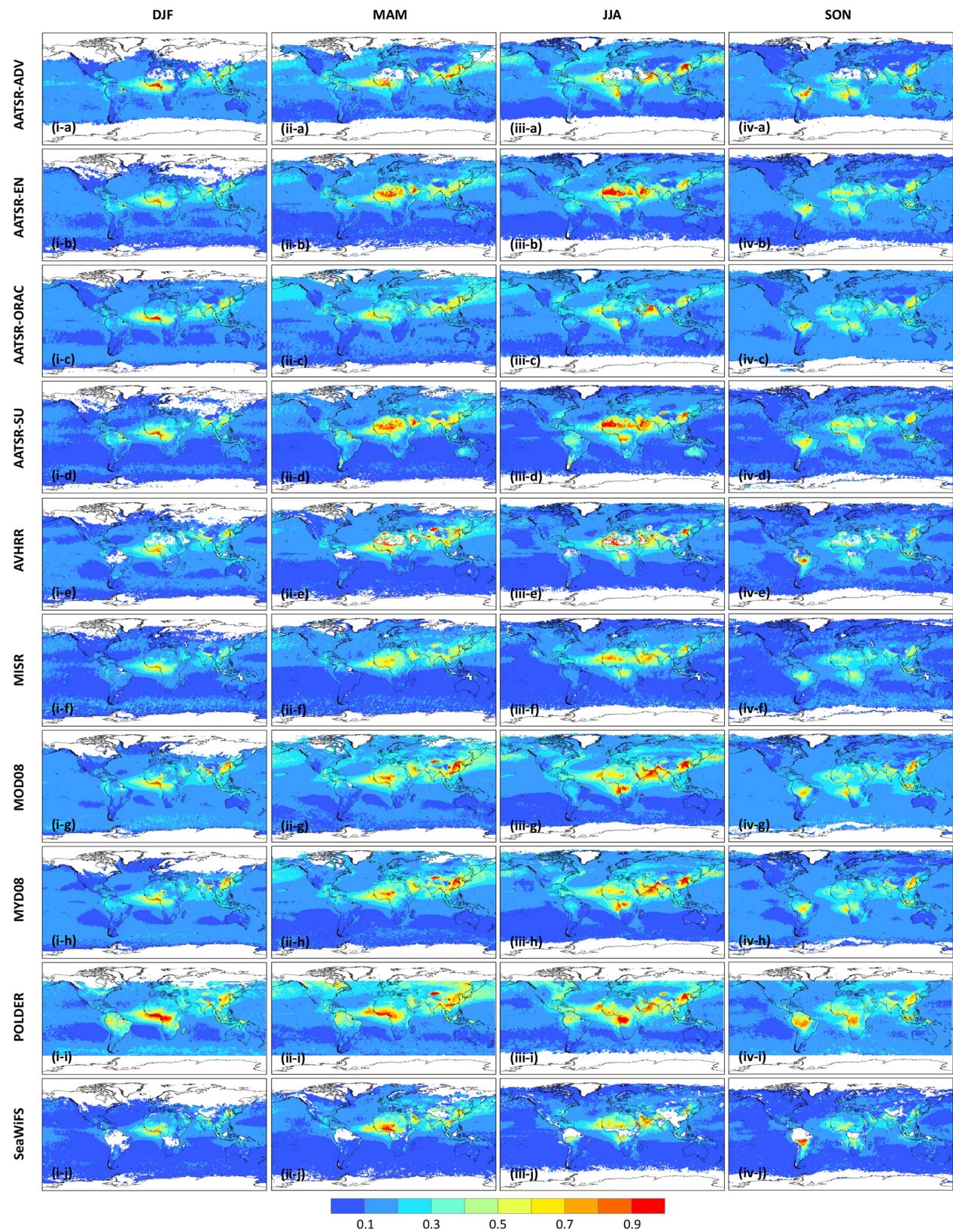

**Figure 9.** Satellite-derived global seasonal averaged AOD$_S$ maps at 550 nm from 2006 to 2010.

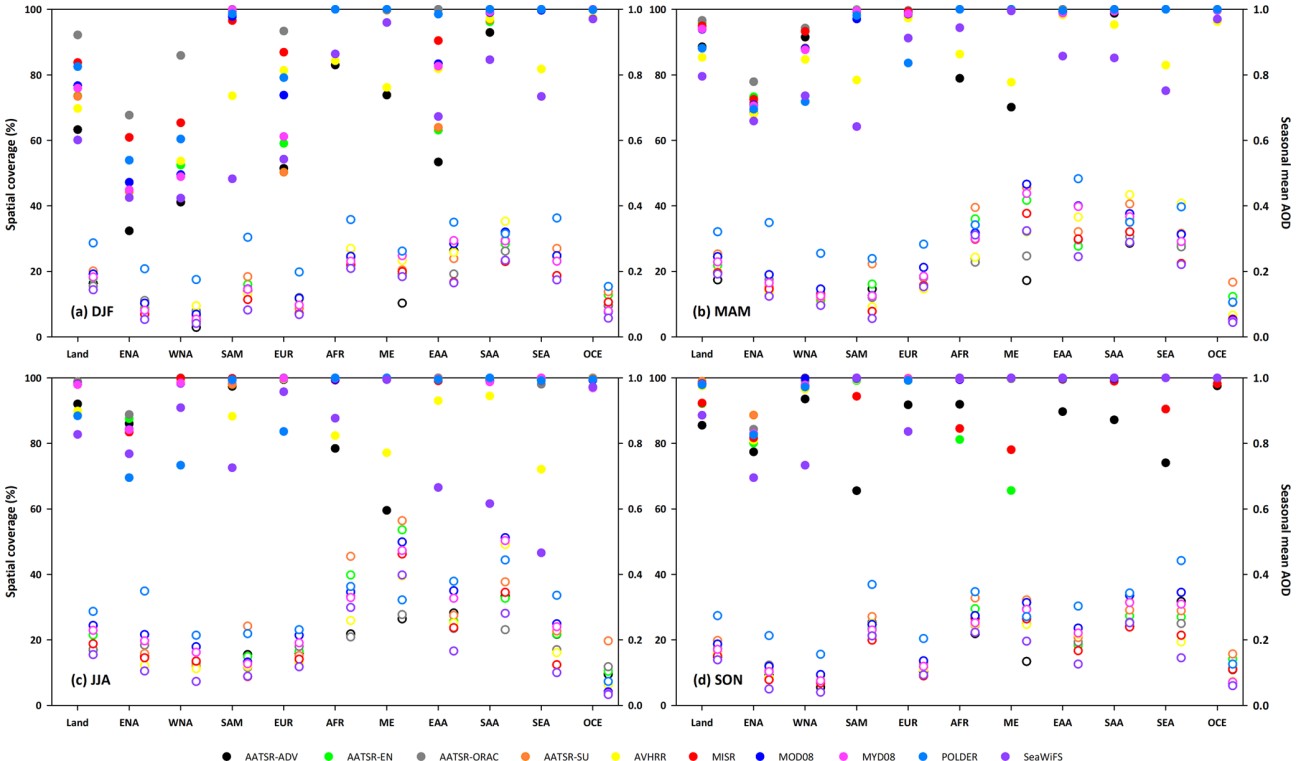

**Figure 10.** AOD$_S$ spatial coverage (marked as solid circles) and seasonal mean (marked as hollow circles) for each customized region over land (refer to Fig. 1) from 2006 to 2010.

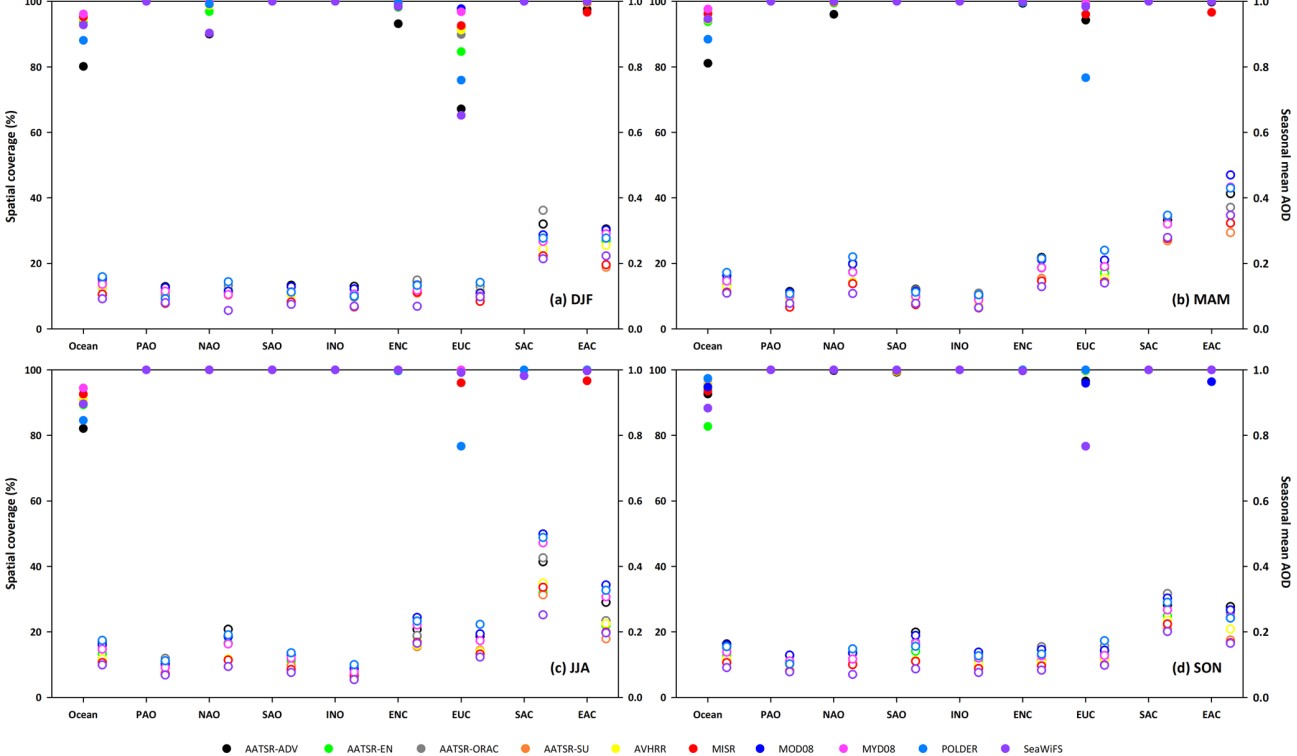

**Figure 11.** Same as Fig. 10 but for each customized region over ocean.

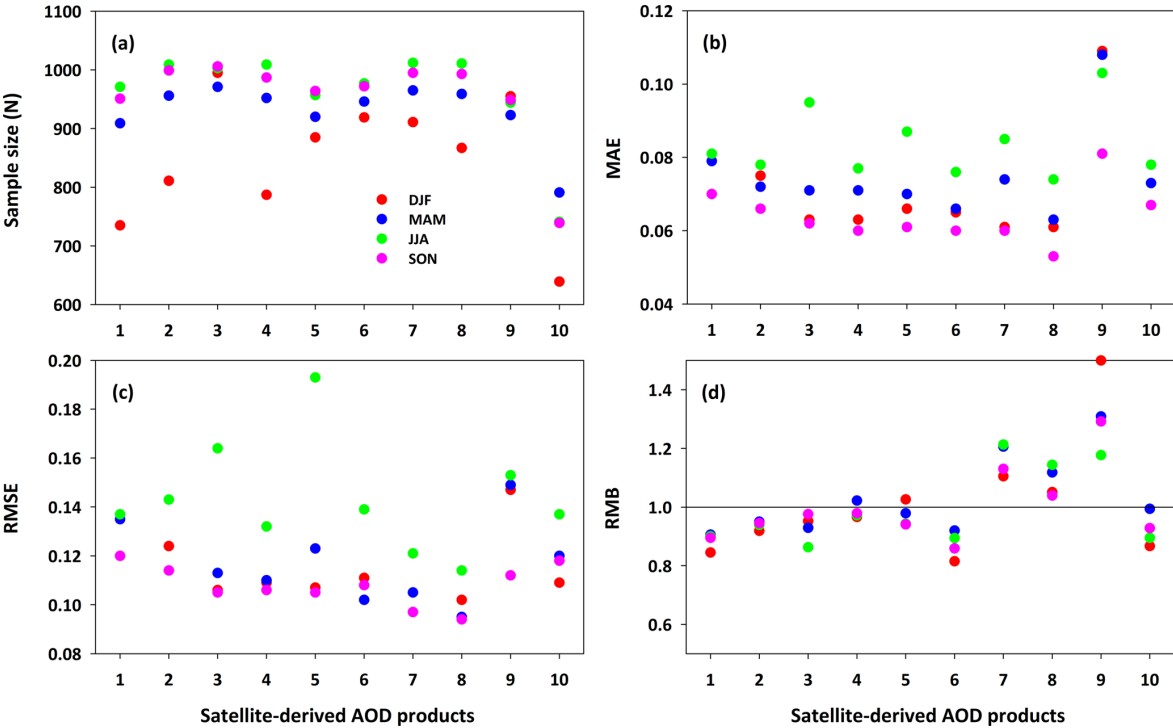

**Figure 12.** Seasonal performance for satellite-derived $AOD_S$ against AERONET $AOD_A$ measurements from 2006 to 2010 in terms of **(a)** sample size ($N$), **(b)** MAE, **(c)** RMSE, and **(d)** RMB, where numbers 1–10 on the $x$ axis represent the AATSR-ADV, AATSR-EN, AATSR-ORAC, AATSR-SU, AVHRR, MISR, MOD08, MYD08, POLDER, and SeaWiFS products.

dle East due to a large number of missing retrievals. For the remaining aerosol products, the range of the seasonal mean $AOD_S$ is greater than 0.2 over Africa, southern Asia, eastern Asia, southeastern Asia, and the coastal areas of southern Asia, and eastern Asia. The main reason for this wide range could be the complex aerosol types from multiple sources (e.g. natural dust mixed with anthropogenic fine particles) that cannot be resolved by current aerosol retrieval algorithms. For the remaining land and ocean regions, the range in seasonal AOD values is generally within 0.1 among these aerosol products. The main reason for this result may be the differences in satellite scanning widths and pixel selection during the reprocessing of the monthly aerosol products.

### 5.2 Comparison between seasonal and annual $AOD_S$ and $AOD_A$

Figure 12 compares the satellite-derived seasonal mean $AOD_S$ value for each satellite over AERONET sites with the ground-based $AOD_A$ values over land and ocean, and the statistical results are given in Table S4. The best performance with the smallest MAE (Fig. 12b) and RMSE (Fig. 12c) values are always found in SON. In contrast, the worst performances with the largest estimation uncertainties (i.e. MAE and RMSE) among the 10 aerosol products are found in JJA. In general, the MODIS and POLDER products overestimate the aerosol loads in the four seasons, and the remaining seven

aerosol products underestimate them (Fig. 12d). The performance of the AATSR-ORAC and AVHRR products is poor with large estimation uncertainties in JJA but much improved in the other three seasons. The AATSR-SU product shows the smallest estimation bias (RMB = 0.95–1.05) in all four seasons among all products. In general, the Aqua MODIS product performs best with almost all the best evaluation metrics (e.g. $N$, MAE, and RMSE) compared to the other products on the seasonal level.

In Fig. 13 we also compare the annual mean $AOD_S$ values from each satellite product with the AERONET $AOD_A$ values at available sites from 2006 to 2010. The results indicate that similar conclusions can be drawn for both seasonal and annual scales. The AATSR-SU product performs superior among the four ESA-CCI AATSR products. The AVHRR and MISR products show similar performances with close MAE (0.049 and 0.050) and RMSE (0.082 and 0.083) values but underestimate the annual mean AOD (RMB = 0.972 and 0.881). However, these products are better overall than the ESA-CCI AATSR products. The POLDER and SeaWiFS products exhibit poor performance due to the notable overestimation (RMB = 1.307) and the smallest number of matchup samples, respectively. The MODIS products have noticeably high correlations with ground measurements ($R > 0.92$), but MOD08 shows a $\sim 17\%$ overestimation. In general, the MYD08 product has the best performance

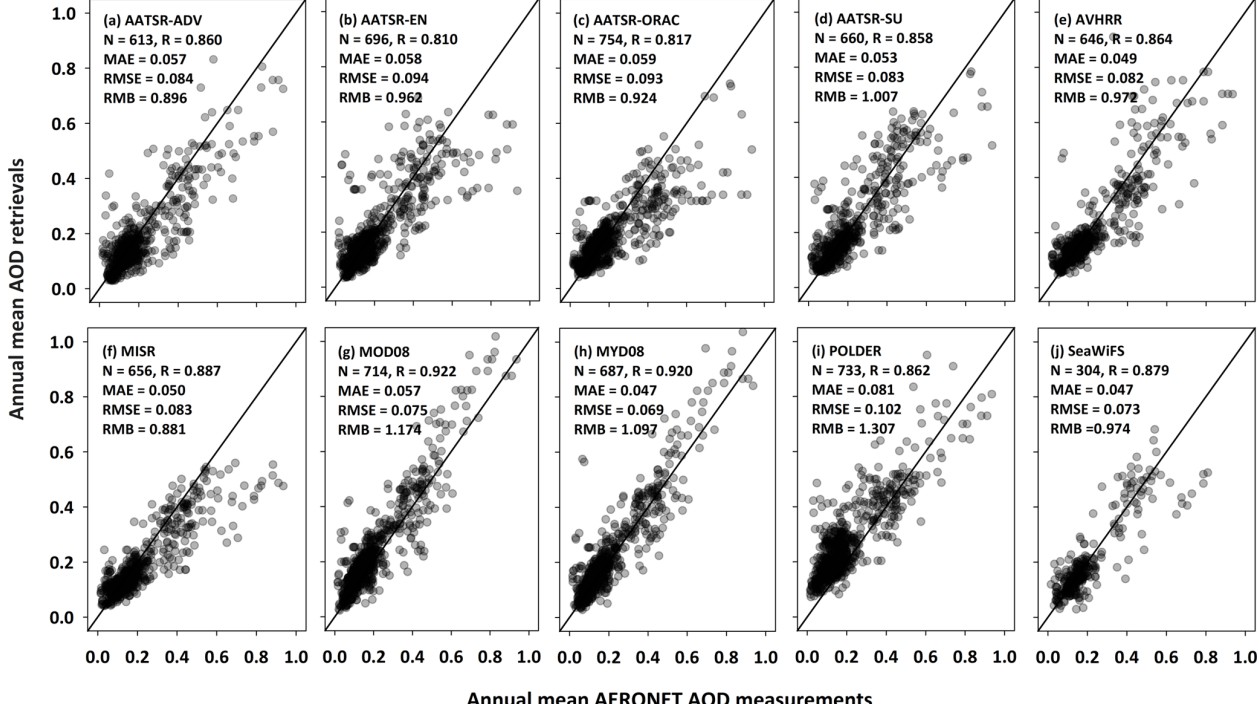

**Figure 13.** Comparisons between the annual global mean satellite-derived $AOD_S$ and AERONET-based $AOD_A$ at 550 nm for all matchup sites throughout the world. The solid black line represents the 1 : 1 line.

with the smallest estimation uncertainties (MAE = 0.047 and RMSE = 0.069) among all the aerosol products.

## 6 AOD temporal variation and trend

### 6.1 Global and regional AOD trend

In this section, we focus on the comparison of the temporal trends of global and regional AOD products. Because the AVHRR and POLDER products provide less than 10 years of aerosol observations in this study, only the remaining eight long-term aerosol products are compared for a common observation period. To ensure that the long-term trend is not impacted by the trends of the aerosol products themselves, we calculated the autocorrelation coefficient of each product with a 1-month lag (Fig. S3).

The linear trends are derived from the de-seasonalized monthly anomaly of each $AOD_S$, and a two-sided test is conducted to present the statistical significance of the temporal trends, where the trends that are significant at the 95 % confidence level ($p < 0.05$) are marked with black dots in Fig. 14. Considering the multiple-hypothesis testing (many data sets and locations are being tested for trends), there could be a significant fraction of false positives. Therefore, the FDR test at the 95 % significance level ($\alpha = 0.05$) is performed to address this issue. We see that the false-positive points can be adequately eliminated after the FDR adjustment and that the

statistically significant areas are more or less reduced (comparing Fig. 14 with Fig. S4). After these processes, the trends are able to represent the time evolution of aerosols realistically.

The global AOD trend distribution shows similar overall spatial patterns among all aerosol products. Over land, significantly positive trends ($a > 0.01$, $p < 0.05$, where $a$ and $p$ are defined in Sect. 3.2) are mainly found in the Middle East and southern Asia, indicating increasing air pollution. In contrast, significantly negative aerosol trends ($a < -0.01$, $p < 0.05$) are mainly observed in eastern North America, Europe, and central Africa, indicating improved air quality. Trends greater than 0.01 yr$^{-1}$ but not statistically significant are found in a few areas of northern Africa and eastern Asia. Strong negative but statistically nonsignificant trends are found in central South America and parts of southeastern Asia. The large trends indicate the importance of aerosol evolution, and the lack of significance may be attributed to the complex aerosol sources; thus, more attention should be placed on these areas to better understand the temporal variations in aerosols. The magnitude of the aerosol trend is generally small ($|a| < 0.005$) over the ocean. However, significantly decreasing aerosol trends ($a > 0.01$, $p < 0.05$) TS2 are observed along the west coast of South America, the east coast of North America and the east coast of Asia. A significant increase in aerosol trends ($a < -0.01$, $p < 0.05$) TS3 was observed along the Indian coast. On the other hand, the four ESA-CCI and MISR aerosol products are not significant

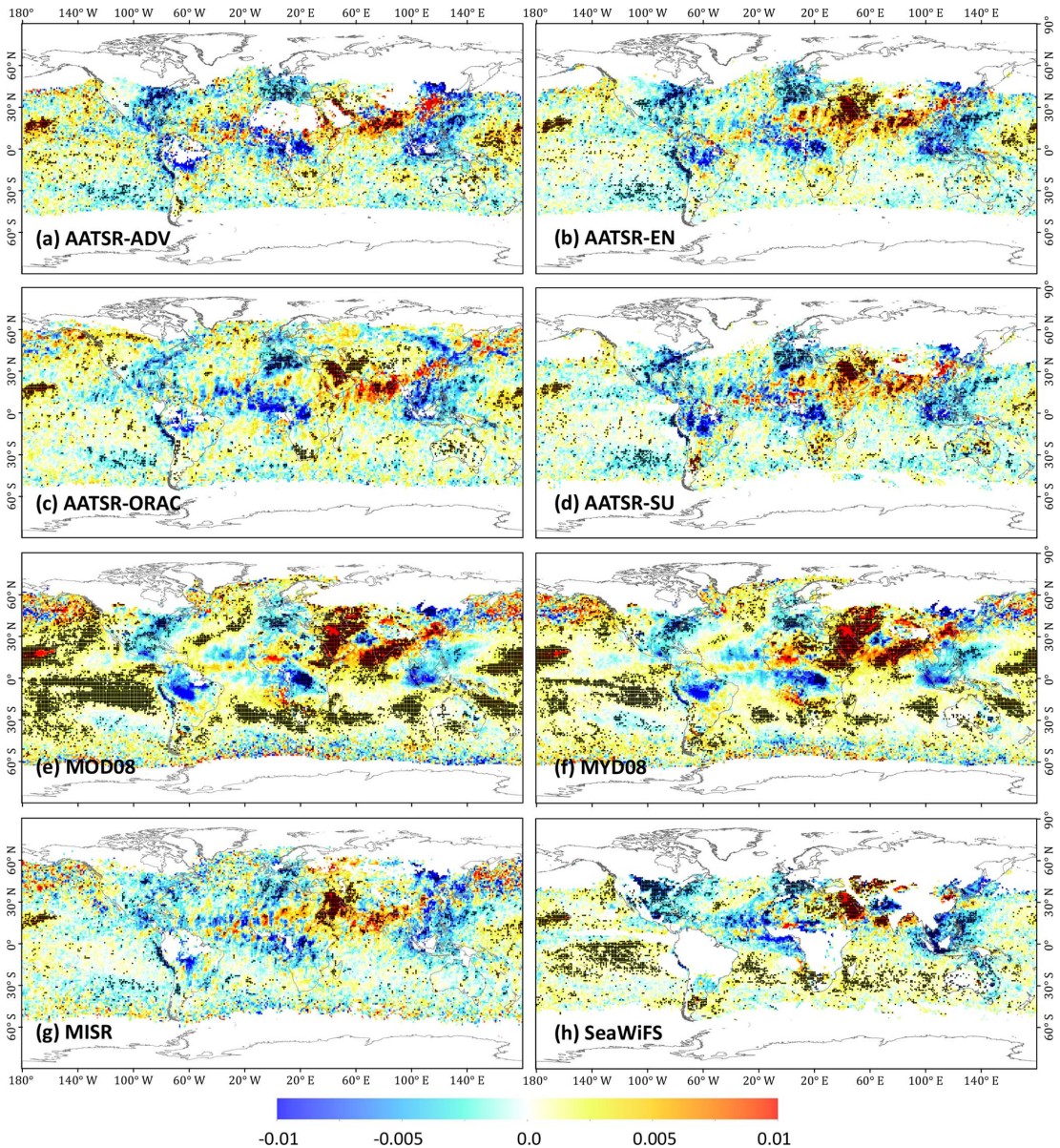

**Figure 14.** Linear trend based on deseasonalized monthly $AOD_S$ anomalies from 2003 to 2010. Units are $AOD\,yr^{-1}$. Black dots indicate a significant trend at the 95 % confidence level ($p < 0.05$).

in most ocean areas, even for the open seas. MODIS and Sea-WiFS products have similar spatial patterns in most ocean areas, such as the significantly increasing trends observed over the Pacific and Indian oceans.

Figure 15 compares the regional aerosol trends among the eight satellite $AOD_S$ values, and Tables 4–5 show the statistics of the regional $AOD_S$ trends and uncertainties over land and ocean. Over land, most small trends are not statistically significant, indicating unconfirmed temporal trends over most land regions (Fig. 15a, Table 4). However, most products show significantly increasing trends over the Middle East ($a = 0.0048$–$0.0111\,yr^{-1}$, $p < 0.05$) and southern

Asia ($a = 0.0034$–$0.0047\,yr^{-1}$, $p < 0.05$), confirming the robust enhancement of aerosols in these two regions. Some products also exhibit obvious decreasing aerosol trends over eastern North America, western North America, Europe, and southeastern Asia. The robustness of the decreasing trends is credible in eastern North America and Europe but unsure in western North America. Over the ocean, the aerosol trends are generally small (Fig. 15b, Table 5), especially for the three open-ocean areas (i.e. Pacific, Indian, and Atlantic oceans). However, the aerosol changes in the four coastal areas exceed $0.002\,yr^{-1}$. The downward trends on the eastern North American coast and European coast and the rising

**Table 4.** Regional temporal trends and uncertainties derived from eight AOD products for the period 2003–2010 over land, where * indicates trends significant at 95 % confidence level.

| Region | Metrics | Aerosol product | | | | | | | |
|--------|---------|-----------|----------|------------|----------|------|-------|-------|---------|
| | | AATSR-ADV | AATSR-EN | AATSR-ORAC | AATSR-SU | MISR | MOD08 | MYD08 | SeaWiFS |
| Land | Trend | −0.0009 | −0.0001 | 0.0002 | −0.0004 | −0.0002 | 0.0006 | 0.0012 | −0.0012 |
| | uncertainty | 0.0007 | 0.0005 | 0.0005 | 0.0006 | 0.0004 | 0.0007 | 0.0007 | 0.0007 |
| ENA | Trend | −0.0031* | −0.0021* | −0.0005 | −0.0031* | −0.0019* | −0.0016 | −0.0016 | −0.0042* |
| | uncertainty | 0.0011 | 0.0009 | 0.0011 | 0.0010 | 0.0007 | 0.0012 | 0.0010 | 0.0008 |
| WNA | Trend | −0.0008 | −0.0004 | 0.0010 | −0.0006 | −0.0005 | 0.0003 | −0.0001 | −0.0029* |
| | uncertainty | 0.0013 | 0.0011 | 0.0012 | 0.0010 | 0.0009 | 0.0019 | 0.0018 | 0.0010 |
| SAM | Trend | −0.0021 | −0.0014 | −0.0010 | −0.0016 | −0.0015 | −0.0019 | −0.0011 | −0.0006 |
| | uncertainty | 0.0031 | 0.0019 | 0.0017 | 0.0027 | 0.0022 | 0.0037 | 0.0034 | 0.0017 |
| EUR | Trend | −0.0021* | −0.0018* | −0.0007 | −0.0024* | −0.0009 | 0.0000 | −0.0004 | −0.0031* |
| | uncertainty | 0.0009 | 0.0009 | 0.0010 | 0.0010 | 0.0007 | 0.0010 | 0.0009 | 0.0011 |
| AFR | Trend | −0.0005 | 0.0005 | −0.0007 | 0.0000 | 0.0000 | 0.0001 | 0.0017 | −0.0018 |
| | uncertainty | 0.0012 | 0.0012 | 0.0009 | 0.0011 | 0.0010 | 0.0012 | 0.0013 | 0.0016 |
| ME | Trend | 0.0048* | 0.0083* | 0.0050* | 0.0073* | 0.0077* | 0.0084* | 0.0111* | 0.0079* |
| | uncertainty | 0.0020 | 0.0024 | 0.0013 | 0.0022 | 0.0025 | 0.0036 | 0.0035 | 0.0025 |
| EAA | Trend | −0.0011 | −0.0004 | −0.0001 | −0.0008 | −0.0019 | 0.0003 | 0.0009 | −0.0019 |
| | uncertainty | 0.0037 | 0.0022 | 0.0021 | 0.0026 | 0.0028 | 0.0039 | 0.0038 | 0.0023 |
| SAA | Trend | 0.0040* | 0.0034 | 0.0047* | 0.0044* | 0.0018 | 0.0037 | 0.0046 | −0.0044 |
| | uncertainty | 0.0024 | 0.0019 | 0.0014 | 0.0020 | 0.0017 | 0.0027 | 0.0028 | 0.0023 |
| SEA | Trend | −0.0059 | −0.0041 | −0.0041 | −0.0050 | −0.0020 | −0.0034 | −0.0020 | −0.0041* |
| | uncertainty | 0.0052 | 0.0030 | 0.0025 | 0.0037 | 0.0027 | 0.0054 | 0.0047 | 0.0019 |
| OCE | Trend | 0.0002 | 0.0001 | 0.0007 | 0.0000 | −0.0005 | 0.0004 | 0.0006 | −0.0004 |
| | uncertainty | 0.0004 | 0.0007 | 0.0006 | 0.0012 | 0.0007 | 0.0006 | 0.0006 | 0.0003 |

**Table 5.** Same as Table 4 but for ocean.

| Region | Metrics | Aerosol product | | | | | | | |
|--------|---------|-----------|----------|------------|----------|------|-------|-------|---------|
| | | AATSR-ADV | AATSR-EN | AATSR-ORAC | AATSR-SU | MISR | MOD08 | MYD08 | SeaWiFS |
| Ocean | Trend | −0.0003 | −0.0004 | 0.0000 | −0.0004 | −0.0004 | 0.0009* | 0.0006 | −0.0006 |
| | uncertainty | 0.0004 | 0.0003 | 0.0005 | 0.0003 | 0.0003 | 0.0004 | 0.0004 | 0.0004 |
| PAO | Trend | 0.0002 | −0.0001 | 0.0001 | −0.0001 | −0.0002 | 0.0015* | 0.0008* | 0.0010* |
| | uncertainty | 0.0004 | 0.0002 | 0.0005 | 0.0003 | 0.0003 | 0.0004 | 0.0004 | 0.0004 |
| NAO | Trend | −0.0006 | 0.0001 | 0.0010 | 0.0005 | −0.0002 | 0.0021* | 0.0019* | −0.0006 |
| | uncertainty | 0.0009 | 0.0007 | 0.0009 | 0.0006 | 0.0006 | 0.0010 | 0.0009 | 0.0006 |
| SAO | Trend | 0.0001 | 0.0000 | 0.0001 | −0.0002 | −0.0004 | 0.0014 | 0.0011 | 0.0003 |
| | uncertainty | 0.0007 | 0.0005 | 0.0007 | 0.0004 | 0.0004 | 0.0008 | 0.0007 | 0.0004 |
| INO | Trend | 0.0000 | −0.0001 | 0.0003 | −0.0001 | −0.0001 | 0.0012* | 0.0008 | 0.0007 |
| | uncertainty | 0.0004 | 0.0003 | 0.0005 | 0.0004 | 0.0004 | 0.0005 | 0.0005 | 0.0005 |
| ENC | Trend | −0.0037* | −0.0032* | −0.0022* | −0.0021* | −0.0023* | −0.0020* | −0.0024* | −0.0026* |
| | uncertainty | 0.0010 | 0.0008 | 0.0007 | 0.0007 | 0.0008 | 0.0008 | 0.0008 | 0.0007 |
| EUC | Trend | −0.0026* | −0.0021* | −0.0017 | −0.0021* | −0.0018* | −0.0008 | −0.0011 | −0.0025* |
| | uncertainty | 0.0011 | 0.0009 | 0.0011 | 0.0008 | 0.0008 | 0.0010 | 0.0009 | 0.0010 |
| SAC | Trend | 0.0041* | 0.0030 | 0.0055* | 0.0019 | 0.0030* | 0.0064* | 0.0049* | −0.0002 |
| | uncertainty | 0.0019 | 0.0016 | 0.0026 | 0.0014 | 0.0016 | 0.0025 | 0.0023 | 0.0020 |
| EAC | Trend | −0.0030 | −0.0025 | −0.0011 | −0.0010 | −0.0029 | 0.0008 | −0.0001 | −0.0026 |
| | uncertainty | 0.0024 | 0.0016 | 0.0021 | 0.0012 | 0.0022 | 0.0023 | 0.0022 | 0.0021 |

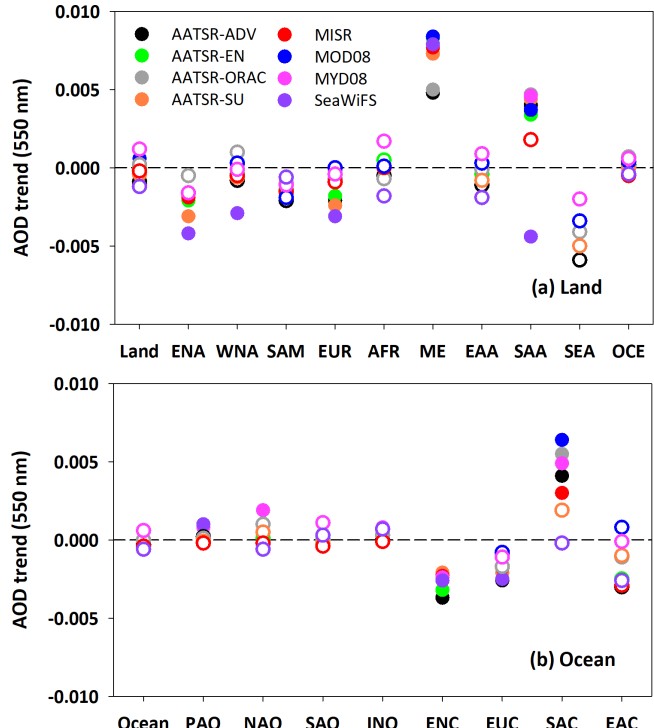

**Figure 15.** Regional linear trends based on de-seasonalized monthly $AOD_S$ anomalies over land and ocean from 2003 to 2010, where the hollow and solid circles represent statistically nonsignificant and significant trends at the 95 % confidence level ($p < 0.05$), respectively.

trend on the southern Asian coast are robust. The temporal trend over the eastern Asian coast is unassured.

### 6.2 Comparison between $AOD_S$ and $AOD_A$ trends

In this section, the satellite-derived $AOD_S$ trends are compared against the AERONET $AOD_A$ trends from ground measurements. To ensure the statistical significance of the trend calculations, only the AERONET sites with at least 5 years (120 months) of effective observations are selected. Figure 16 plots the $AOD_S$ and $AOD_A$ trends at all available sites for the eight satellite products from 2003 to 2010 over the world. The results show that most products can capture the correct AOD trends at 40 % to 45 % of the available sites over land and ocean. For four ESA-CCI aerosol products, the satellite-derived AOD trends are consistent with AERONET-based trends with average MAEs ranging from 0.45 to 0.49 and RMSEs ranging from 0.65 to 0.78. The MISR product shows an overall better performance with a lower MAE of 0.418 and RMSE of 0.589 than the ESA-CCI products. However, the SeaWiFS product has valid comparisons at only 59 sites due to the lower spatial coverage over land, and the $AOD_S$ trend exhibits the worst performance with the largest MAE and RMSE values among all the aerosol products. By contrast, [CE3] the MODIS products capture the tem-

poral $AOD_S$ trend most accurately with the lowest MAE and RMSE errors. Terra and Aqua show similar performances with almost equal CTPs of 42 %, and the MODIS products capture the temporal $AOD_S$ trend most accurately with the lowest MAE and RMSE errors.

### 7 Summary and conclusion

This study focuses on the similarities and differences in the spatial variations and temporal trends of the current satellite-derived AOD products. For this purpose, 11 global monthly aerosol products at coarse spatial resolutions are collected and compared against the ground measurements from 308 AERONET sites throughout the world, including four products from the European Space Agency's Climate Change Initiative (AATSR-ADV, AATSR-EN, AATSR-ORAC, and AATSR-SU) and AVHRR, MISR, Terra and Aqua MODIS, POLDER, SeaWiFS, and VIIRS products. These data are evaluated in three ways: (1) direct comparison of monthly retrievals against the AERONET observations on global, continent, and site scales; (2) comparison of the global and regional AOD spatial coverage and distribution; and (3) comparison of the global and regional AOD temporal variations and trends. Our results may help readers to better understand the features of different satellite aerosol products and select a suitable aerosol dataset for their studies.

In terms of the performance of multiple products on different spatial scales, we show that the four ESA-CCI aerosol products show similar performance and are generally worse than the AVHRR and MISR products. The SeaWiFS product provides the smallest sample size despite an overall good performance. The seven abovementioned products underestimate the aerosol loads, especially the MISR and AATSR-ADV products. The POLDER product performs worst with the largest estimation uncertainties and significantly overestimates the aerosol loads. The MODIS products (especially Aqua MODIS) show superior performance among all products with small estimation uncertainties in most regions and sites but overestimate AOD overall. In general, most products exhibit consistently good performance over dark surfaces in Europe and North America but perform worse over bright and complex surfaces in southern Asia, eastern Asia, Africa, and the Middle East.

In terms of the aerosol spatial distribution, the AATSR-ADV and SeaWiFS products have lower spatial coverage ($\sim 68$ % and 69 %) with numerous missing values, while the MODIS products can provide the highest spatial coverage ($\sim 87$ %) throughout the world. Most products show the highest spatial coverage in SON but the lowest aerosol concentrations in DJF. In general, the seasonal aerosol spatial distributions over the ocean are more consistent among the different aerosol products. However, noticeable spatial heterogeneity and numerical differences are observed over land, especially over Africa, Asia, and some coastal areas, which

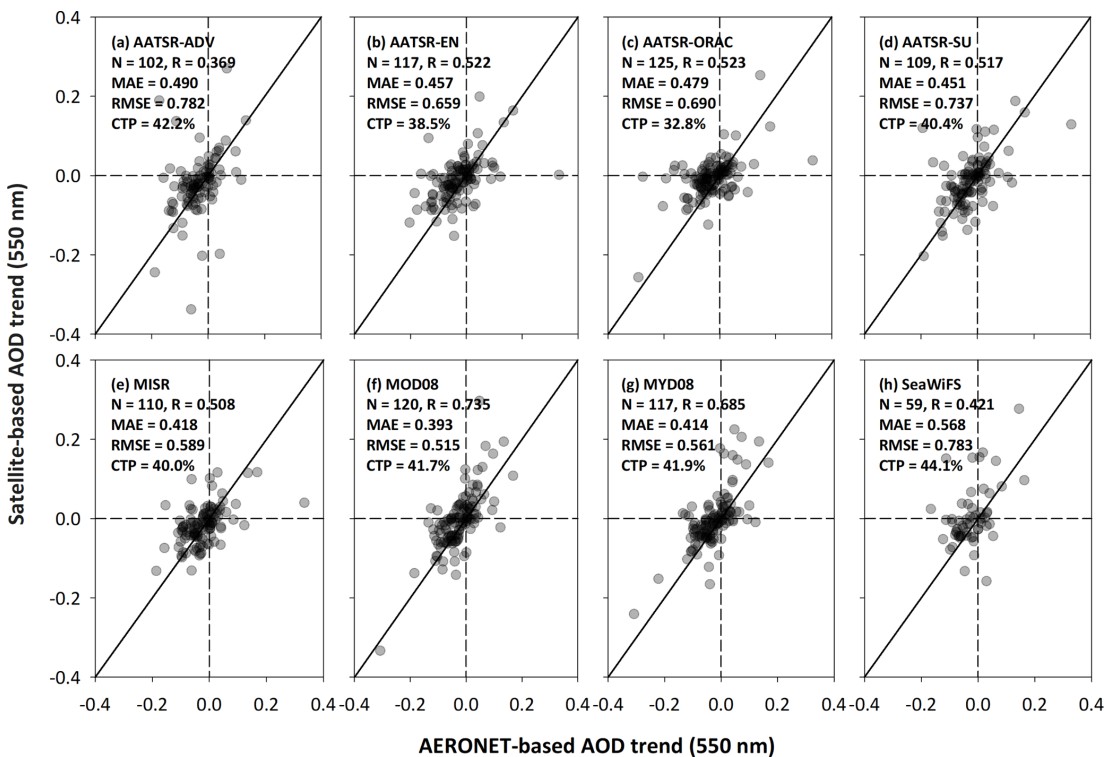

**Figure 16.** Comparisons between the linear trends based on the de-seasonalized monthly AOD$_S$ anomalies from 2003 to 2010. Units are AOD per decade. The solid black line represents the 1 : 1 line.

are possibly due to the complex aerosol sources and the limitations of the different aerosol retrieval algorithms. In general, the best performance for describing the seasonal aerosol distributions is always observed in SON, but the worst is observed in JJA. The Aqua MODIS product performs best with almost all the best evaluation metrics (e.g. MAE and RMSE) among all the products at the seasonal and annual levels.

In terms of the temporal aerosol trends, most products exhibit similar spatial patterns throughout the world, where significantly positive trends are found over the Middle East, southern Asia, and southern Asian coasts. In contrast, significantly decreasing trends are observed over eastern North America, Europe, and their coastal areas. In general, most products can capture the correct AOD trends at more than approximately 40 % of the AERONET sites. In general, the MODIS products show the best performance with the best evaluation metrics in describing the temporal aerosol variations.

This study has comprehensively evaluated the performance of the atmosphere level 3 aerosol products derived from multi-source satellite sensors in describing temporal and spatial aerosol variations and provided users with preliminary data selection and suggestions for their particularly special studies. Due to large differences in the performance (especially for local regions) and operation time for different aerosol products, a better selection of more accurate aerosol products may lead to more reliable research conclusions. Meanwhile, by making full use of multi-source aerosol products, newly combined or merged approaches can be further explored to reduce the estimate bias for reproducing more accurate global aerosol products. This might be specifically critical for validating the aerosol simulation and prediction using global climate models. Furthermore, aerosol retrieval over highly bright (e.g. desert, bare land) and heterogeneous (e.g. urban) areas over land still have large estimation uncertainties, which brings great challenges due to high surface brightness and intense human activities. Therefore, the aerosol algorithm teams may need put more effort into optimizing the estimation in surface reflectance and the assumption of aerosol types over these areas to improve the data quality of aerosol retrievals and thus increase the spatial coverage and decrease the diversity among different data sets. These could be the major points of aerosol research in the future.

*Data availability.* Data are available by contacting the author.

*Supplement.* The supplement related to this article is available online at: https://doi.org/10.5194/acp-19-1-2019-supplement.

*Author contributions.* YP designed the research, and JW carried out the research and wrote the initial draft of this manuscript. RM, LS and JG helped review the manuscript. All authors made substantial contributions to this work.

*Competing interests.* The authors declare that they have no conflict of interest.

*Acknowledgements.* The ESA-CCI AATSR monthly products are obtained from the ICARE Data and Services Centre (http://www.icare.univ-lille1.fr/cci, last access: 1 October 2018). The MODIS, MISR, AVHRR and SeaWiFS monthly products are available at https://search.earthdata.nasa.gov/ (last access: 1 October 2018). The POLDER product is available at https://www.grasp-open.com/products/polder-data-release/ (last access: 1 October 2018), and the AERONET measurements are available from the NASA Goddard Space Flight Center (https://aeronet.gsfc.nasa.gov/, last access: 1 October 2018). We appreciate the four anonymous reviewers for their constructive suggestions that largely improved the paper.

*Financial support.* This research has been supported by the National Natural Science Foundation of China (grant nos. 71690243, 41775137 and 41761144056) and the National Important Project of the Ministry of Science and Technology (grant no. 2017YFC1501404).

*Review statement.* This paper was edited by Stelios Kazadzis and reviewed by four anonymous referees.

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

**Remarks from the language copy-editor**

CE1    It is our standard to write numbers 10 and above in numerals.

CE2    It does not make sense for "latitude" to be plural because only one latitude is named here.

CE3    Please note that the proofreading stage is for catching technical errors and substantial changes to content cannot be made because the review process has already taken place.

**Remarks from the typesetter**

TS1    This change needs to be approved by the editor.

TS2    This change needs to be approved by the editor.

TS3    This change needs to be approved by the editor.