# Peer review of "Inter-comparison in spatial distributions and temporal trends derived from multi-source satellite aerosol products"

_Atmospheric Chemistry and Physics, 2018_

## Referee Comment (RC1) · Anonymous Referee #3 · 17 Dec 2018

The authors provide a comparison of nine satellite-derived global AOD data sets, with ground-based AERONET (land) and MAN (ocean) AOD data as reference. They apply different statistical metrics and look at the data sets on different spatial scales: global, regional and per reference site. They also look at trends. Differences and agreements between data sets are described. The manuscript provides an interesting overview of AOD data sets available in the public domain, although some recent data sets like those from VIIRS are missing. Also I wonder why for AVHRR only the over-ocean AOD is included and the recent over-land data sets described by Sayer and Hsu in JGR, 2017, were not included. It would be interesting to see how these data sets, retrieved from a sensor not designed for aerosol retrieval, compares to those from dedicated sensors

like MISR and MODIS. Likewise, a comparison with PARASOL (POLDER) would have been interesting. As regards the title, I would recommend to change "inconsistency" to "Intercomparison", because not all and not always are the data sets inconsistent, they are often also consistent.

Specific comments (line numbers refer to the pdf published online) 46: suggest "composition and short life time of atmospheric aerosol particles" 57: remove "observable" 79: remove "seemingly" 80: This sentence suggests that some studies have indeed focused on exploring . . . ; hence references to these studies are needed here 85: suggest "evaluation and comparison" 91: validation 103: ADV was first published by Veefkind et al., 1998a, for retrieval over land; Over Ocean ASV was first developed by Veefkind et al., 1998b 112: A more recent reference for the Swansea algorithm is Bevan et al., 2012 117: Holzer-Popp 122: "AVHRR aerosol product is only available": this is NOT true, see my general comment and references to Sayer and Hsu Sect. 2.2: not only AERONET is used, AOD over ocean is provided by the Marine Aerosol Network (Smirnov et al., 2009) 196: could you reword the text to make more clear how the lsq fit is applied 199: trend symbols: same direction of the trend Para starting at 230: An important indicator is also the EE, and the above and under EE which clearly indicate overestimation (e.g. for MODIS) and underestimation (e.g. for MISR). Here and in the next paragraphs, I do not understand how MISR can have a similar number of collocations as MODIS in spite of it's much smaller swath; MISR should have an N similar to AATSR 235: I am not sure that your judgement of ADV is completely fair, since indeed MAE and RMSE are worse, but not EE; looking at the statistics in Table 2, it seems that none of the sensors has the best statistics for all numbers, so it is hard to make such statements 236: smaller number of retrieval collected: I think this should be a smaller number of collocation pairs since less references data are available; again, how can MISR provide a similar number of data collections as MODIS? 244: SeaWiFS is not improved, but it's performance is better 251: what is the statistical parameter indicating estimation uncertainty and accuracy? 257 and 263 and 275-277: a high R does not imply that the performance is better: MODIS has high R, but figure 2 shows that

MODIS overestimates, so actually it's performance in estimating AOD is not so good. This should be re-worded in the text 266: RSA, typo and you mean ESA? Sect. 5.2: there are very large differences in the mean AOD values; yet they all compare well with AERONET (Fig. 2 and 3): why are these differences not visible in the scatterplots? 304: suggest to plot the eight-year mean value in the figures Sect. 5.3 title not clear: suggest to change the Section title to " Comparison of satellite- and AERONET- derived annual mean AOD at each site 340: this sentence is not accurate: you compare annual mean AOD for each satellite over an AERONET sites with the AERONET annual mean value 375-376: I do not understand the sentence "Four . . . areas." Why are the first 4 similar and the other 2 consistent? What do you mean with that? MYD08 and SeaW-iFs show quite some differences. Could you re-word so it is more clear? 379: what do you mean with "treatment in neighbouring pixels": did you describe that in the text? Sect. 6.4: Linear trends were fitted, so it may be that upward and downward trends are compensated over this long period of 18 years and thus the trends in Fig 14 are not representative. Could you please add a comment in the text? Figure Captions: 2 and 3: Density scatterplot of the monthly averages of satellite-derived AOD (operational products) versus AERONET AOD 8: replace aerosols with AOD 10: ".. with annual mean AERONET AOD data for all sites . . ." 11: trends of AOD at 550 nm 12: replace "aerosol trends" with "trends of AOD at 550 nm" 13: I think you show trends of AOD at 550 nm, not annual mean aerosols? 15: remove "variations"

References: Bevan, S. L., North, P. R. J., Los, S. O., & Grey, W. M. F. (2012). A global dataset of atmospheric aerosol optical depth and surface reflectance from AATSR. Remote Sensing of Environment, 116, 119–210. Hsu, N. C., Lee, J., Sayer, A. M., Carletta, N., Chen, S. H., Tucker, C. J., . . . Tsay, S. C. : Retrieving near-global aerosol loading over land and ocean from AVHRR. Journal of Geophysical Research: Atmospheres, 122. doi:10.1002/2017JD026932, 2017 Sayer, A. M., N. C. Hsu, J. Lee, N. Carletta, S.‐H. Chen, and A. Smirnov: Evaluation of NASA Deep Blue/SOAR aerosol retrieval algorithms applied to AVHRR measurements, J. Geophys. Res. Atmos., 122, doi:10.1002/2017JD026934, 2017. Smirnov, A., et al. (2009), Maritime

Aerosol Network as a component of Aerosol Robotic Network, J. Geophys. Res., 114, D06204, doi:10.1029/2008JD011257. Veefkind, J.P., G. de Leeuw and P.A. Durkee (1998a). Retrieval of aerosol optical depth over land using two-angle view satellite radiometry during TARFOX. Geophys. Res. Letters. 25(16), 3135-3138. Veefkind, J.P. and de Leeuw, G., 1998b, A new algorithm to determine the spectral aerosol optical depth from satellite radiometer measurements. Journal of Aerosol Sciences, 29, 1237-1248.

---

## Referee Comment (RC2) · Anonymous Referee #4 · 21 Dec 2018

**Review report**

The objective of the present study is the intercomparison of various spaceborne retrievals which are widely utilized in aerosol studies. The analysis has been performed at different spatial scales and for a long-term period thus increasing the robustness of the obtained findings. Nevertheless, the major weakness is that the interpretation of the results is poor without providing insight and sufficient answers about the potential reasons which can explain the apparent differences. More specifically, throughout the manuscript the authors are restricted just to a description of the figures which can be easily done by a reader without reading the text. Therefore, I strongly believe that the manuscript needs a major revision before it can be acceptable for publication in ACP. Below are listed my comments/questions which I hope will help the authors to improve their work.

1. Which version of the AERONET data is utilized?
2. You have to provide a better description of the satellite datasets (version, spatial resolution, temporal resolution, temporal availability, where these data are stored, literature etc.).
3. **Page 6 – Lines 177-179:** This sentence is confusing for me. Are you using monthly means or daily retrievals which are used in order to calculate the monthly averages? What do you mean "*…with sufficiently high-quality…*"? Are you applying any quality assurance flag or are you using the raw data as is?
4. **Page 6 – Lines 180-181:** Please rephrase this sentence.
5. **Figure 2:** I cannot understand why the comparison versus AERONET is made for the periods where each dataset is available and not for the common period (Table 2 and 3). In the scatterplots, the EE dashed lines are common for all satellite data. This is not correct since each satellite sensor has different uncertainty limits (which are not stated in the text).
6. **Page 8 – Lines 228-244:** Is there any interpretation for these results? The authors must consider previous evaluation analyses in their discussion.
7. **Section 4.2:** You have to repeat the analysis for EE using the corresponding limits for each satellite sensor. Moreover, you have to compare your results with other existing works.
8. **Section 5.1:** There are several points which must be discussed in Figure 7. For example, the differences among AATSR-ORAC, AATSR-SU, MODIS and SeaWIFS recorded across N. Africa. Likewise, in E. Asia, it seems that there is a strong diversity, in terms of AOD values, among the datasets. In AATSR-ORAC, there is an abrupt change of AODs between maritime and continental areas in the eastern tropical Atlantic Ocean as well as in the Arabian Sea. Finally, it would be useful to reproduce the maps by considering common points in all datasets

separately over land (exclude AATSR in order to have available observations over Sahara and in the Middle East) and sea.

9. **Figure 9**: For the computation of the regional means based on the satellite observations are used all the grid cells of the domain of interest or only the pixels in which AERONET stations reside? Why there is an increasing trend for MODIS data in EAA as well as in EUR? On the contrary, in SAA the agreement between MODIS and AERONET improves gradually. Why this is happening?

10. **Section 6.4:** Are your results in agreement with other similar studies? In the global map, there are clear signals over wide areas of the planet which are not discussed appropriately in the text. Which factors regulate (meteorology, emissions, teleconnections, land use, etc.) the obtained pattern?

11. **Figure 1:** First of all, there are mistakes on the region names. Please correct the European Coast as well as the South Africa (it is not in Asia!). Which is the domain for the European Coast? Replace Atlantic Ocean with South Atlantic Ocean.

12. **Figure 11:** Replace 2017 with 2010.

13. **Page 3 – Lines 64-77:** In this part of the manuscript the authors are stating only studies representative for China. Satellite observations have been also used for other regions of the planet such as the Mediterranean, Europe, Atlantic Ocean etc.

---

## Referee Comment (RC3) · Anonymous Referee #2 · 24 Dec 2018

The authors take 9 satellite aerosol optical depth (AOD) monthly mean data sets, and perform comparisons against each other and AERONET monthly mean data. These come from a variety of satellite instruments and algorithms. They look at similarities in spatial and temporal patterns. This research area is important because understanding aerosol influences on the Earth requires understanding the strengths and limitations of each data set.

This is a pretty big task and it is good to see it being tackled, because as the authors note there has not been a great deal of attention to data set choice in some satellite analyses. However, I think this version of the paper has problems. The statistics and

analysis are very superficial, and the metrics used do not always make sense or are incorrect. For example, autocorrelation and false discovery rate are ignored, a level 2 error metric is used for level 3 analysis. The terminology has errors in some sections (e.g. "validation" when this is not a validation analysis). And in several places the authors omit relevant references and use out of date ones, or instead insert excessive self-citations. There is also a possible wavelength issue with the AVHRR product used. I recommend major revisions and would like to review the revised version. This paper felt to me like the authors just downloaded a bunch of data and ran a bunch of statistical metrics against it, without thinking about what was being done or why. I suggest that when revising, they focus on what science question they are trying to answer, and then figure out the right tools to answer it, and provide a detailed discussion. Otherwise this feels not like a scientific research paper but rather the output of some automated data processing software.

After writing this review, I read the other two comments currently posted on ACPD for this paper. I generally agree with the other reviewers' comments.

My comments in support of my recommendation are as follows:

Line 20, and elsewhere: Operational is not the right word here. It implies something produces as part of routine agency operations while a mission is ongoing. Most of the products do not fit that definition; in fact I think only MODIS and AVHRR do as they are produced with a few hours latency to support assimilation applications. I suggest deleting this word throughout.

Title, line 30, line 35 and elsewhere: Terms like "significant inconsistencies" (or just "inconsistencies" alone), "seriously" are used a lot in this paper. But most of the time they are used as "weasel words", i.e. in a non-specific way which can lead people to get a certain impression which is not necessarily warranted. For example, "inconsistencies". Taken to an extreme, any two data sets will not be identical so are to some extent "inconsistent". The relevant question is, for any particular application, is the level of consistency between them sufficient? For example, if one wants to look at seasonal variations, AOD magnitude might not be as important as the pattern throughout the year. But if one wants to look at radiative effects, magnitude is more important. If one wants to see large-scale features, then a broader swath to improve sampling at the expense of some accuracy might be desirable. The point is that these are all different instruments with different characteristics. We expect them to not be identical. The wording in this paper (these examples and elsewhere) seems designed to send a message that aerosol remote sensing has big problems. In my opinion, that's an overly pessimistic assessment. There are differences but in general the reasons for those are understood. So which data set is best to use for a given study depends on the type of science question you are trying to answer. There is no "best" data set. This recent paper by Sayer et al in JGR (https://agupubs.onlinelibrary.wiley.com/doi/10.1029/2018JD029465) covers some similar ground to the current study, in that part of it compares time series and maps of various over-water satellite AOD data sets. That paper goes into a lot of discussion about the differences between them and why they might be. So although there is a lot of diversity in the over-water AOD, the reasons are generally known. My personal opinion is that over much of the world, differences are probably more due to sampling differences (swath and pixel selection) than algorithm. I suggest the authors refer to that paper in their revised manuscript, and go from describing things as "inconsistent" to try to for example make recommendations as to which data sets might be better or worse suited for different applications. Recommendations like that, with evidence, are more useful than just declaring "inconsistency". Perhaps "comparisons" or "consistency assessment" is a better way to describe the analysis in title and text.

Line 50: I think this should say 20th century, not 19th. I am not aware of any observation networks before the late 20th century. If there are, please provide references. Aerosol science didn't really start until John Aitken in the late 1800s.

Line 118: This should be "Holzer-Popp" not "Holzerpopp". The author's name is double-barrelled.

Line 121: This should be changed to indicate it is the NOAA AVHRR aerosol product. There is also a NASA GISS aerosol product (GACP), which is monthly-only and ocean-only, and a NASA Deep Blue aerosol product, which also covers land but is presently only available for limited time periods (I know 2006-2011 is available). It would be good to clarify what is used and why here. Deep Blue and GISS also provide 550 nm while NOAA AVHRR does not. Perhaps one of those could be added.

Line 125: Authors should state more clearly here that they are using the 0.63 micron AOD (aot1 SDS), as it is important to note that this is different from the 550 nm AOD provided by most other data sets, and would result in offsets dependent on aerosol type. The authors do not seem to mention this later in the paper (e.g. line 179 says the satellites are at 550 nm). Was the AVHRR AOD somehow extrapolated to 550 nm like the others? Or was it left at 630 nm and the wavelength dependence neglected?

Line 139: Authors are missing references for the version 23 algorithm they are using here. Martonchik/Kalashnikova are out of date. The water approach is discussed by Witek et al (2018): https://www.atmos-meas-tech.net/11/429/2018/ The land approach is discussed by Garay et al (2017): https://www.atmos-chem-phys.net/17/5095/2017/ I suggest authors read and cite these papers, since it appears they have been referring to older documents.

Line 155: Sayer et al (2014): https://agupubs.onlinelibrary.wiley.com/doi/full/10.1002/2014JD022453 is a more complete reference for the DTB products than Levy et al (2013). It also provides a comparison for DB, DT, and DTB. It will also be useful for the authors' analysis since it provides similar discussion about the level of consistency between the data sets. All the papers cited here are about Collection 6 but I know the MODIS teams and they did not publish papers about Collection 6.1 yet (still in review).

Line 163: I am not sure that the FM acronym for "forward model" is needed here. I don't think it is used later.

Line 166: Somewhere in this section I would add a note to state that this is not a

validation but a comparison, because the authors are using monthly data and not instantaneous data. So there are sampling differences contribution as well as retrieval quality. The authors are not performing a true validation exercise here.

Line 189: The authors insert four self-citations for a one-line equation developed by other people something like 75 years ago. This seems a little excessive. Please remove these citations or replace with ones to the original work by Angstrom.

Line 190: I recommend the authors account for lag 1 month autocorrelation in the time series. This is commonly done in AOD trend analyses as the data can be significantly autocorrelated on these scales (because large-scale systems and seasonal patterns can persist for weeks to months). This will keep the same trend values but affects the estimated uncertainties on the trend. See Weatherhead et al (1998): https://agupubs.onlinelibrary.wiley.com/doi/abs/10.1029/98JD00995 for examples how to calculate this.

Lines 197-200: I am not sure that "correct trend percentage" makes sense. If a trend is close to 0, you will end up with a lot of apparently "wrong" trends if the sign is wrong, even if the conclusion that there is almost no trend is correct. For example, if you had trends of +0.01 from AERONET and +0.1 from satellite the authors would say this is "correct" even though the difference is huge. But if you had +0.005 from AERONET and -0.005 from satellite the authors would classify it as "incorrect", even though they are both small and probably statistically indistinguishable within trend uncertainties. A further problem is that this makes the implicit assumption that AERONET trends are perfect when of course they also have some measurement uncertainty and sampling uncertainty. I suggest that a better metric would be to report the "consistent trend percentage". This could be calculated by checking whether the satellite and AERONET trends are consistent within each uncertainty or not. This is a more fair and statistically appropriate test. The authors could also report those situations in which the AERONET estimate is too uncertain to be useful. I doubt that five years is enough to estimate a trend robustly in many cases, due to significant annual variability. So quite possibly the

uncertainty on the AERONET estimates even is quite high.

I also wonder if seasonal trends would be better than annual, because we know that aerosol patterns show strong seasonal features (so trends in seasonal behaviour could be masked in an annual trend analysis). The authors need to justify this more strongly.

Line 209: The subscripts are very long. I suggest replacing AOD_RETRIEVAL with AOD_R (for "retrieval") or AOD_S (for "satellite"), and AOD_AERONET with AOD_A. This will make it more readable.

Line 210: Correlation is not useful when the data range is small compared to the uncertainty on the data. You could have a great data set but still have a small correlation. For example, over the open ocean AOD does not change much, so a low correlation is scientifically not much of a problem for most scientific applications, as long as bias and RMSE are low. The authors should note this because a lot of the maps and discussion rely on correlation.

Line 211: This EE is an expected envelope for level 2 error over land only, not for level 3 and not for water. It is not meaningful for level 3 data, and it is misleading to apply it that way. There is at present no error estimate for satellite level 3 products. I suggest the authors remove this quantity because it is misleading. In my view the other statistics are enough. This also requires removing from the discussion later on. Either remove it or create and justify some metric for what is an acceptable EE on the monthly data. My feeling is that a monthly level 3 EE should be smaller than the level 2 one, because some error sources should cancel out.

Line 219, 230, 468, Table 2, Figure 10, and elsewhere: No, this is not a validation, it is a comparison, because you are using monthly mean products and not level 2. Validation requires a ground truth. There is no ground truth for monthly data because there is no instrument sampling continuous monthly data. AERONET is only a validation for level 2 data. The authors should change the wording because it is misleading, and word choice matters. The analysis the authors are doing here is fundamentally different from

the dozens of published level 2 validation papers, and it is important not to muddle the issue.

Line 295: Again, it is not ideal to provide a single self-citation here when these issues have been documented by many algorithm teams for many years.

Line 362: Throughout section 6 the authors talk a lot about trend significance. However something which has been overlooked is that since there is multiple hypothesis testing going on (many data sets and locations are being tested for trends), there could be a significant fraction of false positives. See e.g. Wilks (2006) for more on this: https://journals.ametsoc.org/doi/10.1175/JAM2404.1 So, the authors should make some quantification about the expected false discovery rate. Further, statistical significance is only one factor. Figures 11 and 14 are a prime example of this problem. Scientific significance is another. If you get a trend of 0.001 with an uncertainty of 0.0001, that is statistically significant but scientifically not important because it is so small. But if you get a trend of 0.1 with an uncertainty of 0.1, that is not statistically significant by traditional tests, but is potentially very important, because 0.1 is a large potential trend. The authors here seem to focus on statistical significance and sign rather than actually looking at the numbers. This is quite superficial. I would like to see the whole section reconsidered.

Line 496: "Goddard", not "Godard".

Figure 7 (and associated discussion): I do not like annual mean maps in general because AOD patterns and sampling are strongly dependent on season. So in some areas there will be a difference just because data are coming from different months. And in some areas things could look to be in closer agreement than they really are, if biases in different seasons are opposite and can cancel out. Annual mean AOD is also not meaningful for most applications. I would prefer to see this figure and discussion instead as a composite of four sets of seasonal plots. This would be a closer to apples to apples comparison, and also allow an examination of seasonal variability.

Figure 8, 12: Could this be redrawn to show coloured symbols instead of bars? In some cases the bars are overlapping and so it is hard to tell which is which. It can also give misleading impressions. For example, in land ENAM the black and pink are overlapped. I guess black was drawn first and pink second, so pink is on top. So the impression is that black is lower than pink, because we can only see the bottom of black. But in reality, because so much of black is hidden, it probably means that black and pink are very similar. Coloured symbols instead of bars would be clearer and easier to tell.

Figure 9: since this is not a validation but a comparison, it would be better to say "offset" rather than "bias" here. Bias implies an offset with reference to a truth, and we have no truth. Word choice is important.

---

## Author Comment (AC2) · 29 Mar 2019

Reviewer: 4

The objective of the present study is the intercomparison of various spaceborne retrievals which are widely utilized in aerosol studies. The analysis has been performed at different spatial scales and for a long-term period thus increasing the robustness of the obtained findings. Nevertheless, the major weakness is that the interpretation of the results is poor without providing insight and sufficient answers about the potential reasons which can explain the apparent differences. More specifically, throughout the manuscript the authors are restricted just to a description of the figures which can be easily done by a reader without reading the text. Therefore, I strongly believe that the manuscript needs a major revision before it can be acceptable for publication in ACP. Below are listed my comments/questions which I hope will help the authors to improve their work.

Response: We appreciate the time and effort the reviewer spent on this manuscript and the insightful comments and constructive suggestions. In light of your opinion, we have carefully revised our manuscript. The responses to the questions raised in your report are as follows.

1. Which version of the AERONET data is utilized?

Response: We use the newly released AERONET Version 3 Level 2.0 monthly AOD observations in this study, and we have clarified this information in Section 2.2.

2. You have to provide a better description of the satellite datasets (version, spatial resolution, temporal resolution, temporal availability, where these data are stored, literature etc.).

Response: We have provided more detailed descriptions (including the data version, spatial and temporal resolution, temporal availability, scientific dataset, and literature) of the satellite-derived aerosol products in the revised version according to your suggestion. Meanwhile, the data acquisition addresses are provided in the acknowledgements.

3. Page 6 – Lines 177-179: This sentence is confusing for me. Are you using monthly means or daily retrievals which are used in order to calculate the monthly averages? What do you mean "…with sufficiently high-quality…"?Are you applying any quality assurance flag or are you using the raw data as is?

Response: We apologize for the confusing sentence. In the paper, we did not apply any additional quality assurance and used the original monthly products for all analyses. The mentioned quality assurance flag is only a type of output control in the MODIS aerosol retrieval algorithm (Levy et al., 2013). We have removed this sentence from the revised version.

4. Page 6 – Lines 180-181: Please rephrase this sentence.

Response: We have rephrased the sentence "For multi-satellite aerosol products, the monthly retrievals at 550 nm are collected from the listed scientific dataset (SDS, Table 1) and used for the current analysis in this study" in the revision.

5. Figure 2: I cannot understand why the comparison versus AERONET is made for the periods where each dataset is available and not for the common period (Table 2 and 3). In the scatterplots, the EE dashed lines are common for all satellite data. This is not correct since each satellite sensor has different uncertainty limits (which are not stated in the text).

Response: We have removed the comparison for the period of each dataset and retained the common-period comparisons in the revision according to your suggestion. The problem regarding the EE dashed lines is explained below in the answer to question 7.

6. Page 8 – Lines 228-244: Is there any interpretation for these results? The authors must consider previous evaluation analyses in their discussion.

Response: We have compared our results with the results of previous studies on the four ESA-CCI products in the paper (Section 4.1). However, for the remaining aerosol products, we used the newest versions that have been released recently (e.g., MODIS C6.1 and AVHRR products available in October 2017; MISR V23 in November 2017; VIIRS V1 in February 2018). Meanwhile, most published studies focus on the validation of the instantaneous retrievals of Level 2 products against surface measurements. Comparative studies on Level 3 monthly products are rare, and we did not find similar evaluation papers; thus, we did not make such comparisons in the current study.

7. Section 4.2: You have to repeat the analysis for EE using the corresponding limits for each satellite sensor. Moreover, you have to compare your results with other existing works.

Response: We have removed the EE quantity throughout the analysis due to its limitations for different satellite monthly aerosol products according to the suggestions from two reviewers.

8. Section 5.1: There are several points which must be discussed in Figure 7. For example, the differences among AATSR-ORAC, AATSR-SU, MODIS and SeaWIFS recorded across N. Africa. Likewise, in E. Asia, it seems that there is a strong diversity, in terms of AOD values, among the datasets. In AATSR-ORAC, there is an abrupt change of AODs between maritime and continental areas in the eastern tropical Atlantic Ocean as well as in the Arabian Sea. Finally, it would be useful to reproduce the maps by considering common points in all datasets separately over land (exclude AATSR in order to have available observations over Sahara and in the Middle East) and sea.

Response: We have added a discussion on this issue as "There is also strong diversity in the seasonal mean AODS over North Africa and East Asia among most datasets.

This diversity is mainly due to the different aerosol algorithms applied over bright surfaces (i.e., desert and urban areas). Both high surface reflectance and complex underlying surfaces increase the difficulty of aerosol retrieval (Wei et al., 2018)" in the revision. There was a mistake when processing the AATSR-ORAC product, and we have corrected and fixed the problem you mentioned. Meanwhile, we have reproduced seasonal maps for land and ocean in the Supplement File (Figures S2-3) following your suggestion.

9. Figure 9: For the computation of the regional means based on the satellite observations are used all the grid cells of the domain of interest or only the pixels in which AERONET stations reside? Why there is an increasing trend for MODIS data in EAA as well as in EUR? On the contrary, in SAA the agreement between MODIS and AERONET improves gradually. Why this is happening?
Response: In the original manuscript, we used only the pixels located over each AERONET station. According to the comment from another reviewer, this is not a validation but a comparison because we use the annual averages, not the instantaneous values, which may be the main reason for these uninterpretable trends. The analysis makes little sense; thus, we have deleted this part in the revision.

10. Section 6.4: Are your results in agreement with other similar studies? In the global map, there are clear signals over wide areas of the planet which are not discussed appropriately in the text. Which factors regulate (meteorology, emissions, teleconnections, land use, etc.) the obtained pattern?
Response: Thank you for your suggestion. We have compared our results with the results of other studies and discussed the main factors regulating the present AOD spatial patterns in the revised version (Section 6.3).

11. Figure 1: First of all, there are mistakes on the region names. Please correct the European Coast as well as the South Africa (it is not in Asia!). Which is the domain for the European Coast? Replace Atlantic Ocean with South Atlantic Ocean.
Response: We apologize for these mistakes, and we have corrected them according to your suggestions. The European coast mainly includes the Eastern European Sea and Mediterranean Sea. To make the border clearer, we have replotted Figure 1 in the revision.

12. Figure 11: Replace 2017 with 2010.
Response: This information has been corrected.

13. Page 3 – Lines 64-77: In this part of the manuscript the authors are stating only studies representative for China. Satellite observations have been also used for other regions of the planet such as the Mediterranean, Europe, Atlantic Ocean etc.
Response: Thank you for your suggestion. We have enriched the introduction and added satellite-based AOD research over Europe, the Mediterranean Sea, Northern Africa, Topical Pacific, North and South Atlantic Oceans in the revised version.

---

## Author Response (AR1)

Reviewer: 2

The authors take 9 satellite aerosol optical depth (AOD) monthly mean data sets, and perform comparisons against each other and AERONET monthly mean data. These come from a variety of
5   satellite instruments and algorithms. They look at similarities in spatial and temporal patterns. This research area is important because understanding aerosol influences on the Earth requires understanding the strengths and limitations of each data set.

This is a pretty big task and it is good to see it being tackled, because as the authors note there has not been a great deal of attention to data set choice in some satellite analyses. However, I think this version
10  of the paper has problems. The statistics and analysis are very superficial, and the metrics used do not always make sense or are incorrect. For example, autocorrelation and false discovery rate are ignored, a level 2 error metric is used for level 3 analysis. The terminology has errors in some sections (e.g. "validation" when this is not a validation analysis). And in several places the authors omit relevant references and use out of date ones, or instead insert excessive self-citations. There is also a possible
15  wavelength issue with the AVHRR product used.

I recommend major revisions and would like to review the revised version. This paper felt to me like the authors just downloaded a bunch of data and ran a bunch of statistical metrics against it, without thinking about what was being done or why. I suggest that when revising, they focus on what science question they are trying to answer, and then figure out the right tools to answer it and provide a detailed
20  discussion. Otherwise this feels not like a scientific research paper but rather the output of some automated data processing software.

After writing this review, I read the other two comments currently posted on ACPD for this paper. I generally agree with the other reviewers' comments.

Response: We appreciate the time and effort the reviewer spent on this manuscript, as well as their
25  insightful and constructive suggestions. In light of your opinion, we have carefully revised our manuscript. The responses to the questions raised in your report are as follows.

My comments in support of my recommendation are as follows:

1. Line 20, and elsewhere: Operational is not the right word here. It implies something produces as part

30 of routine agency operations while a mission is ongoing. Most of the products do not fit that definition; in fact, I think only MODIS and AVHRR do as they are produced with a few hours latency to support assimilation applications. I suggest deleting this word throughout.

Response: Thank you for your suggestion. We have deleted this word throughout the paper.

35 2. Title, line 30, line 35 and elsewhere: Terms like "significant inconsistencies" (or just "inconsistencies" alone), "seriously" are used a lot in this paper. But most of the time they are used as "weasel words", i.e. in a non-specific way which can lead people to get a certain impression which is not necessarily warranted. For example, "inconsistencies". Taken to an extreme, any two data sets will not be identical so are to some extent "inconsistent". The relevant question is, for any particular application, is the level

40 of consistency between them sufficient? For example, if one wants to look at seasonal variations, AOD magnitude might not be as important as the pattern throughout the year. But if one wants to look at radiative effects, magnitude is more important. If one wants to see large-scale features, then a broader swath to improve sampling at the expense of some accuracy might be desirable. The point is that these are all different instruments with different characteristics. We expect them to not be identical. The

45 wording in this paper (these examples and elsewhere) seems designed to send a message that aerosol remote sensing has big problems. In my opinion, that's an overly pessimistic assessment. There are differences but in general the reasons for those are understood. So which data set is best to use for a given study depends on the type of science question you are trying to answer. There is no "best" data set. This recent paper by Sayer et al in JGR

50 (https://agupubs.onlinelibrary.wiley.com/doi/10.1029/2018JD029465) covers some similar ground to the current study, in that part of it compares time series and maps of various over-water satellite AOD data sets. That paper goes into a lot of discussion about the differences between them and why they might be. So although there is a lot of diversity in the over-water AOD, the reasons are generally known. My personal opinion is that over much of the world, differences are probably more due to sampling

55 differences (swath and pixel selection) than algorithm. I suggest the authors refer to that paper in their revised manuscript and go from describing things as "inconsistent" to try to for example make

recommendations as to which data sets might be better or worse suited for different applications. Recommendations like that, with evidence, are more useful than just declaring "inconsistency". Perhaps "comparisons" or "consistency assessment" is a better way to describe the analysis in title and text.

Response: Thank you for the constructive suggestion. We completely agree, and we carefully read the paper you mentioned above (Sayer et al., 2018). In the current version, we focused on describing the performances of multi-source aerosol products through comparisons with ground-based observations at different scales to determine the best product in terms of representing the temporal and spatial AOD variations. Moreover, we provide recommendations to users for the selection of these products for different applications according to your suggestion. We have also changed the title to "Inter-comparison in spatial distributions and temporal trends derived from multi-source satellite aerosol products".

3. Line 50: I think this should say 20th century, not 19th. I am not aware of any observation networks before the late 20th century. If there are, please provide references. Aerosol science didn't really start until John Aitken in the late 1800s.

Response: This term has been corrected.

4. Line 118: This should be "Holzer-Popp" not "Holzerpopp". The author's name is double barrelled.

Response: This name was corrected.

Line 121: This should be changed to indicate it is the NOAA AVHRR aerosol product. There is also a NASA GISS aerosol product (GACP), which is monthly-only and ocean-only, and a NASA Deep Blue aerosol product, which also covers land but is presently only available for limited time periods (I know 2006-2011 is available). It would be good to clarify what is used and why here. Deep Blue and GISS also provide 550 nm while NOAA AVHRR do not. Perhaps one of those could be added.

Response: Thank you for your suggestion. Due to mismatched wavelengths (630 nm) and missing land observations, we have abandoned the use of the NOAA AVHRR AOD product and replaced it with the newly updated NASA AVHRR AOD product (available from 2006 to 2011) in the revision.

85   5. Line 125: Authors should state more clearly here that they are using the 0.63-micron AOD (aot1 SDS), as it is important to note that this is different from the 550 nm AOD provided by most other data sets and would result in offsets dependent on aerosol type. The authors do not seem to mention this later in the paper (e.g. line 179 says the satellites are at 550 nm). Was the AVHRR AOD somehow extrapolated to 550 nm like the others? Or was it left at 630 nm and the wavelength dependence

90   neglected?

Response: Per your previous suggestions, we have abandoned the use of the NOAA AVHRR AOD product due to mismatched wavelengths (630 nm) and missing land observations and instead used the newly updated NASA AVHRR AOD product (550 nm) in this revision.

95   6. Line 139: Authors are missing references for the version 23 algorithm they are using here. Martonchik/Kalashnikova are out of date. The water approach is discussed by Witek et al (2018): https://www.atmos-meas-tech.net/11/429/2018/ The land approach is discussed by Garay et al (2017): https://www.atmos-chem-phys.net/17/5095/2017/ I suggest authors read and cite these papers, since it appears they have been referring to older documents.

100   Response: Thank you for pointing out the out-of-date reference. We have carefully read these papers and cited them instead of the older documents in the paper according to your suggestions.

7. Line 155: Sayer et al (2014): https://agupubs.onlinelibrary.wiley.com/doi/full/10.1002/2014JD022453 is a more complete reference

105   for the DTB products than Levy et al (2013). It also provides a comparison for DB, DT, and DTB. It will also be useful for the authors' analysis since it provides similar discussion about the level of consistency between the data sets. All the papers cited here are about Collection 6 but I know the MODIS teams and they did not publish papers about Collection 6.1 yet (still in review).

Response: We have cited this more complete reference for the DTB products as well as a recently

110   published paper from our team (Wei et al., 2019, AE) for the description of the Collection 6.1 aerosol products in the paper per your suggestion.

8. Line 163: I am not sure that the FM acronym for "forward model" is needed here. I don't think it is used later.

Response: We have removed this acronym from the paper.

9. Line 166: Somewhere in this section I would add a note to state that this is not a validation but a comparison, because the authors are using monthly data and not instantaneous data. So there are sampling differences contribution as well as retrieval quality. The authors are not performing a true validation exercise here.

Response: Thank you for this point. We have added these descriptions to the paper (Section 2.2) and replaced the term "validation" with "comparison" following the suggestions from two reviewers.

10. Line 189: The authors insert four self-citations for a one-line equation developed by other people something like 75 years ago. This seems a little excessive. Please remove these citations or replace with ones to the original work by Angstrom.

Response: We have removed these citations from the paper.

11. Line 190: I recommend the authors account for lag 1-month autocorrelation in the time series. This is commonly done in AOD trend analyses as the data can be significantly autocorrelated on these scales (because large-scale systems and seasonal patterns can persist for weeks to months). This will keep the same trend values but affects the estimated uncertainties on the trend. See Weatherhead et al (1998): https://agupubs.onlinelibrary.wiley.com/doi/abs/10.1029/98JD00995 for examples how to calculate this.

Response: Thank you for the recommendation. We have accounted for the 1-month lag autocorrelation for all AOD time series analyses in the revision (Section 3.2 and 6.1) according to your suggestion.

12. Lines 197-200: I am not sure that "correct trend percentage" makes sense. If a trend is close to 0, you will end up with a lot of apparently "wrong" trends if the sign is wrong, even if the conclusion that there is almost no trend is correct. For example, if you had trends of +0.01 from AERONET and +0.1 from satellite the authors would say this is "correct" even though the difference is huge. But if you had

0.005 from AERONET and -0.005 from satellite the authors would classify it as "incorrect", even though they are both small and probably statistically indistinguishable within trend uncertainties. A further problem is that this makes the implicit assumption that AERONET trends are perfect when of course they also have some measurement uncertainty and sampling uncertainty. I suggest that a better metric would be to report the "consistent trend percentage". This could be calculated by checking whether the satellite and AERONET trends are consistent within each uncertainty or not. This is a more fair and statistically appropriate test. The authors could also report those situations in which the AERONET estimate is too uncertain to be useful. I doubt that five years is enough to estimate a trend robustly in many cases, due to significant annual variability. So quite possibly the uncertainty on the AERONET estimates even is quite high. I also wonder if seasonal trends would be better than annual, because we know that aerosol patterns show strong seasonal features (so trends in seasonal behavior could be masked in an annual trend analysis). The authors need to justify this more strongly.

Response: Thank you very much for your suggestions. We have modified and used the improved metric you mentioned to report the "consistent trend percentage" by checking whether the satellite and AERONET trends were consistent within each level of uncertainty (Section 3.3). We apologize for the misleading statements in the original version of the manuscript and have clarified this information in the revised version. All the trend analyses are based on the de-seasonalized time series of monthly AOD anomalies because the sample points at the annual and seasonal levels are not large enough to analyse the trends. Moreover, we have extended the study period from five years to eight years with approximately 96 monthly values, which is sufficient for long-term trend analysis according to previous studies. We have stressed on this in Section 3.2 in the revision.

13. Line 209: The subscripts are very long. I suggest replacing AOD_RETRIEVAL with AOD_R (for "retrieval") or AOD_S (for "satellite"), and AOD_AERONET with AOD_A. This will make it more readable.

Response: These terms were changed per your suggestion.

14. Line 210: Correlation is not useful when the data range is small compared to the uncertainty on the data. You could have a great data set but still have a small correlation. For example, over the open ocean AOD does not change much, so a low correlation is scientifically not much of a problem for most scientific applications, as long as bias and RMSE are low. The authors should note this because a lot of the maps and discussion rely on correlation.

Response: Thank you for your suggestion. We agree with your opinion about the correlation, and we have removed most of the discussion related to the correlation from the revised version.

15. Line 211: This EE is an expected envelope for level 2 error over land only, not for level 3 and not for water. It is not meaningful for level 3 data, and it is misleading to apply it that way. There is at present no error estimate for satellite level 3 products. I suggest the authors remove this quantity because it is misleading. In my view the other statistics are enough. This also requires removing from the discussion later on. Either remove it or create and justify some metric for what an acceptable EE on the monthly data is. My feeling is that a monthly level 3 EE should be smaller than the level 2 one, because some error sources should cancel out.

Response: Thank you for pointing out this problem, and we completely agree with your opinion. We have removed the EE quantity throughout the paper according to your suggestions.

16. Line 219, 230, 468, Table 2, Figure 10, and elsewhere: No, this is not a validation, it is a comparison, because you are using monthly mean products and not level 2. Validation requires a ground truth. There is no ground truth for monthly data because there is no instrument sampling continuous monthly data. AERONET is only a validation for level 2 data. The authors should change the wording because it is misleading, and word choice matters. The analysis the authors are doing here is fundamentally different from the dozens of published level 2 validation papers, and it is important not to muddle the issue.

Response: We have replaced the term "validation" with "comparison" throughout the paper according to your suggestion.

17. Line 295: Again, it is not ideal to provide a single self-citation here when these issues have been documented by many algorithm teams for many years.

Response: We have removed this citation from the paper.

18. Line 362: Throughout section 6 the authors talk a lot about trend significance. However, something which has been overlooked is that since there is multiple hypothesis testing going on (many data sets and locations are being tested for trends), there could be a significant fraction of false positives. See e.g. Wilks (2006) for more on this: https://journals.ametsoc.org/doi/10.1175/JAM2404.1 So, the authors should make some quantification about the expected false discovery rate. Further, statistical significance is only one factor. Figures 11 and 14 are a prime example of this problem. Scientific significance is another. If you get a trend of 0.001 with an uncertainty of 0.0001, that is statistically significant but scientifically not important because it is so small. But if you get a trend of 0.1 with an uncertainty of 0.1, that is not statistically significant by traditional tests, but is potentially very important, because 0.1 is a large potential trend. The authors here seem to focus on statistical significance and sign rather than actually looking at the numbers. This is quite superficial. I would like to see the whole section reconsidered.

Response: Thank you for your suggestion. We agree with your opinion, and we have added the false discovery rate (FDR) test to exclude the fraction of false positives in our trend significance analysis (Section 3.2). Moreover, we have shifted our focus from statistical significance to actual significance, and we have mainly explained the possible reasons for the regions where the aerosols changed significantly in Section 6.

19. Line 496: "Goddard", not "Godard".

Response: This name was corrected.

20. Figure 7 (and associated discussion): I do not like annual mean maps in general because AOD patterns and sampling are strongly dependent on season. So in some areas there will be a difference just because data are coming from different months. And in some areas things could look to be in closer

agreement than they really are, if biases in different seasons are opposite and can cancel out. Annual mean AOD is also not meaningful for most applications. I would prefer to see this figure and discussion instead as a composite of four sets of seasonal plots. This would be a closer to apples to apples comparison, and also allow an examination of seasonal variability.

Response: We have replaced the annual mean AOD maps with the four sets of seasonal AOD maps in the figures and provided associated discussions in this revision according to your suggestions.

21. Figure 8, 12: Could this be redrawn to show coloured symbols instead of bars? In some cases, the bars are overlapping and so it is hard to tell which is. It can also give misleading impressions. For example, in land ENAM the black and pink are overlapped. I guess black was drawn first and pink second, so pink is on top. So, the impression is that black is lower than pink, because we can only see the bottom of black. But in reality, because so much of black is hidden, it probably means that black and pink are very similar. Coloured symbols instead of bars would be clearer and easier to tell.

Response: We have redrawn these figures with coloured symbols according to your suggestions in the revision.

22. Figure 9: since this is not a validation but a comparison, it would be better to say "offset" rather than "bias" here. Bias implies an offset with reference to a truth, and we have no truth. Word choice is important.

Response: This term was corrected per your suggestion.

Reviewer: 3

The authors provide a comparison of nine satellite-derived global AOD data sets, with ground-based AERONET (land) and MAN (ocean) AOD data as reference. They apply different statistical metrics and look at the data sets on different spatial scales: global, regional and per reference site. They also look at trends. Differences and agreements between data sets are described. The manuscript provides an interesting overview of AOD data sets available in the public domain, although some recent data sets like those from VIIRS are missing. Also, I wonder why for AVHRR only the over-ocean AOD is included and the recent over-land data sets described by Sayer and Hsu in JGR, 2017, were not included. It would be interesting to see how these data sets, retrieved from a sensor not designed for aerosol retrieval, compares to those from dedicated sensors like MISR and MODIS. Likewise, a comparison with PARASOL (POLDER) would have been interesting. As regards the title, I would recommend changing "inconsistency" to "Intercomparison", because not all and not always are the data sets inconsistent, they are often also consistent.

Response: We appreciate the time and effort the reviewer spent on this manuscript and the insightful comments and constructive suggestions. In light of your opinion, we have carefully revised our manuscript. The responses to the questions raised in your report are as follows.
Regarding the AOD product selection, we actually collected the VIIRS aerosol product. However, VIIRS was launched in 2012, and its common matching period with most other satellite products was short. In this revision, we have added the VIIRS monthly product for a simple comparison of the spatial coverage and distributions in Figure 1. Because of the similar sensor parameters and algorithms of VIIRS and Aqua MODIS, both data products have close monthly spatial coverages and mean AODs values throughout the time series. Therefore, we did not include the VIIRS products in the inter-comparison in the following analysis. In addition, we have added the AVHRR and POLDER products over both land and ocean in the revision according to your suggestions. Meanwhile, we have modified the title to "Inter-comparison in spatial distributions and temporal trends derived from multi-source satellite aerosol products".

Specific comments (line numbers refer to the pdf published online)

1. 46: suggest "composition and short life time of atmospheric aerosol particles"

Response: This phrase was modified per your suggestion.

2. 57: remove "observable"

Response: This term was removed per your suggestion.

3. 79: remove "seemingly"

Response: This term was removed per your suggestion.

4. 80: This sentence suggests that some studies have indeed focused on exploring : : : ; hence references to these studies are needed here

Response: Thank you. We have cited the main references in the revision according to your suggestion.

5. 85: suggest "evaluation and comparison"

Response: This phrase was changed per your suggestion.

6. 91: validation

Response: This term was corrected.

7. 103: ADV was first published by Veefkind et al., 1998a, for retrieval over land; Over Ocean ASV was first developed by Veefkind et al., 1998b

Response: We have modified and cited these references in the paper.

8. 112: A more recent reference for the Swansea algorithm is Bevan et al., 2012

Response: The relevant reference has now been cited in the paper.

9. 117: Holzer-Popp

Response: This name has been corrected.

10. 122: "AVHRR aerosol product is only available": this is NOT true, see my general comment and references to Sayer and Hsu Sect.

Response: Yes, according to the suggestions by two reviewers, we have replaced the ocean-only AVHRR data with the NASA AVHRR aerosol product as you mentioned (Sayer and Hsu), which is available over both land and ocean. We have also rephrased this paragraph in the paper.

11. 2.2: not only AERONET is used, AOD over ocean is provided by the Marine Aerosol Network (Smirnov et al., 2009)

Response: We have cited this reference and revised the statement in the paper.

12. 196: could you reword the text to make clearer how the lsq fit is applied

Response: We have rephrased this sentence and cited the main reference on the LSQ method in the paper.

13. 199: trend symbols: same direction of the trend Para starting at

Response: This has been corrected.

14. 230: An important indicator is also the EE, and the above and under EE which clearly indicate overestimation (e.g. for MODIS) and underestimation (e.g. for MISR). Here and in the next paragraphs, I do not understand how MISR can have a similar number of collocations as MODIS in spite of its much smaller swath; MISR should have an N similar to AATSR

Response: We apologize for the incorrect statistics, and we have corrected the sentence to "The Terra MISR product provides a sample size of 8418, which is smaller than the Terra MODIS sample size (N = 9196) and is possibly due to the narrower swath width" in the revised version.

15. 235: I am not sure that your judgement of ADV is completely fair, since indeed MAE and RMSE are worse, but not EE; looking at the statistics in Table 2, it seems that none of the sensors has the best statistics for all numbers, so it is hard to make such statements.

Response: We apologize for the improper description here, and we have removed the statement from the revision.

16. 236: smaller number of retrievals collected: I think this should be a smaller number of collocation pairs since less references data are available; again, how can MISR provide a similar number of data collections as MODIS?

Response: Yes, the small number of data is due to the limited availability of reference data. We have corrected the description in the revision.

17. 244: SeaWiFS is not improved, but its performance is better

Response: This information was revised per your suggestion.

18. 251: what is the statistical parameter indicating estimation uncertainty and accuracy?

Response: In this paper, the accuracy is represented by MAE, and the uncertainty is represented by RMSE and RMB, where RMB > 1.0 or RMB < 1.0 indicate the over- or under-estimation uncertainty. We have clarified this information in Section 3.3 in the revision.

19. 257 and 263 and 275-277: a high R does not imply that the performance is better: MODIS has high R, but figure 2 shows that MODIS overestimates, so actually it's performance in estimating AOD is not so good. This should be re-worded in the text.

Response: We agree with your opinion and revised this information in the results analysis. We mainly use the following indicators (i.e., MAE, RMSE and RMB) to describe the product performance and no longer use the correlation (R) according to the suggestions from two reviewers.

20. 266: RSA, typo and you mean ESA?

Response: This term has been corrected.

360    21. Sect. 5.2: there are very large differences in the mean AOD values; yet they all compare well with AERONET (Fig. 2 and 3): why are these differences not visible in the scatterplots?

Response: These figures have been corrected.

22. 304: suggest plotting the eight-year mean value in the figures

365    Response: This change has been implemented per your suggestion.

23. Sect. 5.3 title not clear: suggest changing the Section title to " Comparison of satellite- and AERONET- derived annual mean AOD at each site

Response: The title has been modified per your suggestion.

370

24. 340: this sentence is not accurate: you compare annual mean AOD for each satellite over an AERONET sites with the AERONET annual mean value

Response: We have revised the sentence as follows: "Furthermore, we also compare the annual mean AODs calculated from each satellite product and AERONET throughout the world from 2003 to 2010

375    (Figure 9)."

25. 375-376: I do not understand the sentence "Four : : : areas." Why are the first 4 similar and the other 2 consistent? What do you mean with that? MYD08 and SeaWiFs show quite some differences. Could you re-word so it is clearer?

380    Response: We apologize for the unclear description, and we have rephrased the descriptions to "On the other hand, the four ESA-CCI and MISR aerosol products are not significant in most ocean areas, even for the open seas. MODIS and SeaWiFS products have similar spatial patterns in most ocean areas, such as the significantly increasing trends observed over the Pacific and Indian Oceans" in the revised version.

385

26. 379: what do you mean with "treatment in neighboring pixels": did you describe that in the text?

Response: We apologize for the incorrect description, and we have removed this information from the revised version.

27. Sect. 6.4: Linear trends were fitted, so it may be that upward and downward trends are compensated over this long period of 18 years and thus the trends in Fig 14 are not representative. Could you please add a comment in the text?

Response: We have added this comment in the revised version according to your suggestion.

28. Figure Captions: 2 and 3: Density scatterplot of the monthly averages of satellite-derived AOD (operational products) versus AERONET AOD

Response: These captions have been corrected.

8: replace aerosols with AOD

Response: This term has been corrected.

10: ".. with annual mean AERONET AOD data for all sites : : :"

Response: This phrase has been corrected.

11: trends of AOD at 550 nm

Response: This phrase has been corrected.

12: replace "aerosol trends" with "trends of AOD at 550 nm"

Response: This phrase has been corrected.

13: I think you show trends of AOD at 550 nm, not annual mean aerosols?

Response: This phrase has been corrected.

15: remove "variations"

415 Response: This term has been removed.

Based on the above conclusions and considering the time length, the Terra MODIS product is selected as a representative to study the aerosol variations over the past two decades. Figure 14 plots the global spatial distribution of the linear MOD08 AOD$_S$ trends from January 2000 to December 2017 using the same approach as in Section 6.1, and Table 5 shows the regional AOD$_S$ trends and uncertainties. Note that the upward and downward trends could be offset over such a long period of 18 years.

The MOD08 AOD$_S$ trends are generally weak. The average trend over the entire land area is 0.0001 yr$^{-1}$ and is not statistically significant. However, the trends in some specific land regions are worth noting. For example, fast-developing countries such as India in South Asia (a = 0.0027 ± 0.0010 yr$^{-1}$ and $p < 0.05$) and the North China Plain in East Asia show significantly increasing aerosol trends. The main reason for these trends is the acceleration of urbanization and increasing anthropogenic pollutant emissions caused by intense human activities (e.g., industrial pollution, fossil fuel combustion and straw burning), which have also been reported in previous studies (Lu et al., 2011; de Meij, et al., 2012; Suresh et al., 2013; Sogacheva et al., 2018). In dust dominant regions such as the Middle East, a significantly positive trend (a = 0.0023 ± 0.0012 yr$^{-1}$, $p < 0.05$) is also observed due to enhanced dust emissions associated with unfavourable meteorological conditions (e.g., increasing temperature and decreasing relative humidity) (Hsu et al., 2012; Klingmüller et al., 2016). Meanwhile, the increasing trends in western North America and central Africa can be attributed to the biomass burning of forest fires (Edwards et al., 2006; Gavin et al., 2007; Kondo et al., 2011; Das et al., 2017). In contrast, significantly negative trends are found over eastern North America (-0.0009 ± 0.0004 yr$^{-1}$, $p < 0.05$), Europe (-0.0014 ± 0.0005 yr$^{-1}$, $p < 0.05$), central South America, central and southeast China, and Japan in East Asia (< -0.01 yr$^{-1}$, $p < 0.05$). These results are in good agreement with the results of other studies, and these negative trends are mainly due to the favourable climatic conditions and the decrease in pollution aerosols associated with government emissions control (Hsu et al., 2012; de Meij et al., 2012; Hu et al., 2017; Li et al., 2019).

1045 Over most of the global ocean, MOD08 $AOD_S$ shows an obvious increasing trend (0.0005 yr$^{-1}$, p < 0.05). At the regional scale, the Pacific Ocean (a = 0.0009 ± 0.0002 yr$^{-1}$, p < 0.05), South Atlantic Ocean (a = 0.0013 ± 0.0003 yr$^{-1}$, p < 0.05), Indian Ocean (a = 0.008 ± 0.0002 yr$^{-1}$, p < 0.05), and coastal areas of South Asia (a = 0.0042 ± 0.0008 yr$^{-1}$, p < 0.05) have notable positive trends. These results are comparable to the results of previous studies (Hsu et al., 2012; Sayer et al., 2018), and the main reason is the transport of mineral dust and smoke from biomass burning (Edwards et al., 2006; Das et al., 2017).

1050 In contrast, significantly negative trends are found over the coastal areas of eastern North America (−0.0019 ± 0.0004 yr$^{-1}$, p < 0.05), Europe (a = −0.0011 ± 0.0003 yr$^{-1}$, p < 0.05), and western South America. The reduction in aerosols over these areas is mainly due to the decreased dust transport from the Sahara and the control/reduction of pollutant emissions by human activities (Hsu et al., 2012; Sayer et al., 2018). Overall, the temporal variations in global aerosol loads are strongly influenced by both natural and human sources, which need to be further investigated in our future studies.

1055 *[Please insert Figure 15 here]*

*[Please insert Table 5 here]*

**7 Summary and conclusion**

[revised manuscript text omitted]

Das, S., Harshvardhan, H., Bian, H., Chin, M., Curci, G., Protonotariou, A. P., et al. (2017). Biomass burning aerosol transport and vertical distribution over the South African-Atlantic region. Journal of Geophysical Research: Atmospheres, 122, 6391–6415. https://doi.org/10.1002/2016JD026421

de Leeuw, G., Holzer-Popp, T., Bevan, S., Davies, W. H., Descloitres, J., & Grainger, R. G., et al. (2015). Evaluation of seven European aerosol optical depth retrieval algorithms for climate analysis. Remote Sensing of Environment, 162, 295-315.

de Meij, A., Pozzer, A., & Lelieveld, J. (2012). Trend analysis in aerosol optical depths and pollutant emission estimates between 2000 and 2009. Atmospheric Environment, 51, 75-85.

Dubovik, O., Herman, M., Holdak, A., Lapyonok, T., Tanre, D., Deuz ˙ e, J. L., et al. (2011). Stastically optimized inversion algorithm for enhanced retrieval of aerosol properties from spectral multi-angle polarimetric satellite observations. Atmospheric Measurement Techniques, 4, 975–1018. https://doi.org/10.5194/amt-4-975-2011

Dubovik, O., Lapyonok, T., Litvinov, P., Herman, M., Fuertes, D., Ducos, F., et al. (2014). GRASP: A versatile algorithm for characterizing the atmosphere. Newsroom: SPIE. https://doi.org/10.1117/2.1201408.005558

Edwards, D. P., Emmons, L. K., Gille, J. C., Chu, A., Attié, J.-L., Wood, S. W., et al. (2006). Satellite-observed pollution from Southern Hemisphere biomass burning. Journal of Geophysical Research, 111, D14312. https://doi.org/10.1029/2005JD00665

1140 Floutsi, A. A., Korrascarraca, M. B., Matsoukas, C., Hatzianastassiou, N., & Biskos, G. (2016). Climatology and trends of aerosol optical depth over the mediterranean basin during the last 12 years (2002-2014) based on Collection 006 MODIS-Aqua data. Science of the Total Environment, 551-552, 292-303.

Gavin, D. G., Hallett, D. J., Hu, F. S., Lertzman, K. P., Prichard, S. J., & Brown, K. J., et al. (2007). Forest fire and climate change in western North America: insights from sediment charcoal records. Frontiers in Ecology and the
1145 Environment, 5(9), 499-506.

[revised manuscript text omitted]

Sayer, A. M., Hsu, N. C., Lee, J., Kim, W. V., Dubovik, O., Dutcher, S. T. et al. (2018). Validation of SOAR VIIRS over-water aerosol retrievals and context within the global satellite aerosol data record. Journal of Geophysical Research:

1240   Atmospheres, 123, 13,496–13,526. https://doi.org/10.1029/2018JD029465

Sayer, A. M., N. C. Hsu, J. Lee, N. Carletta, S.-H. Chen, and A. Smirnov (2017), Evaluation of NASA Deep Blue/SOAR aerosol retrieval algorithms applied to AVHRR measurements, Journal of Geophysical Research: Atmospheres, 122, doi:10.1002/2017JD026934

Sayer, A., Hsu, N., Lee, J., Kim, W., Dubovik, O., Dutcher, S., Huang, D., Litvinov, P., Lyapustin, A., Tackett, J., Winker, D.

1245   (2018). Validation of SOAR VIIRS Over-Water Aerosol Retrievals and Context Within the Global Satellite Aerosol Data Record. Journal of Geophysical Research: Atmospheres, 123, 13496–13526.

Sayer, A.M., Poulsen, C. A., Arnold, C., Campmany, E., Dean, S., Ewen, G. B.L., et al. (2011). Global retrieval of ATSR cloud parameters and evaluation (GRAPE): dataset assessment. Atmospheric Chemistry and Physics, 11, 3913–3936. http://dx.doi.org/10.5194/acp-11-3913-2011

1250   Smirnov, A., Holben, B. N., Eck, T. F., Dubovik, O., & Slutsker, I. (2000). Cloud-screening and quality control algorithms for the AERONET database. Remote Sensing of Environment, 73(3), 337–349.

Smirnov, A., Holben, B. N., Slutsker, I., Giles, D. M., McClain, C. R., Eck, T. F., et al. (2009). Maritime Aerosol Network as a component of Aerosol Robotic Network. Journal of Geophysical Research, 112, D06204. https://doi.org/10.1029/2008JD011257

1255   Sogacheva, L., de Leeuw, G., Rodriguez, E., Kolmonen, P., Georgoulias, A. K., Alexandri, G., Kourtidis, K., Proestakis, E., Marinou, E., Amiridis, V., Xue, Y., and van der A, R. J.: Spatial and seasonal variations of aerosols over China from two decades of multi-satellite observations – Part 1: ATSR (1995–2011) and MODIS C6.1 (2000–2017), Atmospheric Chemistry and Physics, 18, 11389-11407.

Suresh, B. S., Manoj, M. R. , Krishna, M. K. , Gogoi, M. M. , Nair, V. S. , & Kumar, K. S., et al. (2013). Trends in aerosol

1260   optical depth over Indian region: potential causes and impact indicators. Journal of Geophysical Research Atmospheres, 118(20), 11,794–11,806.

Thomas, G. E., Poulsen, C. A., Sayer, A.M., Marsh, S. H., Dean, S. M., Carboni, E., et al. (2009). The GRAPE aerosol retrieval algorithm. Atmospheric Measurement Techniques, 2, 679–701. http://dx.doi.org/10.5194/amt-2-679-2009.

Veefkind, J.P., de Leeuw, G., and, Durkee, P.A. (1998a). Retrieval of aerosol optical depth over land using two-angle view

1265   satellite radiometry during TARFOX. Geophysical Research Letters. 25(16), 3135-3138.

Veefkind, J.P. and de Leeuw, G. (1998b). A new algorithm to determine the spectral aerosol optical depth from satellite radiometer measurements. Journal of Aerosol Sciences, 29, 1237-1248.

Weatherhead, C., Reinsel, C., Tiao, C., Meng, L., Choi, S., & Cheang, K., et al. (1998). Factors affecting the detection of trends: statistical considerations and applications to environmental data. Journal of Geophysical Research Atmospheres, 103(15), 1241-1255.

Wei, J., Li, Z., Peng, Y., and Sun, L. (2019). MODIS Collection 6.1 aerosol optical depth products over land and ocean: validation and comparison. Atmospheric Environment, 201, 428-440. https://doi.org/10.1016/j.atmosenv.2018.12.004

Wei, J., Sun, L., Peng, Y., Wang, L., Zhang, Z., Bilal, M., & Ma, Y. (2018). An improved high-spatial-resolution aerosol retrieval algorithm for MODIS images over land. Journal of Geophysical Research: Atmospheres, 123, 12,291–12,307. https:// doi.org/10.1029/2017JD027795

Witek, M. L., Garay, M. J., Diner, D. J., Bull, M. A., and Seidel, F. C. (2018). New approach to the retrieval of AOD and its uncertainty from MISR observations over dark water, Atmospheric Measurement Techniques, 11, 429-439, https://doi.org/10.5194/amt-11-429-2018.

Wilks, & D., S. (2006). On "field significance" and the false discovery rate. Journal of Applied Meteorology and Climatology, 45(9), 1181-1189.

Zdaniuk, B. (2014). Ordinary Least-Squares (OLS) Model. Springer Netherlands.

Zhao, X.-P, T., Chan, & Pui, K. (2013). A global survey of the effect of cloud contamination on the aerosol optical thickness and its long-term trend derived from operational AVHRR satellite observations. Journal of Geophysical Research Atmospheres, 118(7), 2849-2857.

Table 1. Summary of satellite-derived and ground-observed monthly aerosol products used in this study

| Product | Version | Spatial resolution | Temporal Resolution | Temporal availability | Scientific Data Set | Literature |
|---------|---------|--------------------|--------------------|-----------------------|---------------------|------------|
| AATSR-ADV | V2.31 | 1°×1° | Monthly | 2002.05-2012.04 | AOD550_mean | Veefkind et al., 1998a, Veefkind and de Leeuw, 1998b |
| AATSR-SU | V4.3 | 1°×1° | Monthly | 2002.05-2012.04 | AOD550_mean | North, 1999; 2002; Bevan et al., 2012 |
| AATSR-ORAC | V4.01 | 1°×1° | Monthly | 2002.07-2012.04 | AOD550_mean | Thomas et al., 2009; Sayer et al., 2011; Poulsen et al., 2012 |
| AATSR-EN | V2.6 | 1°×1° | Monthly | 2002.07-2012.04 | AOD550 | Holzer-Popp et al., 2013 |
| MISR | V23 | 0.5°×0.5° | Monthly | 2000.03-2017.12 | Optical depth average (550 nm) | Garay et al., 2017; Witek et al., 2018 |
| MOD08 | C6.1 | 1°×1° | Monthly | 2000.03-2017.12 | AOD_550_Dark_Target_Deep_Blue_Combined_Mean | Sayer et al., 2014 |
| MYD08 | C6.1 | 1°×1° | Monthly | 2002.07-2017.12 | AOD_550_Dark_Target_Deep_Blue_Combined_Mean | Sayer et al., 2014 |
| SeaWiFS | V4 | 0.5°×0.5° | Monthly | 1997.09-2010.12 | aerosol_optical_thickness_550_land | Hsu et al., 2013; Sayer et al., 2012 |
| AVHRR (NOAA-18) | V1 | 1°×1° | Monthly | 2006.01-2011.12 | aerosol_optical_thickness_550_land_ocean_mean | Hsu et al., 2017 |
| VIIRS | V1 | 1°×1° | Monthly | 2012.03-2017.12 | Aerosol_Optical_Thickness_550_Land_Ocean_Mean | Hsu et al., 2013; Sayer et al., 2012 |
| POLDER | V1.1 | 1°×1° | Monthly | 2005.03-2013.10 | AOD550 | Dubovik et al., 2011, 2014 |
| AERONET | V3 | site | Monthly | 2003.01-2010.12 | AOD | Giles et al., 2019 |

Table 2. Comparison between satellite-derived and ground-based monthly AOD$_S$ values during 2006-2010 over land and ocean

| Products | Land | | | | | Ocean | | | | |
|---|---|---|---|---|---|---|---|---|---|---|
| Metrics | N | R | MAE | RMSE | RMB | N | R | MAE | RMSE | RMB |
| AATSR-ADV | 6979 | 0.734 | 0.086 | 0.153 | 0.868 | 959 | 0.712 | 0.068 | 0.100 | 1.149 |
| AATSR-EN | 7739 | 0.745 | 0.082 | 0.140 | 0.941 | 1023 | 0.711 | 0.061 | 0.093 | 1.063 |
| AATSR-ORAC | 8401 | 0.713 | 0.081 | 0.143 | 0.896 | 1066 | 0.696 | 0.069 | 0.100 | 1.204 |
| AATSR-SU | 7503 | 0.766 | 0.081 | 0.140 | 0.997 | 1026 | 0.693 | 0.058 | 0.098 | 0.988 |
| AVHRR | 7331 | 0.743 | 0.082 | 0.152 | 0.970 | 1051 | 0.783 | 0.047 | 0.078 | 1.004 |
| MISR | 7464 | 0.795 | 0.074 | 0.128 | 0.869 | 954 | 0.587 | 0.070 | 0.118 | 0.977 |
| MOD08 | 8108 | 0.875 | 0.074 | 0.113 | 1.162 | 1088 | 0.814 | 0.069 | 0.093 | 1.309 |
| MYD08 | 7945 | 0.870 | 0.068 | 0.110 | 1.090 | 1088 | 0.812 | 0.056 | 0.082 | 1.191 |
| POLDER | 7956 | 0.733 | 0.108 | 0.162 | 1.292 | 1027 | 0.694 | 0.071 | 0.110 | 1.247 |
| SeaWiFS | 4516 | 0.819 | 0.072 | 0.117 | 0.920 | 775 | 0.746 | 0.057 | 0.088 | 1.053 |

Table 3. Percentage of sites within certain ranges of evaluation metrics for different satellite-derived monthly AOD$_S$ products from 2006 to 2010

| Products | N | MAE | | RMSE | | RMB | | |
|---|---|---|---|---|---|---|---|---|
| | > 6 | < 0.08 | > 0.12 | < 0.08 | > 0.12 | < 0.8 | [0.9, 1.1] | > 1.2 |
| AATSR-ADV | 92 | 59 | 19 | 47 | 28 | 35 | 22 | 16 |
| AATSR-EN | 96 | 66 | 16 | 55 | 25 | 26 | 28 | 23 |
| AATSR-ORAC | 99 | 69 | 20 | 56 | 28 | 26 | 24 | 30 |
| AATSR-SU | 95 | 63 | 19 | 56 | 28 | 18 | 32 | 17 |
| AVHRR | 96 | 67 | 17 | 57 | 25 | 20 | 29 | 18 |
| MISR | 95 | 69 | 15 | 50 | 25 | 25 | 30 | 23 |
| MOD08 | 99 | 67 | 12 | 52 | 23 | 9 | 14 | 54 |
| MYD08 | 97 | 71 | 12 | 60 | 21 | 12 | 24 | 34 |
| POLDER | 93 | 35 | 31 | 21 | 47 | 2 | 17 | 61 |
| SeaWiFS | 79 | 56 | 14 | 46 | 21 | 12 | 27 | 24 |

Table 4. Seasonal statistics of spatial coverage and global means of satellite-derived AOD$_S$ from 2003 to 2010

| Products | Spatial coverage (%) | | | | Mean AOD | | | |
|---|---|---|---|---|---|---|---|---|
| | DJF | MAM | JJA | SON | DJF | MAM | JJA | SON |
| AATSR-ADV | 54 | 66 | 67 | 62 | 0.16±0.10 | 0.17±0.13 | 0.17±0.12 | 0.16±0.10 |
| AATSR-EN | 68 | 75 | 77 | 75 | 0.13±0.08 | 0.16±0.13 | 0.15±0.11 | 0.14±0.09 |
| AATSR-ORAC | 75 | 76 | 80 | 79 | 0.15±0.09 | 0.16±0.10 | 0.16±0.10 | 0.16±0.08 |
| AATSR-SU | 70 | 77 | 79 | 78 | 0.12±0.09 | 0.15±0.14 | 0.15±0.13 | 0.13±0.09 |
| AVHRR | 69 | 74 | 76 | 73 | 0.13±0.09 | 0.14±0.14 | 0.14±0.13 | 0.13±0.09 |
| MISR | 73 | 77 | 79 | 78 | 0.12±0.08 | 0.13±0.12 | 0.14±0.11 | 0.12±0.08 |
| MOD08 | 71 | 79 | 82 | 80 | 0.16±0.09 | 0.19±0.14 | 0.19±0.13 | 0.17±0.10 |
| MYD08 | 71 | 79 | 82 | 80 | 0.15±0.09 | 0.17±0.14 | 0.17±0.12 | 0.15±0.09 |
| POLDER | 64 | 63 | 66 | 66 | 0.19±0.13 | 0.20±0.15 | 0.21±0.15 | 0.19±0.12 |
| SeaWiFS | 65 | 71 | 72 | 72 | 0.10±0.08 | 0.12±0.11 | 0.13±0.12 | 0.11±0.08 |

Table 5. Regional trends and uncertainties of the Terra MODIS AOD$_S$ anomalies from 2000 to 2017, where ** and * indicate the trends significant at the 95% and 90% confidence levels, respectively.

| Land | | ENA | | WNA | | SAM | |
|---|---|---|---|---|---|---|---|
| Trend | Uncertainty | Trend | Uncertainty | Trend | Uncertainty | Trend | Uncertainty |
| 0.0001 | 0.0004 | -0.0009** | 0.0004 | 0.0005 | 0.0006 | -0.0009 | 0.0010 |
| EUR | | AFR | | ME | | EAA | |
| Trend | Uncertainty | Trend | Uncertainty | Trend | Uncertainty | Trend | Uncertainty |
| -0.0014** | 0.0005 | 0.0002 | 0.0005 | 0.0023* | 0.0012 | -0.0012 | 0.0011 |
| SAA | | SEA | | OCE | | Ocean | |
| Trend | Uncertainty | Trend | Uncertainty | Trend | Uncertainty | Trend | Uncertainty |
| 0.0036** | 0.0010 | 0.0010 | 0.0018 | 0.0000 | 0.0002 | 0.0005** | 0.0002 |
| PAO | | NAO | | SAO | | INO | |
| Trend | Uncertainty | Trend | Uncertainty | Trend | Uncertainty | Trend | Uncertainty |
| 0.0009** | 0.0002 | 0.0003 | 0.0004 | 0.0013** | 0.0003 | 0.0008** | 0.0002 |
| ENC | | EUC | | SAC | | EAC | |
| Trend | Uncertainty | Trend | Uncertainty | Trend | Uncertainty | Trend | Uncertainty |
| -0.0019** | 0.0004 | -0.0011** | 0.0003 | 0.0042** | 0.0008 | -0.0013 | 0.0008 |

[Figure]

Figure 1. Locations of the AERONET sites and geographical bounds of the custom regions used in this study, where red and green dots represent land and ocean sites, respectively.

[Figure]

1305 Figure 2. Density scatterplots of the monthly averages of satellite-derived $AOD_S$ versus AERONET $AOD_A$ throughout the world

[Figure]

1310 Figure 3. Continent-scale performance for satellite-derived monthly $AOD_S$ against AERONET monthly $AOD_A$ measurements from 2006 to 2010 in terms of (a) sample size (N), (b) MAE, (c) RMSE and (d) RMB

[Figure]

Figure 4. Site-scale performance map for satellite-derived monthly AOD$_S$ against AERONET monthly AOD$_A$ measurements from 2006 to 2010 in terms of (i) sample size (N), where black dots represent the sites with zero matchup samples, (ii) MAE, (iii) RMSE and (iv) RMB

[Figure]

Figure 5. Time series of global spatial coverage and mean value of satellite-derived monthly aerosol products for their respective available periods from 1997-2017.

[Figure]

Figure 6. Satellite-derived global seasonal averaged AOD$_S$ maps at 550 nm from 2003 to 2010

[Figure]

1325 Figure 7. AOD$_S$ spatial coverage (marked as solid circles) and seasonal mean (marked as hollow circles) for each customized region over land and ocean (refer to Figure 1) from 2003 to 2010.

[Figure]

Figure 8. Seasonal performance for satellite-derived $AOD_S$ against AERONET $AOD_A$ measurements from 2003 to 2010 in terms of (a) sample size (N), (b) MAE, (c) RMSE and (d) RMB, where numbers 1-10 on the X-axis represent the AATSR-ADV, AATSR-EN, AATSR-ORAC, AATSR-SU, AVHRR, MISR, MOD08, MYD08, POLDER, and SeaWiFS products, respectively.

1330

[Figure]

1335     Figure 9. Comparisons between the annual global mean satellite-derived AOD$_S$ and AERONET-based AOD$_A$ at 550 nm for all matchup sites throughout the world. The solid black line represents the 1:1 line.

[Figure]

Figure 10. Spatial distribution of autocorrelation coefficient with a lag of one month based on de-seasonalized monthly $AOD_S$ anomalies at 550 nm from 2003 to 2010.

[Figure]

1340

Figure 11. Linear trend based on de-seasonalized monthly $AOD_S$ anomalies from 2003 to 2010. Units are AOD yr$^{-1}$. Black dots indicate a significant trend at the 95% confidence level ($p < 0.05$).

[Figure]

Figure 12. Regional linear trends based on de-seasonalized monthly AOD$_S$ anomalies over land and ocean from 2003-2010, where the hollow and solid circles represent statistically nonsignificant and significant trends at the 95% confidence level (p < 0.05), respectively.

1345

[Figure]

Figure 13. Comparisons between the linear trends based on the de-seasonalized monthly AOD$_S$ anomalies from 2003-2010. Units are AOD decade$^{-1}$. The solid black line represents the 1:1 line.

[Figure]

1350

Figure 14. Linear trend based on the de-seasonalized monthly Terra MODIS AOD$_S$ anomalies from 2000-2017. Units are AOD yr$^{-1}$. Black dots indicate that the trend is significant at the 95% confidence level ($p < 0.05$).

---

## Author Response (AR2)

Reviewer: 2

For final publication, the manuscript should be reconsidered after major revisions. I would be willing to review the revised paper, if the editor considers it necessary.

5 Suggestions for revision or reasons for rejection (will be published if the paper is accepted for final publication)

I reviewed the previous version of this manuscript and had a number of scientific and technical comments. The authors have put a lot of work into this revision, addressing most of my concerns and those of the other reviewers. I appreciate that. The study is much improved as a result.

10

I do have some remaining comments on the paper, mostly because it has been substantially rewritten from last time. I recommend further revisions to address these. I would be happy to review the next version. I realize there are a lot of comments below, but that should not be taken negatively: this is important work, and the revised version really is a lot stronger than the original submission. I just want

15 to make sure that the final paper is fully correct and as strong as it reasonably could be. So please take these comments positively. I think the next frontier is how to best assess level 3 uncertainty and combine different data records and this analysis will be an important step in that direction, by helping lay out the current state of many well-used data sets.

Response: We appreciate for the time and efforts the reviewer spent on this manuscript and the

20 insightful comments and constructive suggestions. In the light of your opinion, we have revised our manuscript carefully. The questions raised in your report are responded as follows.

Line 18: this says 10 products but line 91 says 11. I count 11 satellite products in e.g. Table 1. Response: We have corrected it in the revision.

25

Line 55: Satellites have been launched since before the 1990s and the word 'continuous' feels a little strange as we really only get a new aerosol-capable mission every few years. It might be better to say

something like "For the last few decades, satellite instruments have been launched with increasing capability for remote sensing of aerosol measurements, which ...".

30 Response: We have rephrased it as "For the last few decades, satellite instruments have been launched with increasing capability for remote sensing of aerosol measurements, which have provided long-term data records with wide spatial coverage." in the revision.

Line 63: there are other instruments too so I would add "among others" at the end of this sentence.

35 Otherwise a non-expert reader might this this is a complete list.Response: We have added it in the revision.

Line 80: Tropical, not topical. Response: Corrected.

40

Line 97: It would be worth mentioning that while data until 2017 are used, many of these products (MISR. MODIS, VIIRS) are ongoing as the instruments are still returning data. Response: We have mentioned this in the revision.

45 Lines 117-119: aside from Thomas (2009), the other ORAC references (Sayer 2011, Poulsen 2012) are for ORAC cloud products and not ORAC aerosol products. Sayer AMT 2010 (https://doi.org/10.5194/amt-3-813-2010) and the Thomas (2009) paper might be better references for the aerosol data used here.

Response: We have carefully read and cited the reference in the revision. Thanks for pointing out it.

50

Line 142: I would say "dark land" for DT and "bright and dark land" for DB, as the spatial coverage of the two is quite different.

Response: We have corrected them in the revision.

55 Line 149: The SeaWiFS data base is only for SeaWiFS Deep Blue. MODIS Deep Blue has its own data base from MODIS measurements. I suggest replacing "using the SeaWiFS surface reflectance products" with "using atmospherically-corrected data from the long time series of measurements". This is more general to the Deep Blue products.

Response: We have rephrased it according to your suggestion in the revision.

60

Lines 149, 150: There is a new reference for DB C6.1 which can be used instead of Hsu et al (2004, 2006, 2013) here. That is Hsu et al (2019), https://doi.org/10.1029/2018JD029688. Some of the methodology has changed a lot from the older work. I realise that 2019 paper is very new so it is understandable the authors prepared their revision before seeing it.

65 Response: Thanks for informing us about the newly published paper. We have carefully read and cited the new reference and rephrased some of the methodology in the revision.

Line 150: I would also mention that the C6.1 DT land algorithm has an update to reduce biases in urban areas by using a different surface reflectance model there. This is described in Gupta et al (2016),

https://doi.org/10.5194/amt-9-3293-2016.
 Response: We have mentioned and cited the corresponding reference in the revision.

Line 155: the authors might cite Sayer et al (2019) https://doi.org/10.1029/2018JD029598 here in addition to their own work, as that is the Deep Blue team evaluation of the C6.1 data. Again I realise

3

75 this is very new.

Response: We have cited the new reference here in the revision.

Line 158-159: I'd add Hsu et al (2019) in here again since that paper covers VIIRS too. Response: We have cited it here in the revision.

Line 160: I'd add this Sayer (2018) https://doi.org/10.1002/2017JD027412 paper here as it covers the VIIRS application of SOAR. Note that this is a separate Sayer (2018) paper from the one already cited, so one should be 2018a and the other 2018b.

Response: We have cited this reference here in the revision.

**85**

Line 213: it might be worth noting that here sigma is the one standard deviation uncertainty. This is important because for the significance testing later the focus is on 90% or 95% significance thresholds, which are closer to the two standard deviation uncertainty. So it's good to make this clear. Response: We have mentioned this in the revision.

90

Line 219: I'd add another sentence here because p-values are often interpreted incorrectly, especially by non-specialist readers. Something like: "The p-value represents the probability of obtaining results at least as extreme as those found, under the null hypothesis of there being no relation between AOD and time."

95 Response: We have added this sentence here in the revision.

Line 220: I'd say "decrease" the fraction, not "exclude" the fraction. You can never know for certain if and where you have false positives, only estimate the number you can expect to have across your data set.

**100 Response: We have corrected it as "decrease" in the revision.**

Line 227: I'd expand on alpha a bit more here. My understanding is that this mean you expect that no more than 5% of your significant results are in fact false positives. But please check and clarify this. Response: We have clarified this according to your suggestion in the revision.

105

Line 228: I'd say "Statistical metrics" instead of "Statistical metric". Response: Corrected. Section 3.3: one criticism from several reviewers of the previous version was the use of expected errors

- (EE) for level 2 products on the level 3 data. In this version the authors have removed the EEs, which is good. However I think it might be worth mentioning that they exist in case non-expert readers wonder. So somewhere in here I would add a sentence like: "Although several satellite products provide an expected level of uncertainty on AOD, this refers to level 2 products and not the level 3 products used here. As a result these level 2 expected uncertainties are not applicable to studies like this. Level 3
- 115 uncertainty estimates have not yet been developed for these AOD products."
  Response: Thanks for the suggestion and we have added such sentences in the revision.

120

Line 236: this should be clarified. Consistent within one standard deviation or more? And compared by combining trend uncertainties linearly or in quadrature? (I think it should be in quadrature, if one can assume uncertainty estimates from different data sets are independent.)

Response: Yes, it is in quadrature. We have clarified this according to your suggestion in the revision.

Line 249: I would say "differences" rather than "estimation uncertainties". Also, looking at Table 2, I am not sure exactly which statistics are being referred to here? For example the text says of the

- 125 AATSRs, SU is best but ADV is worst. But it really depends on which statistic is of most interest. For example ORAC has lowest correlation over land, but has highest N. And some of the differences are tiny: is R of 0.766 (SU) really statistically distinguishable from 0.734 (ADV), for example? All of these summary statistics presented here are only estimates of the true population behavior, and no uncertainty estimates are provided for these correlations, biases, etc. I am concerned that much of early section 4.1
- 130 is over-interpreting small and possibly indistinguishable differences between very similar numbers. I do not think it is good to emphasize the global-scale comparison for that reason, particularly when there is such regional diversity (as the authors note at the start of section 4.2). The importance of not over-interpreting small global-scale statistics must at least be acknowledged in the text. I would rather see this shortened (and probably put the table in the Supplement to avoid over-interpretation by readers) and
- 135 put more attention to the regional analysis (i.e. move the relevant material out of the Supplement and into the main paper). Otherwise the authors have to justify why the global-scale numbers are important

for any particular purpose, and why it is relevant to give them to three decimal places when it is doubtful that these estimates are robust to that level of precision. If the authors want to keep Table 2, I'd assess the expected level of uncertainty on these statistics, and round them to that level. My intuition is

140 that this would be two decimal places in most cases.

Response: Thanks for your suggestion and we have replaced "estimation uncertainties" with "differences" here. We have moved Table 2 into the Supplement and shorten the analysis (i.e., remove the discussions on Table 2) on global scale in the revision. Meanwhile, we have also moved the relevant material from the Supplement into the main paper.

145

Line 261 (and again 299-300): the reason for low SeaWiFs coverage is partly the time period studied. The instrument operated from 1997-2010 but the comparison is only for 2006-2010. It had some temporary failures in 2008, 2009, and 2010 which cause missing monthly data. While only a small fraction of the record is missing out of the 13-year mission, pretty much all the gaps are in the 5-year

- 150 period the authors study. So the numbers here are not really representative of SeaWiFS as a whole. This is an example of over-interpreting results without looking for the bigger picture, and again should be acknowledged in the text. Using a different time period SeaWiFS would still have less coverage than some of the other instruments, but the difference would be a lot less stark. Later on (section 5.1) the authors acknowledge this but it's really a relevant point for this section too.
- 155 Response: Thanks for your suggestion and we have explained the reason for low SeaWiFS coverage in the revision.

Figure 4: As this contains 40 panels, it is somewhat tricky to make out the details. I appreciate what the authors are trying to do here: give a rough illustration of the statistical metrics across all of the sites and
data sets. But I think it is too hard to see. I am not sure how best to deal with this. One option would be to make it into 4 figures (one for each of N, MAE, RMSE, RMB) and arrange them 2 across by 5 down. That would give each panel 4 times the area it has now, which would make things easier to distinguish. Another option would be to put this plot in the continental-scale comparison, and instead of showing data for each site, color each continent with the continental-average results. That would get across the

- point of showing regional and inter-data set variability but since each panel would have only a dozen or so larger shaded areas instead of 300 dots, it would be more readable. Then the current Figure 4 could be moved to the SI. I will leave it up to the authors and Editor. My point here is, the goal of this figure is a good one, but 40 panels with about 300 different dots on each is too much for a single figure. We just can't see the details at that level.
- 170 Response: We have divided the Figure into four separate Figures (one for each of N, MAE, RMSE, RMB) to make it clearer according to your suggestions in the revision.

Figure 5 and discussion: how is this spatial coverage and AOD calculated? Is it area-weighted? The retrievals are done on equal-area grids but the L3 aggregates used here are equal-angle. High latitude

- 175 grid cells will therefore be weighted disproportionately high if the L3 coverage is just averaged. I think area-weighting (i.e. each grid cell is weighted by cosine (central latitude) is the better way to do it, since it's more representative of actual surface area on the Earth. Please clarify and replot if this was not what was done. The text does not say. My guess is that it might be a simple equal-angle average rather than area-weighted, as the maximum spatial coverage around 70% seems quite low for monthly products.
- 180 Response: Thanks very much for your suggestion and we have recalculated the spatial coverage using the area-weighting approach where each grid cell is weighted by cosine of central latitude for different aerosol products in the revision.

Table 4, Figure 7: same question and suggestion as for Figure 5.

185 Response: We have re-made all the Tables and Figures related to the spatial coverage in the revision.

Lines 431-432: I would delete this sentence. Weak autocorrelation doesn't really imply much about suitability for trend analysis. All it tells you is how strongly the data are autocorrelated. This is something that is part of the uncertainty analysis but by itself isn't informative for suitability.

7

190 Response: We have deleted this sentence in the revision.

Line 443: please define "a" explicitly here. I infer from later in the paragraph that it is the AOD trend per year? It needs to be stated clearly.

Response: We have defined the "a" in Section 3.2, which represents the AOD trend per year in the revision. We now add an indication here to readers.

Line 473: "lack of retrieval over land" is misleading. There are tens of thousands of retrievals over land. I suggest "lower coverage over land".

Response: We have rephrased it as "lower spatial coverage over land" in the revision.

200

Section 6.3: I would delete this section from the paper. I don't feel that it really fits with the rest, so it's just adding length. The authors state the results are similar to previous trend studies. So it's just going to add a page or two to the paper's length without much new material. I think it's better to keep the paper focused on the main topic, comparing the data sets.

205 Response: Thanks for your insightful suggestion and we have deleted this section from the revision.

Line 530-531: Typically "spatial continuity" refers to unrealistic jumps in the data values rather than gaps in the data (which is what the authors are talking about here). Also, it feels like the authors exaggerate on line 531 when they say MODIS provides "almost full coverage" when earlier figures say

- 210 the number is more like 70%, which I don't think is "almost full". I suggest the authors say about SeaWiFS and ADV that they have "lower spatial coverage". The comment about MODIS might be ok if the coverage is recalculated as an area-weighted instead (see earlier comment), but in my view "almost full" implies something like 90% or more. Otherwise I'd just say MODIS gives "highest coverage". I'd also give the numbers for the lowest and highest cases so the reader has a reminder how much
- 215 difference we're talking about.

Response: We have rephrased them and given the numbers for the spatial coverage according to your suggestions in the revision.

Line 543-546: Following from my earlier comment about section 6.3, I suggest deleting these sentences.

Conclusion: I think an extra paragraph or two needs to be added at the end. So we have done all this work but I am missing concrete suggestions for what to do next. In my view we've learned that on a monthly scale, coverage differs a lot between data sets. So if we want to get a bigger picture, what do

- 225 we do? Should we next try averaging products together on a daily or monthly scale? Or should we pick products based on time period? Should we try some bias correction? Where are the most important areas for algorithm teams to focus effort to improve coverage or decrease the diversity? These are the sorts of questions I think the Conclusion should discuss, to guide future research.
  Response: We have added a paragraph to mention about the possible future works in the Conclusion
- 230 part.

Reviewer: 4

For final publication, the manuscript should be accepted as is accepted as is.

235 Response: Thanks very much for accepting our manuscript.

[revised manuscript text omitted]

|        | Metrics     |               | Aerosol Product |                |              |           |          |          |           |  |  |
|--------|-------------|---------------|-----------------|----------------|--------------|-----------|----------|----------|-----------|--|--|
| Region |             | AATSR-
ADV | AATSR-
EN    | AATSR-
ORAC | AATSR-
SU | MISR      | MOD08    | MYD08    | SeaWiFS   |  |  |
| Land   | Trend       | -0.0009       | -0.0001         | 0.0002         | -0.0004      | -0.0002   | 0.0006   | 0.0012   | -0.0012   |  |  |
|        | Uncertainty | 0.0007        | 0.0005          | 0.0005         | 0.0006       | 0.0004    | 0.0007   | 0.0007   | 0.0007    |  |  |
| ENA    | Trend       | -0.0031**     | -0.0021**       | -0.0005        | -0.0031**    | -0.0019** | -0.0016  | -0.0016  | -0.0042** |  |  |
| LIUI   | Uncertainty | 0.0011        | 0.0009          | 0.0011         | 0.0010       | 0.0007    | 0.0012   | 0.0010   | 0.0008    |  |  |
| WNIA   | Trend       | -0.0008       | -0.0004         | 0.0010         | -0.0006      | -0.0005   | 0.0003   | -0.0001  | -0.0029** |  |  |
| WINA   | Uncertainty | 0.0013        | 0.0011          | 0.0012         | 0.0010       | 0.0009    | 0.0019   | 0.0018   | 0.0010    |  |  |
| CAM    | Trend       | -0.0021       | -0.0014         | -0.0010        | -0.0016      | -0.0015   | -0.0019  | -0.0011  | -0.0006   |  |  |
| SAM    | Uncertainty | 0.0031        | 0.0019          | 0.0017         | 0.0027       | 0.0022    | 0.0037   | 0.0034   | 0.0017    |  |  |
| ELID   | Trend       | -0.0021**     | -0.0018**       | -0.0007        | -0.0024**    | -0.0009   | 0.0000   | -0.0004  | -0.0031** |  |  |
| EUK    | Uncertainty | 0.0009        | 0.0009          | 0.0010         | 0.0010       | 0.0007    | 0.0010   | 0.0009   | 0.0011    |  |  |
| AED    | Trend       | -0.0005       | 0.0005          | -0.0007        | 0.0000       | 0.0000    | 0.0001   | 0.0017   | -0.0018   |  |  |
| АГК    | Uncertainty | 0.0012        | 0.0012          | 0.0009         | 0.0011       | 0.0010    | 0.0012   | 0.0013   | 0.0016    |  |  |
| ME     | Trend       | 0.0048**      | 0.0083**        | 0.0050**       | 0.0073**     | 0.0077**  | 0.0084** | 0.0111** | 0.0079**  |  |  |
| MIL    | Uncertainty | 0.0020        | 0.0024          | 0.0013         | 0.0022       | 0.0025    | 0.0036   | 0.0035   | 0.0025    |  |  |
| ЕЛА    | Trend       | -0.0011       | -0.0004         | -0.0001        | -0.0008      | -0.0019   | 0.0003   | 0.0009   | -0.0019   |  |  |
| LAA    | Uncertainty | 0.0037        | 0.0022          | 0.0021         | 0.0026       | 0.0028    | 0.0039   | 0.0038   | 0.0023    |  |  |
| 5 4 4  | Trend       | 0.0040**      | 0.0034*         | 0.0047**       | 0.0044**     | 0.0018    | 0.0037*  | 0.0046*  | -0.0044*  |  |  |
| SAA    | Uncertainty | 0.0024        | 0.0019          | 0.0014         | 0.0020       | 0.0017    | 0.0027   | 0.0028   | 0.0023    |  |  |
| SEA    | Trend       | -0.0059       | -0.0041         | -0.0041        | -0.0050      | -0.0020   | -0.0034  | -0.0020  | -0.0041** |  |  |
|        | Uncertainty | 0.0052        | 0.0030          | 0.0025         | 0.0037       | 0.0027    | 0.0054   | 0.0047   | 0.0019    |  |  |
| OCE    | Trend       | 0.0002        | 0.0001          | 0.0007         | 0.0000       | -0.0005   | 0.0004   | 0.0006   | -0.0004   |  |  |
| OCE    | Uncertainty | 0.0004        | 0.0007          | 0.0006         | 0.0012       | 0.0007    | 0.0006   | 0.0006   | 0.0003    |  |  |

Table 5. Same as Table 4 but for ocean.

|        |                      | Aerosol Product   |                   |                   |                   |                   |               |                   |                   |
|--------|----------------------|-------------------|-------------------|-------------------|-------------------|-------------------|---------------|-------------------|-------------------|
| Region | Metrics              | AATSR-
ADV     | AATSR-
EN      | AATSR-
ORAC    | AATSR-
SU      | MISR              | MOD08         | MYD08             | SeaWiFS           |
| Ocean  | Trend                | -0.0003           | -0.0004           | 0.0000            | -0.0004           | -0.0004           | 0.0009**      | 0.0006            | -0.0006           |
|        | Uncertainty          | 0.0004            | 0.0003            | 0.0005            | 0.0003            | 0.0003            | 0.0004        | 0.0004            | 0.0004            |
| РАО    | Trend                | 0.0002            | -0.0001           | 0.0001            | -0.0001           | -0.0002           | 0.0015**      | 0.0008**          | 0.0010**          |
|        | Uncertainty          | 0.0004            | 0.0002            | 0.0005            | 0.0003            | 0.0003            | 0.0004        | 0.0004            | 0.0004            |
| NAO    | Trend                | -0.0006           | 0.0001            | 0.0010            | 0.0005            | -0.0002           | 0.0021**      | 0.0019**          | -0.0006           |
|        | Uncertainty          | 0.0009            | 0.0007            | 0.0009            | 0.0006            | 0.0006            | 0.0010        | 0.0009            | 0.0006            |
| SAO    | Trend                | 0.0001            | 0.0000            | 0.0001            | -0.0002           | -0.0004           | 0.0014*       | 0.0011            | 0.0003            |
|        | Uncertainty          | 0.0007            | 0.0005            | 0.0007            | 0.0004            | 0.0004            | 0.0008        | 0.0007            | 0.0004            |
| INO    | Trend                | 0.0000            | -0.0001           | 0.0003            | -0.0001           | -0.0001           | 0.0012**      | 0.0008*           | 0.0007            |
|        | Uncertainty          | 0.0004            | 0.0003            | 0.0005            | 0.0004            | 0.0004            | 0.0005        | 0.0005            | 0.0005            |
| ENC    | Trend                | -0.0037**         | -0.0032**         | -0.0022**         | -0.0021**         | -0.0023**         | -0.0020**     | -0.0024**         | -0.0026**         |
|        | Uncertainty          | 0.0010            | 0.0008            | 0.0007            | 0.0007            | 0.0008            | 0.0008        | 0.0008            | 0.0007            |
| EUC    | Trend                | -0.0026**         | -0.0021**         | -0.0017           | -0.0021**         | -0.0018**         | -0.0008       | -0.0011           | -0.0025**         |
|        | Uncertainty          | 0.0011            | 0.0009            | 0.0011            | 0.0008            | 0.0008            | 0.0010        | 0.0009            | 0.0010            |
| SAC    | Trend                | 0.0041**          | 0.0030*           | 0.0055**          | 0.0019            | 0.0030**          | 0.0064**      | 0.0049**          | -0.0002           |
|        | Uncertainty          | 0.0019            | 0.0016            | 0.0026            | 0.0014            | 0.0016            | 0.0025        | 0.0023            | 0.0020            |
| EAC    | Trend
Uncertainty | -0.0030
0.0024 | -0.0025
0.0016 | -0.0011
0.0021 | -0.0010
0.0012 | -0.0029
0.0022 | 0.0008 0.0023 | -0.0001
0.0022 | -0.0026
0.0021 |

---

## Author Response (AR3)

Reviewer: 2
For final publication, the manuscript should be reconsidered after major revisions.
I would be willing to review the revised paper, if the editor considers it necessary.
For final publication, the manuscript should be
accepted subject to technical corrections

Suggestions for revision or reasons for rejection (will be published if the paper is accepted for final publication)
I appreciate the authors' additional revisions here. I think it really improves the clarity and utility of the study. I have a couple of tiny comments which could be addressed during the final file upload, but don't feel it rises to the level of another formal review process. So my personal opinion is to accept pending technical corrections as follows:

I would delete the sentence on lines 437-440 about autocorrelation magnitude, since the wording doesn't quite make sense/is potentially misleading, and I don't think the sentence is necessary.
Response: We have deleted the sentence in the revision.

The paper cited as Sayer et al (2015) was published in 2014.
Response: We have corrected it in the revision.

The 'Temporal Resolution' column in table 1 is not necessary (same value for all data sets, and mentioned in the text that it's monthly). So it can be deleted to make the table simpler to read.
Response: We have deleted the 'Temporal Resolution' column in the revision.

Tables 4, 5: As noted earlier I caution against over-emphasising p-values. Rather than denote both 95% and 90% significant results here, I'd just focus on one (maybe 95%) and remove the annotation for the other one. This will also keep things simpler. Most results significant at 90% also seem to be significant at the 95% level anyway.
Response: We have removed the annotations in Tables 4-5 according to your suggestions in the revision.